# Ago2/CAV1 interaction potentiates metastasis via controlling Ago2 localization and miRNA action

Meng-Chieh Lin[1,5], Wen-Hung Kuo[2,5], Shih-Yin Chen[1,3], Jing-Ya Hsu[1], Li-Yu Lu[1], Chen-Chi Wang[2], Yi-Ju Chen[1], Jia-Shiuan Tsai[1] & Hua-Jung Li [1,4 ✉]

## Abstract

Ago2 differentially regulates oncogenic and tumor-suppressive miRNAs in cancer cells. This discrepancy suggests a secondary event regulating Ago2/miRNA action in a context-dependent manner. We show here that a positive charge of Ago2 K212, that is preserved by SIR2-mediated Ago2 deacetylation in cancer cells, is responsible for the direct interaction between Ago2 and Caveolin-1 (CAV1). Through this interaction, CAV1 sequesters Ago2 on the plasma membranes and regulates miRNA-mediated translational repression in a compartment-dependent manner. Ago2/CAV1 interaction plays a role in miRNA-mediated mRNA suppression and in miRNA release via extracellular vesicles (EVs) from tumors into the circulation, which can be used as a biomarker of tumor progression. Increased Ago2/CAV1 interaction with tumor progression promotes aggressive cancer behaviors, including metastasis. Ago2/CAV1 interaction acts as a secondary event in miRNA-mediated suppression and increases the complexity of miRNA actions in cancer.

**Keywords** Argonaute-2; Caveolin-1; miRNA; Exosome; Metastasis
**Subject Categories** Cancer; RNA Biology

## Introduction

MicroRNAs silence gene expression through translational repression or the degradation of targeted mRNAs, and they regulate more than 60% of all mammalian mRNAs (Friedman et al, 2009). In addition to their action within cells, miRNAs are secreted through RNA-binding proteins or vesicles into the extracellular space to regulate cell physiology (Patton et al, 2015; Valadi et al, 2007). Altering the regulation of miRNA expression may result in developmental defects and cancer (Alvarez-Garcia and Miska, 2005; Esquela-Kerscher and Slack, 2006). In addition to their expression, the distribution of miRNAs can affect physiological and pathological cell behavior (Goldie et al, 2014).

Argonaute-2 (Ago2), an essential catalytic component of the RNA-induced silencing complex (RISC), plays a central role in RNA silencing processes (Meister et al, 2004). Because Ago2 exhibits nuclease activity and is required for RISC function, its location is indicative of miRNA actions. Processing bodies (PBs) are cellular structures in the cytoplasm that dynamically interact with miRNA machinery (Anderson and Kedersha, 2006). PBs are enriched with Ago2 (Cougot et al, 2004). Both miRNAs and their targets are recruited to PBs (Anderson and Kedersha, 2006). However, the amount of Ago2 in PBs accounts for only approximately 1.3% of total cytoplasmic Ago2 (Leung et al, 2006), and PB depletion does not affect miRNA-mediated repression in cells (Liu et al, 2005). These findings suggest that the major actions of miRNAs occur not in PBs but elsewhere. Therefore, the remaining 99% of Ago2 in cells remains to be localized.

Ago2 is overexpressed in certain types of cancer, such as breast cancer (Casey et al, 2019; Conger et al, 2016), colorectal cancer (Li et al, 2010; Papachristou et al, 2011), ovarian cancer (Vaksman et al, 2012), gastric cancer (Zhang et al, 2013), and glioma (Feng et al, 2014). By contrast, its expression is reduced in melanoma (Voller et al, 2013) and clear cell renal cell carcinoma (Lee et al, 2019). Ago2 also differentially regulates oncogenic and tumor-suppressive miRNAs in cancer cells. For example, in hepatocellular cancer cells, its overexpression selectively helps oncogenic miRNAs repress target genes without altering tumor-suppressive miRNAs (Zhang et al, 2014). These differences indicate that Ago2 plays a dual regulatory role in tumor progression. However, how this dual function of Ago2 is regulated in a context-dependent manner remains to be elucidated.

Caveolin-1 (CAV1) is an essential structural protein in lipid rafts that plays a critical role in signal transduction by directly interacting with numerous signaling proteins and regulating their activity. CAV1-interacting proteins include EGFR (Couet et al, 1997b), insulin receptor (Nystrom et al, 1999), eNOS (Garcia-Cardena et al, 1997), PDGFR (Yamamoto et al, 1999), PKC (Lin et al, 2003; Oka et al, 1997), PTEN (Xia et al, 2010), β-catenin (Mo et al, 2010), Cox-2 (Liou et al, 2001), androgen receptor (Lu et al, 2001), and TLR4 (Wang et al, 2009). Protein interactions with CAV1 are commonly mediated by the caveolin scaffolding domain (CSD), which consists of amino acids 82–101 at the N-terminal

[1]Institute of Cellular and System Medicine, National Health Research Institutes, Miaoli 35053, Taiwan. [2]Department of Surgery, National Taiwan University Hospital, Taipei 100229, Taiwan. [3]Institute of Biotechnology, National Tsing Hua University, Hsinchu 30013, Taiwan. [4]Program in Tissue Engineering and Regenerative Medicine, National Chung Hsing University, Taichung City 402, Taiwan. [5]These authors contributed equally: Meng-Chieh Lin, Wen-Hung Kuo. ✉E-mail: annli@nhri.edu.tw

region of CAV1 (Li et al, 1996; Li et al, 1995; Liu et al, 2002). These interactions are responsible for recruiting CAV1-interacting proteins, such as insulin receptor, Ras, glucagon-like peptide 1 receptor, and angiotensin II type 1 receptor, into the plasma membrane and regulating functions of the interacting proteins (Boothe et al, 2016; Song et al, 1996; Syme et al, 2006; Wyse et al, 2003).

In this study, we examined the differential interaction between Ago2 and CAV1 (hereinafter referred to as Ago2/CAV1 interaction) in normal epithelial cells and cancer cells, which can be regulated through the acetylation of Ago2 lysine 212. With this interaction in cancer cells, CAV1 sequesters Ago2 on the plasma membranes of cancer cells and regulates the process of selective Ago2/miRNA-mediated translational repression in a compartment-dependent manner. This interaction increases with tumor progression and promotes aggressive tumor behaviors, such as metastasis and miRNA release via extracellular vesicles (EVs). It also results in the plasma membrane-associated distribution of Ago2 in cancer cells and selective Ago2/miRNA-mediated translational repression in tumor progression. In this study, we identified the relationship between this unique Ago2/CAV1 interaction, Ago2 subcellular location, and cancer cell behavior.

## Results

### Ago2 directly binds to the scaffolding domain of caveolin-1

Ago2 catalyzes miRNA maturation, and its interaction with other proteins affects miRNA function in cells (Muller et al, 2019). In this study, we explored the Ago2/CAV1 interaction in cancer cells (i.e., A549 lung cancer cells, H1299 lung cancer cells, BxPC-3 pancreatic cancer cells, HCC1806 breast cancer cells, and 104-S, C4-2, and R-1 prostate cancer cells) and normal cells (i.e., BEAS-2B human lung epithelial cells, HPNE human pancreatic epithelial cells, and HMLE human mammary epithelial cells). Endogenous CAV1 proteins were coprecipitated with endogenous Ago2 in cancer cells, indicating that Ago2 interacted with CAV1 in cancer cells (Figs. 1A and EV1A). However, no interaction was observed between Ago2 and CAV1 in normal epithelial cells (Figs. 1A and EV1A), indicating that Ago2 interacted differently with CAV1 in cancer cells and normal epithelial cells.

To examine direct Ago2/CAV1 interaction, two recombinant proteins, Ago2-His and CAV1-His, were mixed and incubated in test tubes. As shown in Figs. 1B and EV1B,C, the Ago2-His proteins in the mixture were precipitated with anti-Ago2 antibodies, and the CAV1-His proteins were coprecipitated from the mixture. The relevant findings indicated that Ago2 proteins directly bind to CAV1 proteins. In the next step, the protein domain of CAV1, which mediates the interaction between CAV1 with Ago2, was mapped out. In HEK293 cells, both full-length and truncated CAV1, namely CAV1-Flag(61–178), CAV1-Flag(102–178), CAV1-Flag($\Delta$61–84), and CAV1-Flag($\Delta$84–102), were expressed with HA-Ago2 proteins. Among the four truncated CAV1, only CAV1-Flag(61–178) and CAV1-Flag($\Delta$61–84) were coprecipitated with HA-Ago2 proteins (Figs. 1C and EV1D), suggesting the amino acid fragments containing CSD(82–101) of CAV1 were required for the interaction with Ago2. Then, CSD(82–101) of CAV1 was then

fused with mRuby fluorescent protein to determine whether the CSD domain of CAV1 was sufficient for interacting with Ago2. His-CAV1(82–101) mRuby proteins, but not His-mRuby, were coprecipitated with HA-Ago2 proteins (Figs. 1D and EV1E). It has been shown that the $F^{92}TVT^{95}$ segment plays an essential role in CSD function (Couet et al, 1997a; Hoop et al, 2012; Nystrom et al, 1999; Okamoto et al, 1998). We observed that $F^{92}TVT^{95}$ segment-mutated CAV1 (F92A/V94A) proteins were not coprecipitated with HA-Ago2 (Figs. 1E and EV1F), suggesting that a functional CSD of CAV1 with the essential $F^{92}TVT^{95}$ segment is necessary for Ago2/CAV1 interaction. The relevant findings indicated that CSD(82–101) of CAV1 are necessary and sufficient for mediating Ago2/CAV1 interactions.

### $W^{199}$FGFHQSVRPSLWK$^{212}$, an aromatic amino acid–enriched motif, is a putative caveolin binding motif of Ago2

The protein domain of Ago2, which mediates the interaction of Ago2 with CAV1, was mapped out. In HEK293 cells, fragments of Ago2, namely HA-Ago2(1–580), HA-Ago2(1–445), HA-Ago2(1–175), HA-Ago2(1–226), HA-Ago2(1–200), HA-Ago2(73–859), HA-Ago2(175–859), HA-Ago2(226–859), and 175–226-deleted HA-Ago2($\Delta$175–226) (Fig. EV1G), were expressed with CAV1-Flag proteins. With the exception of HA-Ago2(1–175), HA-Ago2(226–859), and HA-Ago2($\Delta$175–226), CAV1-Flag proteins were coprecipitated with each Ago2 protein fragment (Figs. 1F–H and EV1H–K), suggesting amino acids 175–226 of Ago2 was required for the interaction with CAV1. The fragment of Ago2 amino acids 175–226 was then fused with GFP to examine whether it was sufficient for mediating the interaction with CAV1. As shown in Figs. 1I and EV1L, CAV1-Flag proteins were coprecipitated with eGFP-Ago2(175–226)-His but not with eGFP-His. The results indicated that the domain of Ago2 amino acids 175–226 is necessary and sufficient for mediating interaction with CAV1.

The protein association with the CSD primarily occurs through a caveolin binding motif (CBM), which is enriched in tryptophan and other aromatic amino acids, on the CAV1-binding protein. CBM sequences are identified as $\phi$X$\phi$XXXX$\phi$, $\phi$XXXX$\phi$XX$\phi$, or $\phi$X$\phi$XXXX$\phi$XX$\phi$, where $\phi$ is an aromatic residue (Phe, Tyr, or Trp) (Couet et al, 1997a). In this study, we used Pepinfo, which displays different amino acid properties in a sequence, to search aromatic amino acid–enriched motifs in Ago2 (Fig. EV1M, black bars). We observed an aromatic amino acid–enriched motif in the putative CAV1-interacting domain of Ago2, namely W$^{199}$FGFHQSVRPSLWK$^{212}$ (Fig. EV1N, red amino acid). The aromatic amino acid–enriched sequence W$^{199}$FGFHQSVRPSLWK$^{212}$ was conserved in all Ago2 isoforms, including in isoforms 1, 2, X1, X2, and X3 and was conserved across distant species (Fig. 1J), suggesting that this motif is a major determinant of Ago2 function. To verify whether the aromatic amino acid–enriched motif is a CBM, we examined the interaction between CAV1 and Ago2 with four aromatic residues of W$^{199}$FGFHQSVRPSLWK$^{212}$ mutated simultaneously or individually (WF199/200/202/211 A in Fig. 1K and WF199AA, F202A, and W211A in Fig. EV1O). CAV1 proteins were coprecipitated with Ago2 carrying only one or two of the four aromatic amino acids replaced by alanine (WF199AA, F202A, and W211A in Fig. EV1O). By contrast, CAV1 proteins were not coprecipitated with Ago2 in which all four aromatic

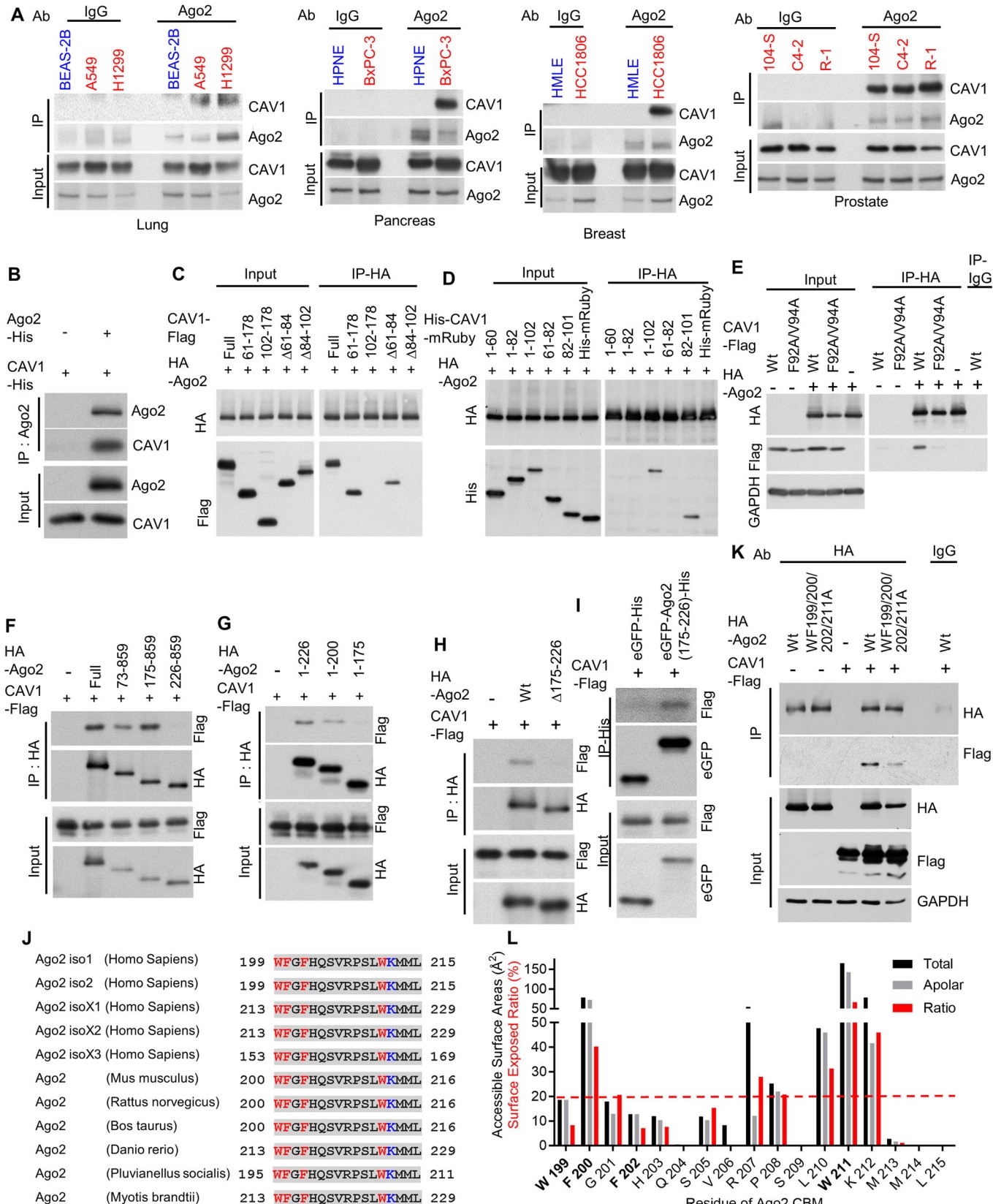

**Figure 1. Ago2 directly binds to the CSD of CAV1 through W$^{199}$FGFHQSVRPSLWK$^{212}$, the CBM of Ago2.**

(A) Ago2/CAV1 interaction analyzed using co-immunoprecipitation. In normal epithelial cells (blue) and cancer cells (red) of different tissues, Ago2 was immunoprecipitated with anti-Ago2 antibodies, and coprecipitation of CAV1 was analyzed. Normal IgG was used as a negative control. (B) Direct interaction of Ago2 with CAV1 in vitro. In the mixture of Ago2-His and CAV1-His, Ago2-His was immunoprecipitated with anti-Ago2 antibodies, and coprecipitation of CAV1 was analyzed. (C) In HEK293 cells, HA-Ago2 was immunoprecipitated with anti-HA antibodies, and coprecipitation of CAV1-Flag and a series of truncated CAV1-Flag proteins was analyzed using anti-Flag antibodies. (D) In HEK293 cells, HA-Ago2 was immunoprecipitated with anti-HA antibodies, and coprecipitation of CAV1 fragment-fused His-mRuby and His-mRuby was analyzed using anti-His antibodies. (E) Ago2/CAV1 interaction mediated by the CSD of CAV1. In HEK293 cells, HA-Ago2 was immunoprecipitated with anti-HA antibodies, and coprecipitation of CSD-mutated (F92A/V94A) or wild-type (Wt) CAV1-Flag was analyzed. (F, G) A series of C-terminal-deleted, N-terminal-deleted, and HA-tagged human Ago2 (HA-Ago2) coexpressed with Flag-labeled human CAV1 (CAV1-Flag) in HEK293 cells. HA-Ago2 was immunoprecipitated with anti-HA antibodies, and coprecipitation of CAV1-Flag was analyzed using anti-Flag antibodies. (H) In HEK293 cells, HA-Ago2 or amino acid 175–226-deleted HA-Ago2 (Δ175–226) was immunoprecipitated with anti-HA antibodies, and coprecipitation of CAV1-Flag was analyzed using anti-Flag antibodies. (I) In HEK293 cells, Ago2 amino acid 175–226-fused eGFP-His [eGFP-Ago2(175–226)-His] or eGFP-His was immunoprecipitated with anti-His antibodies, and coprecipitation of CAV1 was analyzed. (J) Ago2 sequence alignment illustrating CBM conservation. Aromatic amino acids are marked in red, lysine is marked in blue, and conserved sequences are marked in gray. (K) Ago2/CAV1 interaction mediated by a conserved CBM in Ago2. In HEK293 cells, wild-type (Wt) or CBM-mutated (WF199/200/202/211 A) HA-Ago2 was immunoprecipitated with anti-HA antibodies, and coprecipitation of CAV1-Flag was analyzed. (L) Accessible surface area (black and gray bars) and surface exposed ratio SER (red bar) of each residue of Ago2 were estimated using GetArea. The gray bar indicates the accessible surface area of the carbon, halogen, and nonpolar hydrogen atoms of the residue. The quantitation for Western blots in this figure is present in Fig. EV1. Source data are available online for this figure.

residues were mutated (WF199/200/202/211A in Figs. 1K and EV1P). These findings are similar to those for other CAV1-interacting proteins, in which the mutation of a single aromatic residue in the CBM does not abrogate their interaction with CAV1, whereas the simultaneous mutation of all aromatic residues in the CBM disrupts the interaction with CAV1 (Alioua et al, 2008; Brainard et al, 2009; Chun et al, 2005a; Chun et al, 2005b; Jodoin et al, 2003; Nystrom et al, 1999).

Aromatic amino acids are largely hydrophobic and are commonly found buried in the core of proteins to maintain their structure. Surface exposure of aromatic residues to enable direct interaction with other proteins is uncommon (Madhusudan Makwana and Mahalakshmi, 2015). Instead of directly binding CAV1, multiple aromatic amino acids in the CBM may cooperate to maintain the tertiary structure required for CAV1 interaction (Byrne et al, 2012). To validate this notion for Ago2, we calculated the accessible surface area and surface exposed ratio (SER) of each residue of W$^{199}$FGFHQSVRPSLWK$^{212}$ of Ago2 by using GetArea. In GetArea, residues are classified as surface exposed if the SER exceeds 50%, whereas they are categorized as buried if the SER is below 20% (Fraczkiewicz and Braun, 1998). As shown in Fig. 1L, two of the four aromatic residues, namely W199 and F202, were buried in the protein core, whereas only one aromatic residue, namely W211, was surface exposed. Therefore, we used the PoPMuSiC algorithm to predict the ΔΔG thermodynamic stability changes of Ago2 caused by the mutation of aromatic residues (Dehouck et al, 2011). Substitution of buried and partially buried aromatic residues in W$^{199}$FGFHQSVRPSLWK$^{212}$ of Ago2, namely W199, F200, and F202, with alanines had a strong destabilizing effect on the protein structure (ΔΔG > 1.0 kcal/mol; Fig. EV1N) (Tokuriki et al, 2007). These results indicated that the simultaneous substitution of all four aromatic amino acids with alanines may have disrupted the tertiary structure of W$^{199}$FGFHQSVRPSLWK$^{212}$ in Ago2. Therefore, we confirmed that Ago2 binds to the CSD of CAV1 through an aromatic amino acid–enriched CBM, namely W$^{199}$FGFHQSVRPSLWK$^{212}$.

## Ago2 are associated with the plasma membrane of cancer cells through Ago2/CAV1 interaction

To block Ago2/CAV1 interaction in cells, we synthesized cell-permeable peptides according to the sequences of the three parts of

Ago2 amino acids 175–226 (P1, P2, and P3; Fig. EV2A). We then examined their efficiency in blocking Ago2/CAV1 interaction. Because the P3 peptide caused cell death, it was not used for blocking Ago2/CAV1 interaction. Few CAV1 proteins were coprecipitated with Ago2 in both A549 and HCC1806 cancer cells treated with P2 peptides, suggesting that Ago2/CAV1 interaction in cancer cells was interrupted by the P2 peptide (Figs. EV2B,C). By contrast, this interaction was not blocked by the P2S peptide, which contains a scrambled P2 sequence (Fig. EV2C). In contrast to P2S peptides, P2 peptides decreased Ago2 proteins in the membrane fraction (MF) of cancer cells (Fig. 2Ai,Aii). Disruption of Ago2/CAV1 interaction by P2 peptides resulted in the disassociation of Ago2 and other RISC proteins, GW182 and Dicer, from the membranes of cancer cells. However, in addition to Ago2, P2 peptides may also interfere with other CAV1 CSD-interacting proteins. To avoid this scenario, HA-Ago2 (Fig. EV2D, HA-Ago2Wt) and CBM-deleted HA-Ago2 (Fig. EV2D, HA-Ago2Δ) were expressed in Ago2-knockout A549 cancer cells, in which the Ago2 gene was deleted using a CRISPR/Cas9 system (Fig. EV2D, A549$^{Ago2-KO}$). In the MF of A549$^{Ago2-KO/HA-Ago2Wt}$ cells, we observed abundant HA-Ago2Wt. However, CBM-deleted HA-Ago2Δ decreased in the MF of A549$^{Ago2-KO/HA-Ago2Δ}$ cells (Fig. 2Bi,Bii). These results suggest that Ago2/CAV1 interaction is required for membrane association of Ago2 in cancer cells.

Immunofluorescence analysis showed that Ago2 was distributed throughout the nuclei and cytoplasm in normal epithelial cells (i.e., BEAS-2B human lung epithelial cells; Fig. 2C) and was mainly distributed in cytoplasm and membrane of cancer cells (i.e., A549 lung cancer cells; Fig. 2C). Previous studies show that Ago2 is present in the endoplasmic reticulum (ER) membrane (Barman and Bhattacharyya, 2015; Stalder et al, 2013), endosome (McKenzie et al, 2016), and plasma membrane (Shankar et al, 2020). Since we observed that CBM of Ago2 is required for Ago2 association with membranes of cancer cells (Fig. 2B), we further compared the association of Ago2 with various membranes in normal epithelial cells and cancer cells and examined whether deletion of CBM of Ago2 interferences with the association with membranes. The distribution of Ago2, CAV1, endosome marker EEA1, and ER marker Calnexin (CANX) in normal lung epithelial cells BEAS-2B and lung cancer cells A549, A549$^{Ago2-KO/HA-Ago2Wt}$, and A549$^{Ago2-KO/HA-Ago2Δ}$ was analyzed (Figs. 2C and EV2E). The levels of

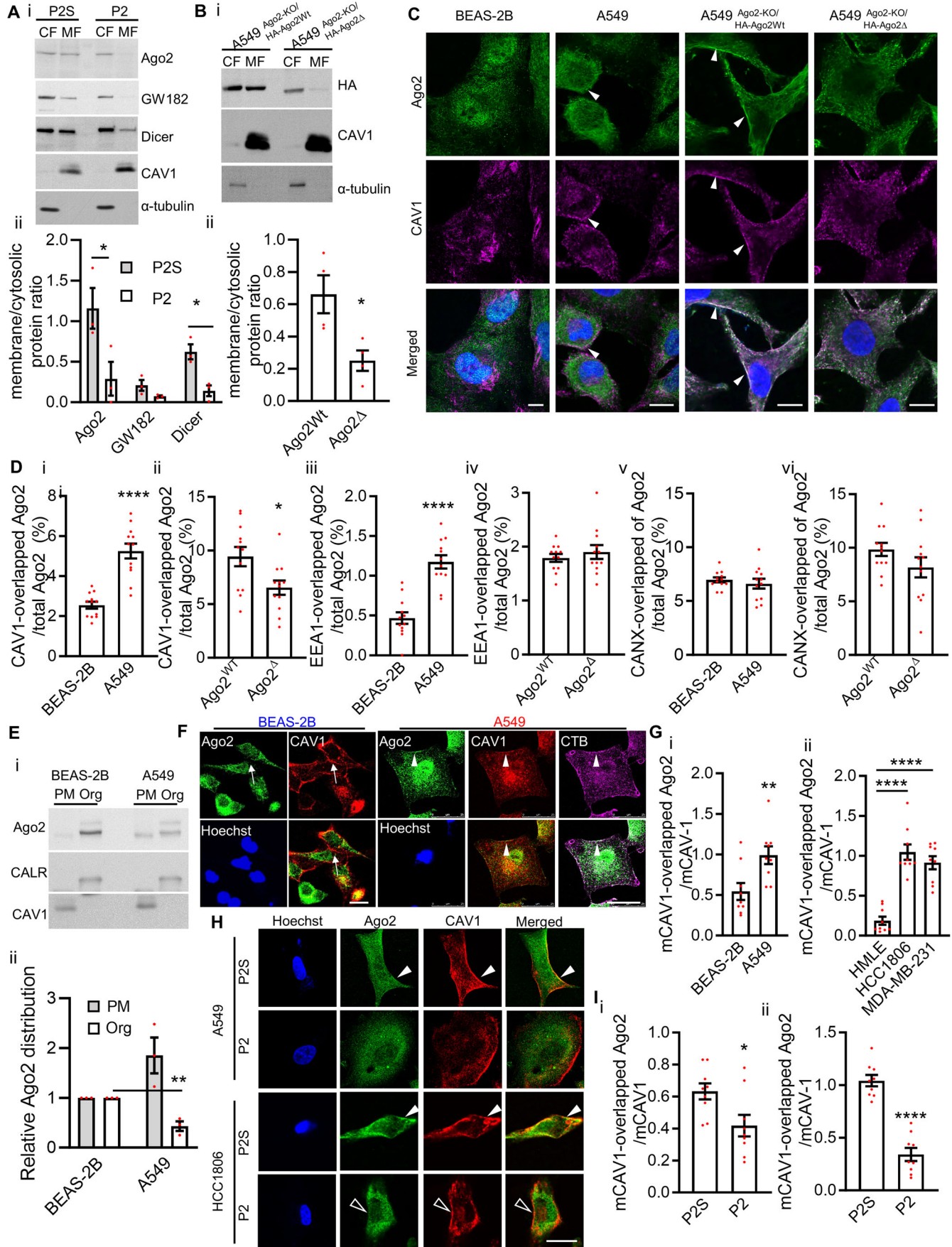

Figure 2. Plasma membrane-associated distribution of Ago2 depends on Ago2/CAV1 interaction.

(A) Membrane association of Ago2 in cancer cells treated with P2. The levels of Ago2, GW182, Dicer, CAV1, and α-tubulin were analyzed in the membrane fractions (MFs) and cytosolic fractions (CFs) of A549 cells treated with P2 or P2S. α-Tubulin: CF marker; CAV1: MF marker. The quantitation is present in panel ii. Each spot describes the ratio of the protein in the membrane fraction to that of the cytosolic fraction. Bars are means ± standard error of the mean (SEM; $n = 3$, biological replicates). Student's t-test, $^*P ≤ 0.05$. (B) Levels of HA-Ago2Wt, HA-Ago2Δ, CAV1, and α-tubulin analyzed in the MFs and CFs of A549$^{Ago2-KO/HA-Ago2Wt}$ and A549$^{Ago2-KO/HA-Ago2Δ}$ cells. The quantitation is present in panel ii. Each spot describes the ratio of the protein in membrane fraction over cytosolic fraction. Bars are means ± SEM ($n = 4$, biological replicates). Student's t-test, $^*P ≤ 0.05$. (C) Distribution of Ago2 (green) and CAV1 (purple) analyzed using immunofluorescence in normal epithelial cells BEAS-2B and cancer cells A549, A549$^{Ago2-KO/HA-Ago2Wt}$ and A549$^{Ago2-KO/HA-Ago2Δ}$. Cell nuclei were stained with Hoechst (blue). The arrowheads indicate the colocalization of Ago2 and CAV1. Scale bar = 10 µm. (D) Quantification of Ago2 localization with CAV1, endosome marker EEA1, and ER marker Calnexin (CANX). Ago2 distribution is evaluated with the percentage of Ago2 overlapping with CAV1, EEA1, or CANX in BEAS-2B, A549, A549$^{Ago2-KO/HA-Ago2Wt}$ and A549$^{Ago2-KO/HA-Ago2Δ}$ cells. Images of representative samples are shown in panel C and Fig. EV2E. Each condition was quantified from four fields per replicate, three biological replicates per experiment. Each spot indicates the percentage of an individual field. Bars are means ± SEM ($n = 12$). Student's t-test, $^*P ≤ 0.05$, $^{****}P ≤ 0.0001$. (E) Levels of Ago2 were analyzed in the plasma membrane (PM) and cellular organelle (Org) fractions of BEAS-2B and A549 cells. CALR: calreticulin as an ER/organelle fraction marker. CAV1: PM marker. The quantitation is present in panel ii. Each spot describes the relative Ago2 protein level of the sample to that of the BEAS-2B sample. Bars are means ± SEM ($n = 3$, biological replicates). Student's t-test, $^{**}P ≤ 0.01$. (F) Distribution of Ago2 (green) and CAV1 (red) was analyzed using immunofluorescence in normal epithelial cells and cancer cells. Lipid rafts were stained with CTB (purple), and cell nuclei were stained with Hoechst (blue). The arrowheads indicate the colocalization of Ago2 and CAV1, and the arrows indicate CAV1 without colocalization with Ago2. Scale bar = 25 µm. (G) Quantification of Ago2 association with CAV1 on the plasma membrane of normal epithelial cells and cancer cells. The association is evaluated with the level of Ago2 overlapping with cell surface CAV1 (mCAV1), normalized with total mCAV1. Images of representative samples are shown in panel F and Fig. EV2F. Each condition was quantified from three fields per replicate, three biological replicates per experiment. Each spot indicates the level of an individual field. Bars are means ± SEM ($n = 9$). Student's t-test, $^{**}P ≤ 0.01$, $^{****}P ≤ 0.0001$. (H) Distribution of Ago2 in cancer cells treated with P2. In cancer cells treated with P2 or P2S peptides, the distribution of Ago2 (green) and CAV1 (red) was analyzed using immunofluorescence. Cell nuclei were stained with Hoechst (blue). The closed arrowheads indicate the colocalization of Ago2 and CAV1, and the open arrowheads indicate CAV1 without colocalization with Ago2. Scale bar = 10 µm. (I) Quantification of Ago2 association with CAV1 on the plasma membrane of cancer cells treated with P2 and P2S peptides. The association is evaluated with the level of Ago2 overlapping with cell surface CAV1 (mCAV1), normalized with total mCAV1. Images of representative samples are shown in panel (H). Each condition was quantified from three fields per replicate, three biological replicates per experiment. Each spot indicates the level of an individual field. Bars are means ± SEM ($n = 9$). Student's t-test, $^*P ≤ 0.05$, $^{****}P ≤ 0.0001$. Source data are available online for this figure.

Ago2 overlapping with CAV1 in A549 cancer cells (Fig. 2C, arrowheads) were higher than that of BEAS-2B normal lung epithelial cells (Fig. 2C,Di). Furthermore, the levels of HA-Ago2Wt overlapping with CAV1 in A549$^{Ago2-KO/HA-Ago2Wt}$ cancer cells (Fig. 2C, arrowheads) were higher than that of CBM-deleted HA-Ago2Δ in A549$^{Ago2-KO/HA-Ago2Δ}$ cells (Fig. 2C,Dii). Colocalization of Ago2 and CAV1 increases in A549 cancer cells and deletion of CBM of Ago2 decreases the colocalization of Ago2 and CAV1 in the cancer cells. The results suggest that CBM of Ago2 is responsible for the increased colocalization of Ago2 and CAV1 in cancer cells.

To analyze the association of Ago2 with endosomes, we observed that the levels of Ago2 overlapping with endosome marker EEA1 in A549 cancer cells were higher than that in BEAS-2B normal lung epithelial cells (Figs. 2Diii and EV2E, left panel). However, the levels of HA-Ago2Wt and CBM-deleted HA-Ago2Δ overlapping with EEA1 in A549$^{Ago2-KO/HA-Ago2Wt}$ and A549$^{Ago2-KO/HA-Ago2Δ}$ cells were not different (Figs. 2Div and EV2E, left panel). The results suggest that the association of Ago2 with endosomes increases in A549 lung cancer cells but CBM of Ago2 is not required for the increased association of Ago2 and endosomes in the cancer cells. It has been shown that S387 phosphorylation of Ago2 controls Ago2 association with endosomes in cancer cells (McKenzie et al, 2016). However, we observed that neither S387A substitution, dephosphorylation mimic, or S387E substitution, phosphorylation mimic, of Ago2, altered Ago2/CAV1 interaction (Fig. EV1H), supporting that Ago2/CAV1 interaction is not responsible for Ago2 association with endosomes in cancer cells. For Ago2 association with ER, we observed that the levels of Ago2 overlapping with ER marker CANX in BEAS-2B normal lung epithelial cells and A549 cancer cells were similar (Figs. 2Dv and EV2E, right panel). In addition, the levels of HA-Ago2Wt and CBM-deleted HA-Ago2Δ overlapping with CANX in A549$^{Ago2-KO/HA-Ago2Wt}$ and A549$^{Ago2-KO/HA-Ago2Δ}$ cells were not

different (Figs. 2Dvi and EV2E, right panel). The results suggest that the association of Ago2 and ER remains the same in both normal lung epithelial cells and lung cancer cells and deletion of CBM of Ago2 does not affect the association of Ago2 with ER in cancer cells.

CAV1 are found predominantly in the plasma membrane but also in vesicles, and at cytosolic locations (Williams and Lisanti, 2004). We further examined whether the distribution of Ago2 between the fractions of plasma membranes, which are enriched with CAV1, and other membrane organelles is associated with Ago2/CAV1 interaction in the cancer cells. Corresponding with the increased CAV1 distribution in the plasma membrane fractions, increased Ago2 proteins were detected in the plasma membrane fractions of cancer cells (Fig. 2Ei, PM). In contrast, the levels of Ago2 decreased in the plasma membrane fractions of BEAS-2B normal epithelial cells (Fig. 2Ei,Eii). The distribution of Ago2 on the plasma membranes of normal epithelial cells and cancer cells was further examined with immunofluorescence analysis. In addition to its cytosolic distribution, Ago2 was also located on the plasma membranes of cancer cells (i.e., HCC1806, MDA-MB-231, and A549 cells; Figs. 2F and EV2F). In normal epithelial cells, Ago2 was not co-located with CAV1 on the plasma membranes (Figs. 2F and EV2F, arrows), whereas in cancer cells, Ago2 proteins were co-located with CAV1 on the plasma membranes (Figs. 2F and EV2F, arrowheads). The locations of certain Ago2 proteins overlapping with those of lipid rafts, labeled with Cholera toxin B subunit (CTB; Figs. 2F and EV2F). The association of Ago2 and CAV1 on the plasma membranes increases in cancer cells (i.e., HCC1806, MDA-MB-231, and A549 cells), compared to that of normal epithelial cells (Fig. 2G).

In cancer cells treated with P2 peptides, Ago2 was disassociated from CAV1 on the plasma membranes, but Ago2 was still co-located with CAV1 on the plasma membrane of P2S-treated cancer

cells (Fig. 2H). The association of Ago2 and CAV1 on plasma membranes was decreased by blocking Ago2/CAV1 interaction with P2 peptides in cancer cells (i.e., HCC1806 and A549 cells; Fig. 2I). Furthermore, gradient ultracentrifugation was used to separate lipid raft fractions from non-lipid raft fractions in the plasma membranes of HEK293 cells (Fig. EV2Gi,Gii). HA-Ago2 proteins were co-purified with the lipid raft component GM1 and CAV1 in the same fractions, but few HA-Ago2Δ proteins existed in the fraction containing lipid rafts. In addition, Ago2 CBM-fused eGFP (eGFP-[175–226]-His) was co-located with CAV1 and lipid rafts, labeled with CTB, on the plasma membranes (Fig. EV2Hi,Hii). These results indicate that Ago2/CAV1 interaction is necessary and sufficient for Ago2 to anchor to the plasma membranes of cancer cells.

## Ago2/CAV1 interaction regulates miRNA-mediated repression

Because Ago2 exhibits nuclease activity and is required for RISC function (O'Brien et al, 2018), its interaction and location may affect miRNA actions. To determine the role of Ago2/CAV1 interaction in Ago2 function, we used miRNA arrays to examine the levels of miRNA in cancer cells in which Ago2/CAV1 interaction was inhibited by P2 or P2S peptides. Disruption of Ago2/CAV1 interaction by P2 peptides resulted in fluctuations in different miRNAs in cancer cells (Fig. 3A; Appendix Fig. S1A). The majority of miRNAs that decreased with the disruption of Ago2/CAV1 interaction (Appendix Fig. S1Ai) were expressed at higher levels in A549 cells than in BEAS-2B cells (Appendix Fig. S1B). Blocking Ago2/CAV1 interaction with P2 peptides suppressed the expression of certain miRNAs, including miRNA-3613-3p, in the cancer cells (Fig. 3A,B). Corresponding to the reduced miR-3613-3p by disruption of Ago2/CAV1 interaction, the expression of miR-3613-3p decreased in normal epithelial cells (i.e., BEAS-2B and HMLE), in which no Ago2/CAV1 interaction was detected (Fig. 3C).

MicroiR-3613-3p plays a role in the pathogenesis or progression of various types of cancer as either an oncogene or a tumor suppressor (Bibi et al, 2016; Boratyn et al, 2016; Castro-Magdonel et al, 2017; Ji et al, 2014; Liu et al, 2020; Pu et al, 2016). This dual role of miR-3613-3p suggests a secondary event that regulates the functions of miR-3613-3p in a context-dependent manner. We therefore set out to investigate whether Ago2/CAV1 interaction, as a secondary event, regulates miRNA-mediated mRNA suppression in cancer cells, using miRNA-3613-3p as an example. Gene targets of miR-3613-3p were searched using miRDB (Appendix Fig. S1C). The effects of Ago2/CAV1 interaction on a target gene involved in regulating cancer cell behavior, namely the suppression of cancer cell invasion (SCAI), were then examined. SCAI mRNA contains 13 potential miR-3613-3p binding sites in the 3′ UTR region (Appendix Fig. S1D). As shown in Fig. 3D, normal lung epithelial BEAS-2B cells expressed higher levels of SCAI mRNAs than did A549 lung cancer cells, corresponding to the levels of SCAI mRNAs in human normal lung tissue and lung tumors (Appendix Fig. S1E). The levels of SCAI mRNAs in normal epithelial BEAS-2B cells and A549 cancer cells are inversely associated with the levels of miR-3613-3p (Fig. 3C, left panel, and Fig. 3D). Blocking Ago2/CAV1 interaction with P2 peptides increased the levels of SCAI mRNAs in A549 cancer cells (Fig. 3E), which are inversely associated with the

levels of miR-3613-3p (Fig. 3B, left panel). In addition, blockage of Ago2/CAV1 interaction increased SCAI proteins, a transcription factor, in the nuclear fraction (NF; Fig. 3F; Appendix Fig. S1F), corresponding to the increase of SCAI mRNAs.

To examine whether blockage of Ago2/CAV1 interaction directly affects miRNA-mediated mRNA repression, we overexpressed miR-3613-3p mimics in A549 cancer cells (Appendix Fig. S1G). As shown in Fig. 3G, these overexpressed miR-3613-3p mimics suppressed the levels of SCAI mRNAs in cancer cells. However, they did not suppress the levels of SCAI mRNAs in P2 peptide–treated A549 cancer cells, although the miR-3613-3p mimics significantly reduced the levels of SCAI mRNAs in P2S peptide–treated A549 cells (Fig. 3H, left panel). These results indicate that Ago2/CAV1 interaction plays an essential role in miR-3613-3p function by suppressing SCAI mRNAs. By contrast, the overexpressed miR-6126 mimics suppressed their target, GRP78 mRNA (Kha, 2018), in both P2- and P2S-treated A549 cancer cells (Fig. 3H, right panel), suggesting that miR-6126-mediated mRNA suppression does not depend on Ago2/CAV1 interaction. Hence, Ago2/CAV1 interaction regulates the expression of certain miRNAs and is required for the miRNA-mediated mRNA repression in cancer cells (e.g., miR-3613-3p).

## Aggressive phenotypes of cancer cells depend on Ago2/CAV1 interaction

Because Ago2/CAV1 interaction is present in cancer cells but not in normal cells, we assume that this interaction is associated with certain cancer cell-specific behaviors. SCAI attenuates epithelial-mesenchymal transition (EMT) (Fintha et al, 2013). In this study, compared with P2S peptide–treated A549 cells, the levels of mesenchymal markers integrin β1, N-cadherin, and fibronectin were lower in the MF of P2 peptide–treated A549 cancer cells (Fig. 3F; Appendix Fig. S1H). In addition, blockage of Ago2/CAV1 interaction in A549 cancer cells with P2 peptides increased the cells' resistance to disassociation through 0.05% trypsin treatment, which is an epithelial phenotype (Fig. 4A). Blockage of Ago2/CAV1 interaction in A549 cancer cells with P2 peptides also made the cells sensitive to anoikis, which is a hallmark of EMT (Fig. 4B).

Since EMT in cancer cells contributes to drug resistance (Singh and Settleman, 2010; Voulgari and Pintzas, 2009), stemness (Mani et al, 2008; Singh and Settleman, 2010), and invasiveness (Brabletz et al, 2005) of cancer cells, we examined the effects of Ago2/CAV1 interaction on these cancer cell properties. The results indicated that blockage of Ago2/CAV1 interaction in A549 cancer cells with P2 peptides decreased cell viability under paclitaxel (Pac) and gefitinib (Gef) treatment, thereby attenuating drug resistance (Fig. 4C). Blocking Ago2/CAV1 interaction in cancer cells with P2 peptides decreased the numbers of tumorsphere-forming cells in the A549 cancer cell populations (Fig. 4Di) and HCC1806 cancer cell populations (Fig. 4Dii). Blocking Ago2/CAV1 interaction in cancer cells with P2 peptides decreased the migration and invasion of A549 cancer cells (Fig. 4Ei) and HCC1806 cancer cells (Fig. 4Eii) in Boyden chambers. These results indicated that Ago2/CAV1 interaction is essential for aggressive phenotypes of cancer cells, including mesenchymal morphology, anoikis resistance, drug resistance, tumorsphere formation, migration, and invasion.

**A**

| Transcript ID | P2 vs P2S Fold Change | P-val |
|---|---|---|
| hsa-miR-6780b-5p | 3.54 | 1.70E-05 |
| hsa-miR-1281 | 3.34 | 0.0364 |
| hsa-miR-6126 | 2.43 | 0.0038 |
| hsa-miR-4485 | 2.42 | 0.0063 |
| hsa-miR-6867-5p | 2.36 | 0.0457 |
| hsa-miR-221-5p | 2.28 | 0.0379 |
| hsa-miR-3136-5p | 2.23 | 0.005 |
| hsa-miR-3195 | 2.22 | 9.24E-05 |
| hsa-miR-3142 | 2.21 | 0.0022 |
| hsa-miR-939-5p | 2.15 | 0.0025 |
| hsa-miR-150-3p | 2.14 | 0.0068 |
| hsa-miR-2278 | 2.07 | 0.026 |
| hsa-miR-501-5p | 2.07 | 0.0212 |
| hsa-miR-1273d | 2.04 | 0.0087 |
| hsa-miR-4453 | 1.88 | 0.0071 |
| hsa-miR-7159-5p | 1.87 | 0.0128 |
| hsa-miR-595 | 1.86 | 0.0325 |
| hsa-miR-4745-5p | 1.85 | 0.0272 |
| hsa-miR-6768-5p | 1.85 | 0.0022 |
| hsa-miR-7515 | 1.84 | 0.0026 |
| hsa-miR-4417 | 1.79 | 0.0139 |
| hsa-miR-346 | 1.79 | 0.0016 |
| hsa-miR-4484 | 1.77 | 0.0013 |
| hsa-miR-4717-3p | 1.76 | 0.0075 |
| hsa-miR-337-5p | 1.76 | 0.0071 |
| hsa-miR-3659 | 1.74 | 0.0321 |
| hsa-miR-8077 | 1.73 | 0.0249 |
| hsa-miR-4478 | 1.72 | 0.0038 |
| hsa-miR-498 | 1.72 | 0.0024 |
| hsa-miR-6754-3p | 1.71 | 0.0153 |
| hsa-miR-4690-5p | 1.69 | 0.038 |
| hsa-miR-302c-5p | 1.66 | 0.0027 |
| hsa-miR-4665-3p | 1.66 | 0.001 |
| hsa-miR-6861-5p | 1.63 | 0.0066 |
| hsa-miR-6074 | 1.62 | 0.0332 |
| hsa-miR-374c-3p | 1.62 | 0.0109 |
| hsa-miR-5090 | 1.61 | 0.049 |
| hsa-miR-582-5p | 1.6 | 0.0196 |
| hsa-miR-1976 | 1.58 | 0.0086 |
| hsa-miR-3117-5p | 1.57 | 0.0045 |
| hsa-miR-3928-5p | 1.56 | 0.0354 |
| hsa-miR-3189-3p | 1.56 | 0.0118 |
| hsa-miR-362-3p | 1.55 | 0.0079 |
| hsa-miR-1278 | 1.55 | 0.0207 |
| hsa-miR-130a-5p | 1.54 | 0.0234 |
| hsa-miR-6852-5p | 1.54 | 0.0361 |
| hsa-miR-302c-3p | 1.5 | 0.0349 |
| hsa-miR-6890-5p | -1.5 | 0.0191 |
| hsa-miR-374a-3p | -1.51 | 0.0253 |
| hsa-miR-146a-5p | -1.55 | 0.0055 |
| hsa-miR-6786-3p | -1.55 | 0.0284 |
| hsa-miR-4493 | -1.62 | 0.0334 |
| hsa-miR-146b-5p | -1.67 | 0.0397 |
| hsa-miR-4655-5p | -1.68 | 0.0075 |
| hsa-miR-615-5p | -1.73 | 0.0019 |
| hsa-miR-4294 | -1.74 | 0.0186 |
| hsa-miR-503-3p | -1.76 | 0.0471 |
| hsa-miR-122-5p | -1.77 | 0.032 |
| hsa-miR-4662a-3p | -1.86 | 0.0052 |
| hsa-miR-186-3p | -1.89 | 0.0047 |
| hsa-miR-6845-5p | -1.9 | 0.0492 |
| hsa-miR-4668-5p | -2.11 | 0.0102 |
| hsa-miR-1183 | -2.95 | 0.0094 |
| hsa-miR-3613-3p | -3.73 | 0.019 |

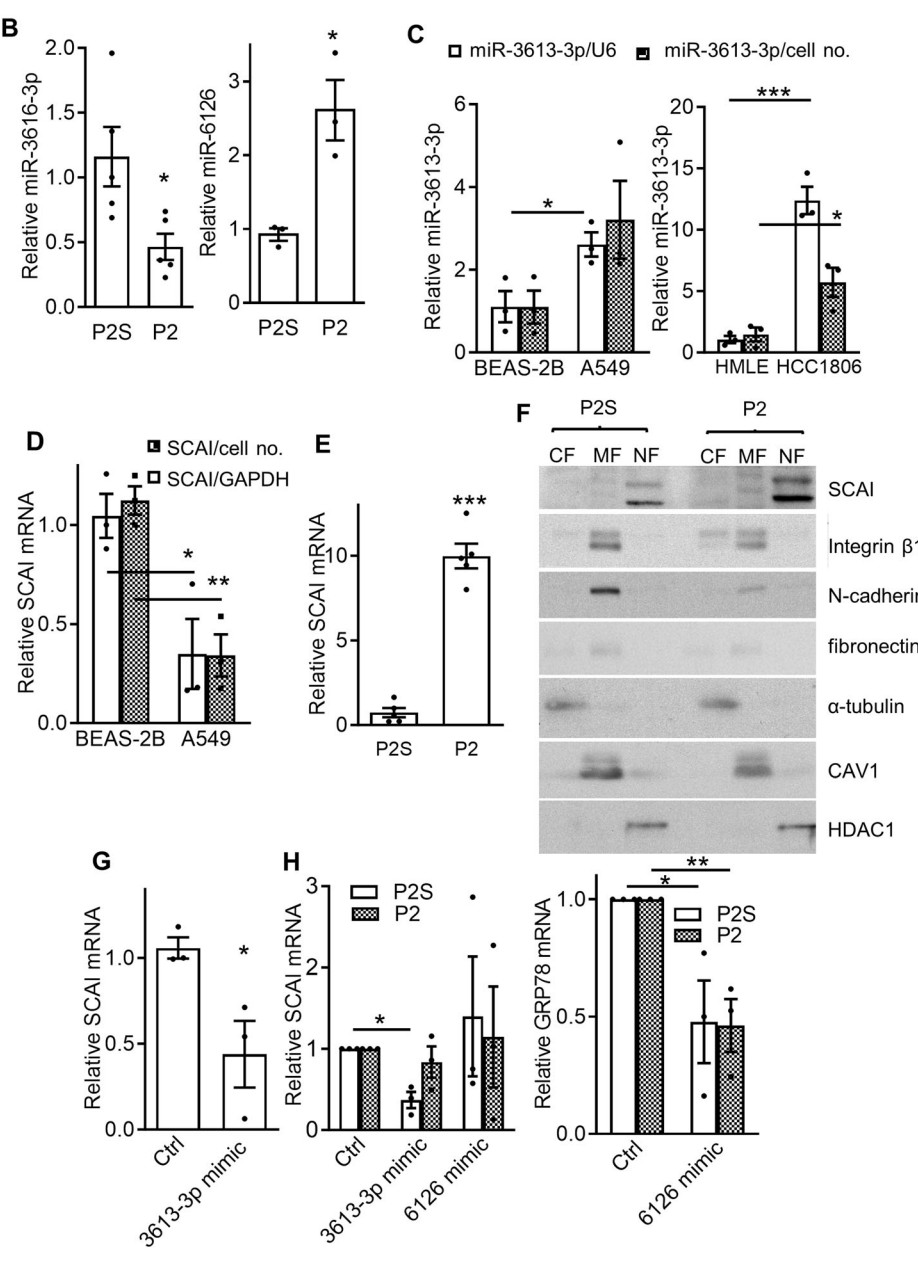

**Figure 3.  Ago2/CAV1 interaction plays a crucial role in the expression and function of miR-3613-3p.**

(A) The miRNAs were differentially expressed in P2- and P2S-treated A549 cancer cells. The green indicates miRNAs with low expression levels in P2-treated cancer cells, with an average P2S/P2 level of >1.5 ($P < 0.05$). The red indicates miRNAs with high expression levels in P2-treated cancer cells, with an average P2/P2S level of >1.5 ($P < 0.05$). (B) Expression of miR-3613-3p (left panel) and miR-6126 (right panel) in P2- and P2S-treated A549 cancer cells. Each spot describes the relative miRNA level of the sample to that of the P2S-treated sample. Bars are means ± SEM ($n = 5$ and 3, biological replicates). Student's $t$-test, $^*P \le 0.05$. (C) Expression of miR-3613-3p in normal epithelial cells and cancer cells. Each spot describes the relative miRNA level of the sample to that of the normal epithelial cells. Bars are means ± SEM ($n = 3$, biological replicates). Open bar: miRNA level normalized to U6 level. Stripped bar: miRNA level normalized to cell number. Student's $t$-test, $^*P \le 0.05$; $^{***}P \le 0.001$. (D) Expression of SCAI mRNAs in BEAS-2B and A549 cells. Each spot describes the relative mRNA level of the sample to that of the BEAS-2B cells. Bars are means ± SEM ($n = 3$, biological replicates). Open bar: mRNA level normalized to GAPDH mRNA level. Stripped bar: mRNA level normalized to cell number. Student's $t$-test, $^*P \le 0.05$; $^{**}P \le 0.01$. (E) Expression of SCAI mRNAs in P2- and P2S-treated A549 cancer cells. Each spot describes the relative mRNA level of the sample to that of the P2S-treated sample. Bars are means ± SEM ($n = 5$, biological replicates). Student's $t$-test, $^{***}P \le 0.001$. (F) Expression of proteins in cancer cells treated with P2 peptides. The levels of proteins were analyzed in the membrane fractions (MFs), cytosolic fractions (CFs), and nuclear fractions (NFs) of A549 cells treated with P2 or P2S peptides. α-Tubulin: CF marker; CAV1: MF marker; HDAC1: NF marker. The quantitation for this experiment is present in Appendix Fig. S1F and S1H. (G) Expression of SCAI mRNAs in A549 cells with miR-3613-3p mimics. Each spot describes the relative mRNA level of the sample to that of the A549 cells (Ctrl). Bars are means ± SEM ($n = 3$, biological replicates). Student's $t$-test, $^*P \le 0.05$. (H) Expression of SCAI (left panel) and GRP78 (right panel) mRNAs in P2- and P2S-treated A549 cells with miR-3613-3p and miR-6126 mimics. Each spot describes the relative mRNA level of the sample to that of the A549 cells without miRNA mimics (Ctrl). Bars are means ± SEM ($n = 3$, biological replicates). Student's $t$-test, $^*P \le 0.05$; $^{**}P \le 0.01$. Source data are available online for this figure.

## Ago2/CAV1 interaction is required for tumor formation and metastasis

Peptides for in vivo applications are limited because of stability and tissue penetrability challenges. In addition, P2 peptides may also interfere with CAV1 CSD-interacting proteins other than Ago2. To avoid these scenarios, we used auxin-inducible degradation (AID) to enable efficient and specific blockage of Ago2/CAV1 interaction in A549$^{Ago2-KO/HA-AID-Ago2Wt/HA-Ago2\Delta}$ cells and HCC1806$^{Ago2-KO/HA-AID-Ago2Wt/HA-Ago2\Delta}$ cells (Fig. EV3A) both in vivo and in vitro. Indeed, controlling protein expression with AID rapidly depleted the protein of interest (i.e., Ago2) in a reversible manner (Fig. EV3Aii; see "Materials and Methods"). In A549$^{Ago2-KO/HA-AID-Ago2Wt/HA-Ago2\Delta}$ cells treated with indole-3-acetic acid (IAA), only the expression of HA-AID-Ago2Wt protein was suppressed, whereas the expression of HA-Ago2Δ was maintained (Fig. EV3Bi,Bii). In A549$^{Ago2-KO/HA-AID-Ago2Wt/HA-Ago2\Delta}$ cells treated with IAA, which only expressed HA-Ago2Δ, CAV1 was not precipitated with HA-Ago2Δ by anti-HA antibodies, indicating that Ago2/CAV1 interaction was blocked in IAA-treated cancer cells with an auxin-inducible degron system (Fig. EV3Ci,Cii). Similar to blocking Ago2/CAV1 interaction with P2 peptides, blocking Ago2/CAV1 interaction with IAA in A549$^{Ago2-KO/HA-AID-Ago2Wt/HA-Ago2\Delta}$ cells decreased miR-3613-3p, increased SCAI mRNAs (Fig. EV3Di,Dii), selectively impaired the suppression of miR-3613-3p mimics on SCAI mRNAs (Fig. EV3E), increase resistance to trypsin disassociation (Fig. EV3F), decrease anoikis resistance (Fig. EV3G), attenuate tumorsphere formation (Fig. EV3H), and suppress invasion (Fig. EV3I) of cancer cells. Blockage of Ago2/CAV1 interaction by the AID system has the same effects as P2 peptides on miRNA regulation and cancer cell behavior, suggesting that the altered miRNA regulation (Fig. 3) and cancer cell behavior (Fig. 4) by P2 peptides indeed result from the disruption of Ago2/CAV1 interaction.

As Ago2/CAV1 interaction promotes mesenchymal and invasive phenotypes of cancer cells in culture, we investigated the role of Ago2/CAV1 interaction in tumor metastasis. We examined whether Ago2/CAV1 interaction is required for cancer cell dissemination from primary tumors. As shown in Fig. 5Ai, A549$^{Ago2-KO/AID-Ago2Wt/Ago2\Delta}$ cancer cells were subcutaneously implanted into SCID mice, and the mice were injected with

phosphate-buffered saline (PBS) or IAA for disruption of Ago2/CAV1 interaction. According to the results of PCR amplification (Fig. 5B, lower panel) and qPCR of human GAPDH gDNA (Fig. 5C), disseminated tumor cells were detected in the lungs of four out of six mice injected with PBS but not in the lungs of mice injected with IAA. Blocking Ago2/CAV1 interaction in cancer cells decreases the dissemination of primary tumor cells in the lungs.

Given that Ago2/CAV1 interaction is required for the dissemination of primary tumor cells in the lungs, we investigated the role of Ago2/CAV1 interaction in lung targeting of circulating cancer cells, the early stage of metastasis. As shown in Fig. 5Aii, at 96 h after tail vein injection with A549$^{Ago2-KO/AID-Ago2Wt/Ago2\Delta}$ cells, PBS- and IAA-treated mice were perfused with PBS to remove non-extravasated cancer cells from the vessels. Cancer cells invading the lung mesenchyme were then measured using qPCR of human GAPDH gDNA. At 96 h after cell injection, the number of IAA-treated A549$^{Ago2-KO/AID-Ago2Wt/Ago2\Delta}$ cells remaining in the perfused lungs was 99.5% less than that of PBS-treated A549$^{Ago2-KO/AID-Ago2Wt/Ago2\Delta}$ cells (Fig. 5D). Blocking Ago2/CAV1 interaction in cancer cells decreased lung targeting of circulating cancer cells.

We next investigated whether blockage of Ago2/CAV1 interaction affects the metastatic tumor formation of circulating cancer cells in organs. As shown in Fig. 5Aiii, luciferase-labeled A549$^{Ago2-KO/AID-Ago2Wt/Ago2\Delta}$ cells were injected into the tail veins of mice, which were then continuously treated with PBS or IAA for 4 weeks. Subsequently, tumor formation in the lungs was quantitated using photon counts measured by IVIS imaging (Fig. 5Ei,Eii), number of tumor nodules (Fig. 5Eiii,Eiv), and number of cancer cells measured by qPCR of human GAPDH gDNA (Fig. 5Ev). Multiple tumors were easily detected in the lungs of all six mice injected with PBS, whereas only a few tumors were detected in mice injected with IAA (Fig. 5E). In addition, A549$^{Ago2-KO/AID-Ago2Wt/Ago2\Delta}$ cancer cells were frequently detected in other organs (e.g., bones, the brain, adrenal glands, and the liver) in PBS-injected mice (Appendix Fig. S2A). Blocking Ago2/CAV1 interaction attenuates the metastatic tumor formation of circulating cancer cells in the lungs and other organs.

We then used an orthotopic breast tumor model to investigate the role of Ago2/CAV1 interaction in tumor formation and metastasis. As shown in Appendix Fig. S2B, luciferase-labeled HCC1806$^{Ago2-KO/AID-Ago2Wt/Ago2\Delta}$ breast cancer cells were injected into the fat pads of SCID mice, which were then continuously treated

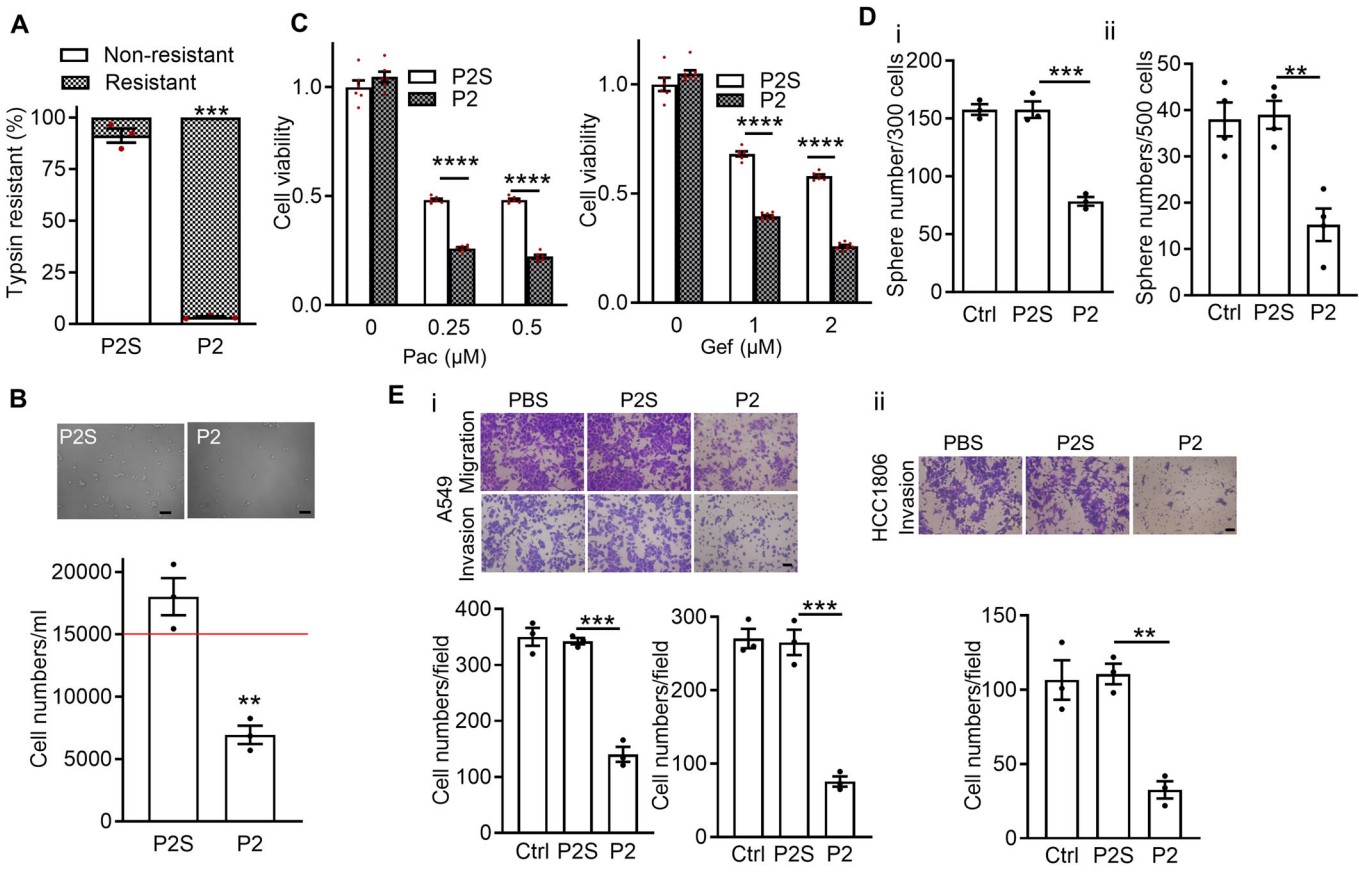

**Figure 4. Disruption of Ago2/CAV1 interaction suppresses the aggressive behaviors of cancer cells.**

(A) Blockage of Ago2/CAV1 interaction increased the resistance of cancer cells to trypsinization. A549 cells were treated with P2 or P2S peptides (10 μM daily) for 2 days. Treated cells were evaluated in terms of resistance to 0.05% trypsin. Bars are means ± SEM ($n = 3$, biological replicates). Student's $t$-test, ***$P \leq 0.001$. (B) Blockage of Ago2/CAV1 interaction decreased the resistance of cancer cells to anoikis. The numbers of P2/P2S-treated A549 cells in suspension were evaluated in terms of resistance to anoikis. The red line indicates the initial cell number on day 0. Bars are means ± SEM ($n = 3$, biological replicates). Student's $t$-test, **$P \leq 0.01$. Scale bar = 50 μm. (C) Blockage of Ago2/CAV1 interaction increased the sensitivity of cancer cells to chemotherapeutic agents. P2- and P2S-treated cells were exposed to increasing doses of Paclitaxel (Pac) or Gefitinib (Gef), and their viability was measured using an MTT assay. Each condition was quantified from two technical replicates per biological replicates, and three biological replicates per experiment Bars are means ± SEM ($n = 6$). Student's $t$-test, ****$P \leq 0.0001$. (D) Blockage of Ago2/CAV1 interaction decreased tumorsphere formation. A549 (panel i) and HCC1806 (panel ii) cells were treated with P2 or P2S peptides. The cells were subjected to tumorsphere assays, and treatment was suspended during the assays. Bars are means ± SEM ($n = 3$ and 4, biological replicates). Student's $t$-test, **$P \leq 0.01$; ***$P \leq 0.001$. (E) Blockage of Ago2/CAV1 interaction decreased the invasion and migration of cancer cells. A549 (panel i) and HCC1806 (panel ii) cells were treated with P2 or P2S peptides. The cells were subjected to invasion and migration assays, and treatment was suspended during the assays. Bars are means ± SEM ($n = 3$, biological replicates). Student's $t$-test, **$P \leq 0.01$, ***$P \leq 0.001$. Scale bar = 100 μm. Source data are available online for this figure.

with PBS or IAA for 2 weeks. Subsequently, tumors in the fat pads were noninvasively monitored using IVIS imaging and extracted at 2 weeks after cell injection (Fig. 5Fi). The weights of the tumors that formed in IAA-treated mice were 48% lower compared with those that formed in PBS-treated mice (Fig. 5Fii). In addition, blocking Ago2/CAV1 interaction in HCC1806[Ago2-KO/AID-Ago2Wt/Ago2Δ] cells decreased primary tumor cell dissemination to lymph nodes by 86% (Fig. 5Fiii) and to the lungs by 84% (Fig. 5Fiv). These results indicate that blockage of Ago2/CAV1 interaction attenuates primary tumor formation and tumor metastasis.

## Positive charge of Ago2 lysine 212 in the CBM is necessary for Ago2/CAV1 interaction

Although the hydrophobic interaction of nonpolar residues is a driving force behind the folding stability of proteins, as a function

of aromatic residues in the CBM, the interaction between charged residues may confer and reinforce protein binding specificity with partners (Zhou and Pang, 2018). To understand the underlying mechanism of different Ago2/CAV1 interactions in cancer cells and normal epithelial cells, we investigated the regulation of Ago2/CAV1 interaction through the charged residues of Ago2. We replaced charged residues in the putative binding fragment (amino acids 175–226) of Ago2 with uncharged amino acids (Fig. EV2A, blue box) to generate mutants of Ago2, namely HA-Ago2(E186F), HA-Ago2(E197F), HA-Ago2(K212A), and HA-Ago2(D218F). With the exception of HA-Ago2(K212A), CAV1-Flag proteins were coprecipitated with these Ago2 mutants (Fig. 6A; Appendix Fig. S3A).

Ago2 lysine 212 is highly conserved across different isoforms and species (Fig. 1J). However, the substitution of Ago2 lysine 212 with alanine is regarded as a neutral mutation ($-1.0 < \Delta\Delta G < 1.0$

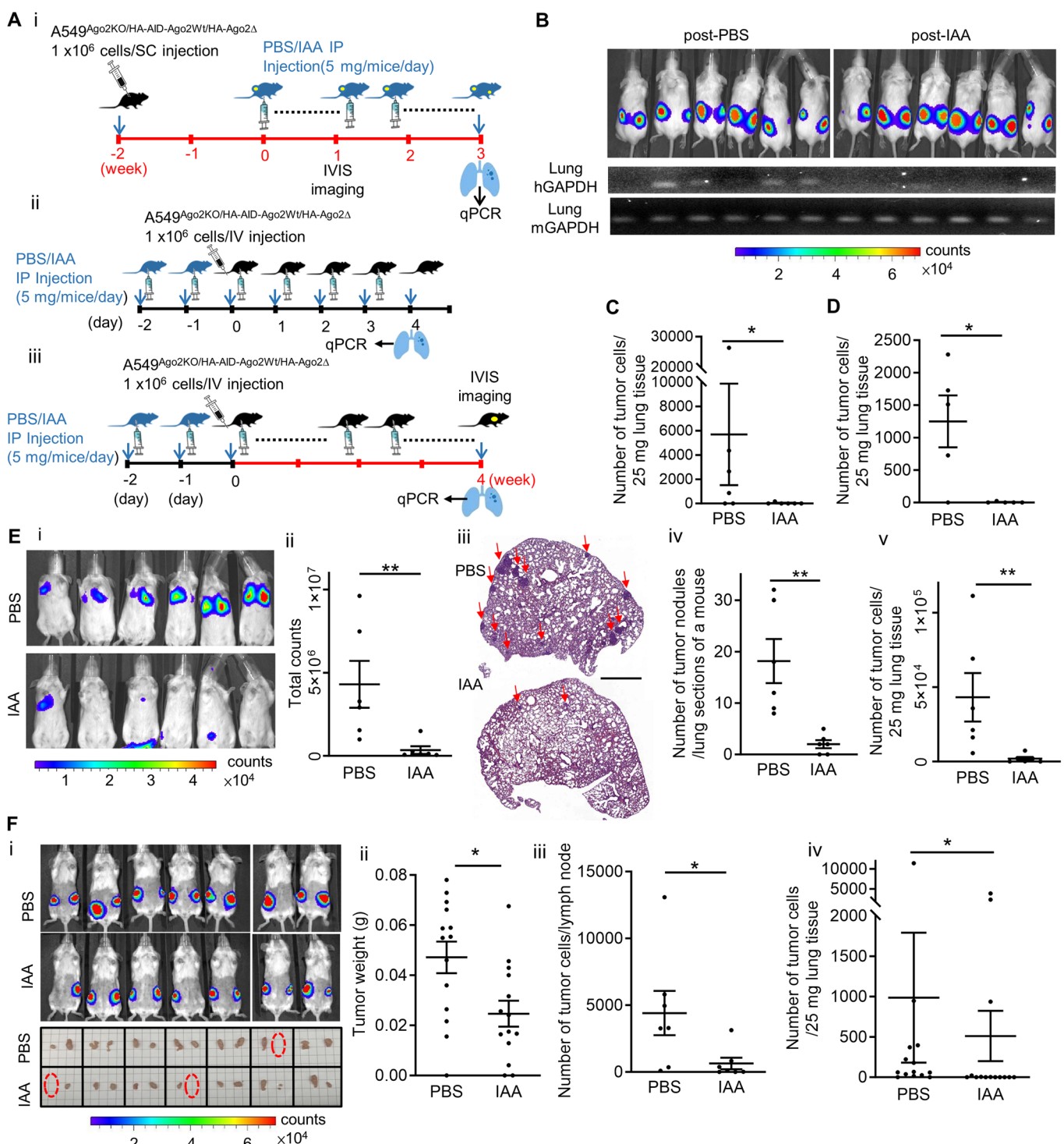

kcal/mol), which does not considerably affect the protein folding of Ago2 (Fig. EV1N). Therefore, we assume that lysine 212 directly contributes to Ago2/CAV1 interaction through its charged side chain. Lysine (K) and arginine (R) are structurally similar and have a positively charged side chain. Substitution of lysine with arginine (R) preserves this positive charge, whereas substitution with alanine (A) neutralizes this charge. In this study, the substitution of Ago2

lysine 212 with either alanine ($\Delta\Delta G = 0.75$) or arginine ($\Delta\Delta G = -0.05$) did not substantially affect the tertiary structure of Ago2 (Fig. EV1N). While the substitution of Ago2 lysine 212 with alanine decreased the interaction with CAV1 proteins (Fig. 6B, HA-Ago2K212A; Appendix Fig. S3B), CAV1 proteins remained coprecipitated with HA-Ago2K212R (Fig. 6B; Appendix Fig. S3B) in HEK293, A549, and HCC1806 cancer cells. Hence, Ago2/CAV1

**Figure 5.  Ago2/CAV1 interaction plays an essential role in tumor growth and metastasis.**

(A) Scheme of animal experiments, including lung metastasis of subcutaneous tumor cells (panel i), lung targeting of circulating cancer cells (panel ii), and lung colonization of circulating cancer cells (panel iii), indicating the time points and routes of PBS/IAA injection, time points and routes of A549$^{Ago2KO/HA-AID-Ago2Wt/HA-Ago2\Delta}$ injection, and time points of IVIS imaging and tissue collection. (B) Growth of PBS- and IAA-treated subcutaneous A549$^{Ago2KO/HA-AID-Ago2Wt/HA-Ago2\Delta}$ tumors expressing luciferase (fLuc), monitored using noninvasive imaging. Lung metastasis was detected using PCR, which specifically amplifies human GAPDH gDNA (hGAPDH). Mouse GAPDH gDNA was used as a loading control (mGAPDH). (C) Quantification of lung metastasis of PBS- and IAA-treated A549$^{Ago2KO/AID-Ago2Wt/Ago2\Delta}$ tumors in panel B by qPCR. Each spot describes the number of tumor cells per 25 mg of lung tissue in an individual mouse. Bars are means ± SEM ($n = 6$). Mann–Whitney $U$-test, $^*P \le 0.05$. (D) Role of Ago2/CAV1 interaction in the lung targeting of cancer cells. Quantification of lung targeting by PBS- and IAA-treated A549$^{Ago2KO/HA-AID-Ago2Wt/HA-Ago2\Delta}$ tumors through qPCR at 96 h after intravenous (IV) injection. Each spot describes the number of tumor cells per 25 mg of lung tissue in an individual mouse. Bars are means ± SEM ($n = 5$). Student's $t$-test, $^*P \le 0.05$. Mann–Whitney $U$-test, $P = 0.06$. (E) Role of Ago2/CAV1 interaction in the colonization of cancer cells. Panel i: Tumor colonization in the lungs was monitored using noninvasive imaging. Panel ii: The luminescent counts of lung area in the images of panel i were calculated. Each spot describes the counts of an individual mouse. Bars are means ± SEM ($n = 6$). Mann–Whitney $U$-test, $^{**}P \le 0.01$. Panels iii and iv: The number of tumor nodules, indicated with arrows, in the lung sections of mice injected with PBS- and IAA-treated A549$^{Ago2KO/HA-Ago2Wt/Ago2\Delta}$ cells was quantified. Each spot describes the number of tumor nodules in the lung sections of an individual mouse. Bars are means ± SEM ($n = 6$). Mann–Whitney $U$-test, $^{**}P \le 0.01$. Scale bar = 0.2 cm. Panel v- Quantification of lung colonization of PBS- and IAA-treated A549$^{Ago2KO/HA-AID-Ago2Wt/HA-Ago2\Delta}$ cells through qPCR. Each spot describes the number of tumor cells per 25 mg of lung tissue in an individual mouse. Bars are means ± SEM ($n = 6$). Mann–Whitney $U$-test, $^{**}P \le 0.01$. (F) Role of Ago2/CAV1 interaction in the growth and metastasis of breast tumors. Panels (i) and (ii): Breast tumors formed by PBS- or IAA-treated HCC1806$^{Ago2KO/HA-AID-Ago2Wt/HA-Ago2\Delta}$ cells in fat pads, as indicated in Appendix Fig. S2B, monitored using noninvasive imaging (panel i, top). Breast tumors isolated from mice were weighed (panel i [bottom] and panel ii). Each spot describes tumor weight at each injection site. Bars are means ± SEM ($n = 14$). Mann–Whitney $U$-test, $^*P \le 0.05$. Panels (iii) and (iv): qPCR quantification of lymph node metastasis (panel iii) and lung metastasis (panel iv) of tumors formed by PBS- and IAA-treated HCC1806$^{Ago2KO/HA-AID-Ago2Wt/HA-Ago2}$ cells. Each spot describes the number of tumor cells in an individual mouse. Bars are means ± SEM [$n = 7$ for panel (iii) and $n = 14$ for panel (iv); see "Materials and Methods"]. Mann–Whitney $U$-test, $^*P \le 0.05$. Source data are available online for this figure.

interaction was disrupted by the substitution of Ago2 lysine 212 with alanine and was maintained by the substitution of Ago2 lysine 212 with arginine. These findings highlight the potential role of the positive charges of Ago2 lysine 212 in Ago2/CAV1 interaction.

## SIRT2 is required for maintaining the positive charge of Ago2 lysine 212 in the CBM for Ago2/CAV1 interaction in cancer cells

To understand the underlying mechanism of Ago2/CAV1 interaction differently mediated by Ago2 lysine 212 in cancer cells and normal epithelial cells, we investigated the regulation of Ago2/CAV1 interaction through modification on Ago2 lysine 212, which interferes with the positive charge of lysine. Acetylation on lysine (K) neutralizes this positive charge and often has substantial consequences (Ali et al, 2018). Compared with HA-Ago2Wt, HA-Ago2K212A in A549$^{Ago2-KO/HA-Ago2K212A}$ cancer cells (Fig. 6Ci; Appendix Fig. S3Ci) and HCC1806$^{Ago2-KO/HA-Ago2K212A}$ cancer cells (Fig. 6Cii; Appendix Fig. S3Cii) had lower levels of total acetylation. Replacing Ago2 lysine 212 decreases the acetylation of Ago2, suggesting the acetylation of Ago2 K212.

Since the positive charge of Ago2 K212 is essential for Ago2/CAV1 interaction and acetylation of Ago2 K212 neutralizes the positive charge, we further explored the deacetylases that may attenuate Ago2 K212 acetylation in cancer cells for Ago2/CAV1 interaction. Among the cytosolic deacetylases responsible for the deacetylation of nonhistone proteins (i.e., HDAC6, SIRT1, and SIRT2), the expression of SIRT1/2 was considerably higher in cancer cells (i.e., A549 and HCC1806 cells) than in normal tissue cells (i.e., BEAS-2B and HMLE cells; Fig. 6D; Appendix Fig. S3D). We investigated the effects of these cytosolic deacetylases on Ago2/CAV1 interaction in the presence of inhibitors, such as HDAC6 inhibitor Nexturastat A (Fig. 6Ei; Appendix Fig. S3Ei), SIRT1/2 inhibitor Sirtinol (Fig. 6Ei; Appendix Fig. S3Ei), SIRT1 inhibitor EX-527 (Fig. 6Eii; Appendix Fig. S3Eii), and SIRT2 inhibitor Thiomyristoyl (TM; Fig. 6Eii,Eiii; Appendix Fig. S3Eii,Eiii). Among these inhibitors, TM attenuated Ago2/CAV1 interaction in both A549 and HCC1806 cancer cells (Fig. 6E; Appendix Fig. S3E).

These results indicated that SIRT2 deacetylases decrease Ago2 K212 acetylation and increase Ago2/CAV1 interaction in cancer cells.

To investigate whether the SIRT2 inhibitor TM decreases Ago2/CAV1 interaction by reducing the acetylation of Ago2 K212, we investigated the interaction of CAV1 with HA-Ago2Wt or HA-Ago2K212R under the effect of TM (Fig. 6Fi). Ago2/CAV1 interaction was measured as the ratio of coprecipitated CAV1 to precipitated HA-Ago2 by anti-HA antibodies, pCAV1/pHA, in A549$^{Ago2-KO/HA-Ago2Wt}$, A549$^{Ago2-KO/HA-Ago2K212A}$, and A549$^{Ago2-KO/HA-Ago2K212R}$ cells (Fig. 6Fii). In Ago2, K212R substitution, which mimics K212 deacetylation, resulted in a 37% increase in Ago2/CAV1 interaction (K212R; Fig. 6Fii), whereas K212A substitution, which mimics K212 acetylation, resulted in a 46% decrease in Ago2/CAV1 interaction (K212A; Fig. 6Fii). TM treatment decreased HA-Ago2Wt/CAV1 interaction to approximately 49%, similar to that of HA-Ago2K212A/CAV1 interaction (Wt vs. Wt+TM and K212A; Fig. 6Fii), indicating that TM had the same effect as that of Ago2K212A substitution on Ago2/CAV1 interaction. By contrast, TM did not decrease HA-Ago2K212R/CAV1 interaction, indicating that K212R substitution prevented TM from disrupting Ago2/CAV1 interaction (K212R vs. K212R + TM; Fig. 6Fii). These results indicated that SIRT2-mediated deacetylation of Ago2 lysine 212 increases Ago2/CAV1 interaction in cancer cells.

## Positive charge of Ago2 lysine 212 is required for tumor formation and dissemination

Similar to blocking Ago2/CAV1 interaction with P2 peptides and with deletion of the CBM domain of Ago2 (Fig. 2C,D,H,I), blocking Ago2/CAV1 interaction with Ago2K212A substitution decreased association of Ago2 with the plasma membranes (Fig. 6G; Appendix Fig. S3F). We analyzed and compared miRNA fluctuation resulting from disruption of Ago2/CAV1 interaction by P2 peptides, deletion of Ago2 CBM, and Ago2K212A substitution with miRNA arrays. There are ten shared upregulated miRNAs and nine shared downregulated miRNAs in P2-treated A549, A549$^{Ago2-KO/HA-Ago2\Delta}$ (Ago2Dm), and A549$^{Ago2-KO/HA-Ago2K212A}$ (Ago2K 212A) cancer cells (Fig. 6Hi), including both miR-3613-3p and miR-6126 (Fig. 6Hii). Ago2K212A substitution resulted in similar

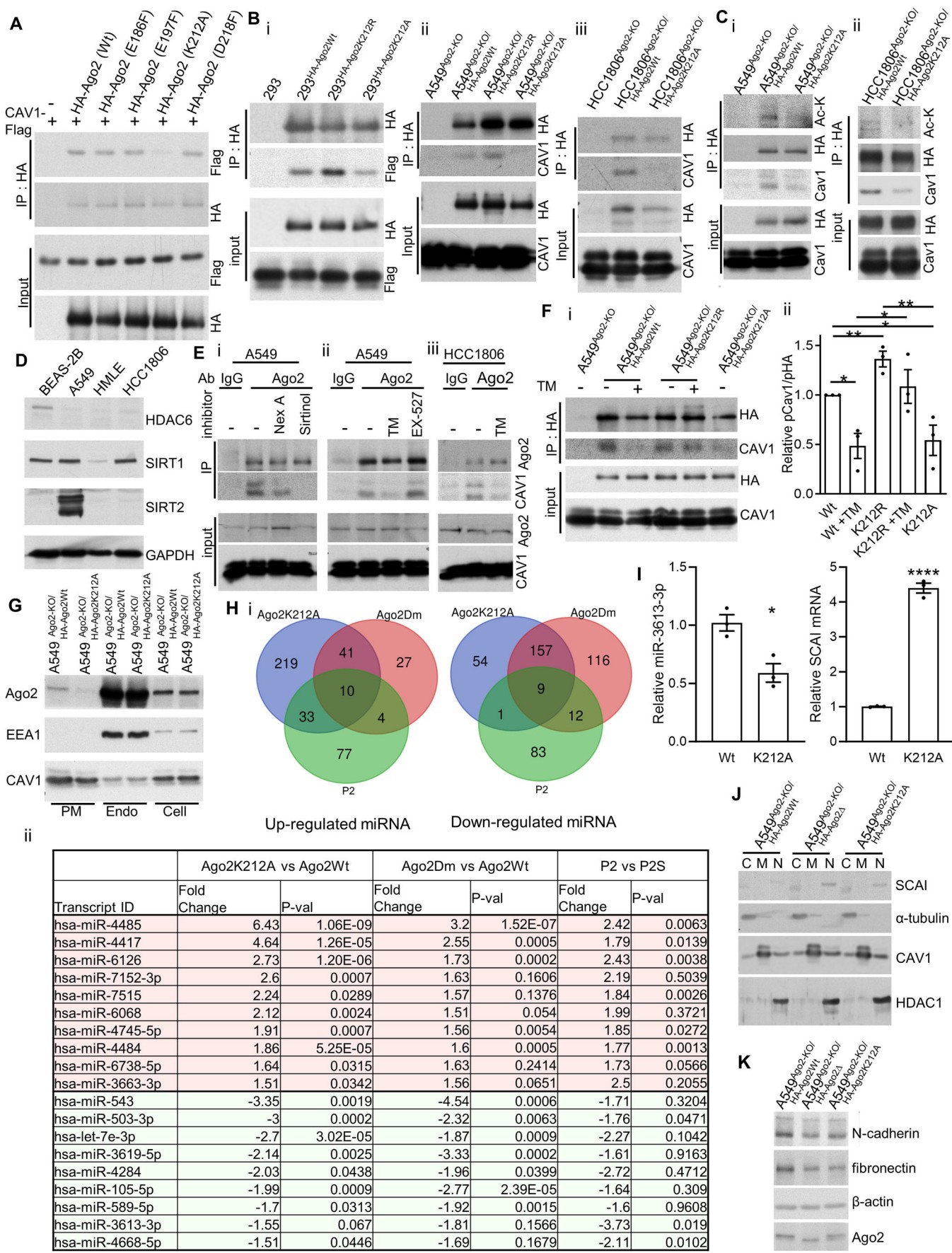

**Figure 6.  SIRT2 is required for maintaining the positive charge of Ago2 lysine 212 in the CBM for Ago2/CAV1 interaction.**

(A) In HEK293 cells, wild-type (Wt) or mutated (E186F, E197F, K212A, and D217F) HA-Ago2 was immunoprecipitated with anti-HA antibodies, and coprecipitation of CAV1-Flag was analyzed. (B) Interaction of Ago2, Ago2K212R, and Ago2K212A with CAV1. In the indicated cells, HA-Ago2 was immunoprecipitated with anti-HA antibodies, and coprecipitation of CAV1-Flag or endogenous CAV1 was analyzed. 293 in panel (i), A549$^{Ago2\text{-}KO}$ in panel (ii), and HCC1806$^{Ago2\text{-}KO}$ in panel (iii) were used as negative controls. (C) Ago2/CAV1 interaction and Ago2 acetylation attenuated by mutations in Ago2 lysine 212 (K212). In A549$^{Ago2\text{-}KO/HA\text{-}Ago2Wt}$, A549$^{Ago2\text{-}KO/HA\text{-}Ago2K212A}$ (panel i), HCC1806$^{Ago2\text{-}KO/HA\text{-}Ago2Wt}$, and HCC1806$^{Ago2\text{-}KO/HA\text{-}Ago2K212A}$ cells (panel ii), HA-Ago2 was immunoprecipitated with anti-HA antibodies. Acetylation of HA-Ago2 and coprecipitation of CAV1-Flag were analyzed. (D) Expression of deacetylase in cancer cells (A549 and HCC1806) and normal epithelial cells (BEAS-2B and HMLE), analyzed using Western blotting. GAPDH was used as a loading control. (E) In cancer cells treated with a vehicle or a deacetylase inhibitor (Nexturastat A, Sirtinol, EX-527, or TM), Ago2 was immunoprecipitated with anti-Ago2 antibodies, and coprecipitation of CAV1 was analyzed. (F) In A549$^{Ago2\text{-}KO/HA\text{-}Ago2Wt}$ and A549$^{Ago2\text{-}KO/HA\text{-}Ago2K212R}$ cells treated with vehicle or TM (10 μM), HA-Ago2 was immunoprecipitated with anti-HA antibodies, and coprecipitation of CAV1 was analyzed (panel i). A549$^{Ago2\text{-}KO}$ and A549$^{Ago2\text{-}KO/HA\text{-}Ago2K212A}$ cells were used as negative controls. The ratio of precipitated Cavl to precipitated HA-Ago2 was measured in three rounds of immunoprecipitation experiments (Panel ii). Bars are means ± SEM ($n = 3$, biological replicates). Student's $t$- test, $^*P \leq 0.05$; $^{**}P \leq 0.01$. (G) Levels of Ago2 in the plasma membrane (PM) and endosome (Endo) fractions of A549$^{Ago2\text{-}KO/HA\text{-}Ago2Wt}$ and A549$^{Ago2\text{-}KO/HA\text{-}Ago2K212A}$ cells and in the whole cells were analyzed with Western blots. EEA1: Endo marker. CAV1: PM marker. (H) The miRNA fluctuation resulting from disruption of Ago2/CAV1 interaction by P2 peptides, deletion of Ago2 CBM, and Ago2K212A substitution analyzed with miRNA arrays. Panel i: The Venn diagrams of upregulated (left) and downregulated (right) miRNAs in P2-treated A549, A549$^{Ago2\text{-}KO/HA\text{-}Ago2\Delta}$ (Ago2Dm), and A549$^{Ago2\text{-}KO/HA\text{-}Ago2K212A}$ (Ago2K212A) cancer cells. Panel ii: There are ten shared upregulated miRNAs (red) and nine shared downregulated miRNAs (green) in P2-treated A549, A549$^{Ago2\text{-}KO/HA\text{-}Ago2\Delta}$ (Ago2Dm), and A549$^{Ago2\text{-}KO/HA\text{-}Ago2K212A}$ (Ago2K212A) cancer cells, with a fold-change >1.5. (I) Expression of miR-3613-3p miRNA (left panel) and SCAI mRNA (right panel) in A549$^{Ago2\text{-}KO/HA\text{-}Ago2Wt}$ (Wt), and A549$^{Ago2\text{-}KO/HA\text{-}Ago2K212A}$ (K212A) cells. Each spot describes the relative miRNA level of the sample to that of the A549$^{Ago2\text{-}KO/HA\text{-}Ago2Wt}$ sample. Bars are means ± SEM ($n = 3$, biological replicates). Student's $t$-test, $^*P \leq 0.05$; $^{****}P \leq 0.0001$. (J) Expression of SCAI proteins in A549$^{Ago2\text{-}KO/HA\text{-}Ago2Wt}$, A549$^{Ago2\text{-}KO/HA\text{-}Ago2\Delta}$, and A549$^{Ago2\text{-}KO/HA\text{-}Ago2K212A}$ cancer cells. The levels of proteins were analyzed in the MFs (M), CFs (C), and NFs (N) of the cells. α-Tubulin: CF marker; CAV1: MF marker; HDAC1: NF marker. (K) Expression of N-cadherin and fibronectin in A549$^{Ago2\text{-}KO/HA\text{-}Ago2Wt}$, A549$^{Ago2\text{-}KO/HA\text{-}Ago2\Delta}$, and A549$^{Ago2\text{-}KO/HA\text{-}Ago2K212A}$ cancer cells. The quantitation for Western blots in this figure is presented in Appendix Fig. S3. Source data are available online for this figure.

effects as P2 peptides (Fig. 3) and deletion of Ago2 CBM domain (Fig. EV3), including miR-3613-3p decrease (Fig. 6I, left panel), SCAI mRNA (Fig. 6I, right panel) and protein (Fig. 6J; Appendix Fig. S3Gi) increases, and mesenchymal marker decreases (Fig. 6K; Appendix Figs. S3Gii,Giii). These results indicate that blockage of Ago2/CAV1 interaction by Ago2K212A substitution has the same effects as those of P2 peptides and Ago2 CBM deletion on miRNA expression and function.

Because deacetylated Ago2 K212 is responsible for Ago2/CAV1 interaction and the interaction-mediated miRNA function in cancer cells, we evaluated the effects of Ago2 K212 acetylation on the Ago2/CAV1 interaction-dependent behaviors of cancer cells (i.e., invasion and tumorsphere formation). A549$^{Ago2\text{-}KO/HA\text{-}Ago2K212R}$ cells preserved their invasion and tumorsphere formation capabilities, but A549$^{Ago2\text{-}KO/HA\text{-}Ago2K212A}$ cells exhibited decreased invasion (Fig. 7Ai,Aii) and tumorsphere formation (Fig. 7Aiii). Blockage of Ago2/CAV1 interaction with Ago2 K212 substitution with alanine, which mimics K212 acetylation, reduced the invasion and tumorsphere formation of cancer cells. TM reduced the invasion of A549$^{Ago2\text{-}KO/HA\text{-}Ago2Wt}$ cells by 54%, but it reduced that of A549$^{Ago2\text{-}KO/HA\text{-}Ago2K212R}$ cells by only 20% (Fig. 7Bi,Bii). TM reduced the tumorsphere formation of A549$^{Ago2\text{-}KO/HA\text{-}Ago2Wt}$ cells by 46%, but it did not affect that of A549$^{Ago2\text{-}KO/HA\text{-}Ago2K212R}$ cells (Fig. 7C). Suppression of SIRT2-mediated deacetylation reduced the invasion and tumorsphere formation of cancer cells, and K212R substitution in Ago2 prevented TM from exerting the same effect. These results indicate that SIRT2-mediated deacetylation of Ago2 lysine 212 plays a crucial role in the Ago2/CAV1 interaction-dependent behaviors of cancer cells (i.e., invasion and tumorsphere formation).

To further investigate the role of Ago2 K212 in the lung targeting and tumor formation of cancer cells in vivo, A549$^{Ago2\text{-}KO/HA\text{-}Ago2Wt}$, A549$^{Ago2\text{-}KO/HA\text{-}Ago2K212R}$, and A549$^{Ago2\text{-}KO/HA\text{-}Ago2K212A}$ cells were injected into SCID mice through tail veins (Fig. 7Di,Ei). While cancer cells were detected by qPCR in the lungs of mice injected with A549$^{Ago2\text{-}KO/HA\text{-}Ago2Wt}$ and A549$^{Ago2\text{-}KO/HA\text{-}Ago2K212R}$ cells, no cancer cells were detected in the lungs of mice injected with A549$^{Ago2\text{-}KO/HA\text{-}Ago2K212A}$ cells at 96 h

after injection (Fig. 7Dii). At 4 weeks after injection, multiple tumors were detected using IVIS imaging (Fig. 7Eii,Eiii), histological analysis (Fig. 7Eiv,Ev), and qPCR (Fig. 7Evi) in the lungs of mice injected with A549$^{Ago2\text{-}KO/HA\text{-}Ago2Wt}$ and A549$^{Ago2\text{-}KO/HA\text{-}Ago2K212R}$ cells, whereas only a few tumors were detected in the lungs of mice injected with A549$^{Ago2\text{-}KO/HA\text{-}Ago2K212A}$ cells. In addition, multiple cancer cells were detected in the lymph nodes of mice injected with A549$^{Ago2\text{-}KO/HA\text{-}Ago2Wt}$ and A549$^{Ago2\text{-}KO/HA\text{-}Ago2K212R}$ cells but not in those injected with A549$^{Ago2\text{-}KO/HA\text{-}Ago2K212A}$ cells (Fig. 7Evii). These results indicate that Ago2 K212 plays an essential role in the early stage of lung targeting, tumor formation, and lymph node spread of cancer cells.

## CAV1 interaction and plasma membrane association of Ago2 increase in human metastatic tumors

To support our findings, we examined Ago2/CAV1 interaction in paired human breast primary (P) and metastatic (M) tumors (Fig. 8A,B). We also analyzed the levels of coprecipitated CAV1 by anti-Ago2 antibodies, normalized to the levels of precipitated Ago2, in primary tumors and paired metastatic tumors in the lymph nodes (M) of each patient (Fig. 8B). Compared with primary tumors, the levels of Ago2-bound CAV1 were considerably higher in paired metastatic tumors in 90% of the patients, indicating that metastatic tumors are associated with increased Ago2/CAV1 interaction.

Because Ago2/CAV1 interaction regulates the distribution of Ago2 in intracellular compartments of cancer cells in culture (Fig. 2), we examined the effects of this interaction on the distribution of Ago2 in tumors in vivo. In tumors of A549$^{Ago2\text{-}KO/HA\text{-}Ago2Wt}$ cells, wild-type Ago2 was located at the plasma membrane, which was labeled with anti-cytokeratin (8/18) antibodies (Fig. 8C, Wt) (Hembrough et al, 1996; Nava-Acosta and Navarro-Garcia, 2013; Wells et al, 1997). By contrast, in tumors of A549$^{Ago2\text{-}KO/HA\text{-}Ago2\Delta}$ cells, CBM-deleted Ago2 (Ago2Δ) was distributed throughout both the nuclei and cytoplasm (Fig. 8C, Dm). Therefore, the plasma membrane association of Ago2 in tumor cells depends on the Ago2/CAV1 interaction mediated by the CBM of Ago2. We further

evaluated the distribution of Ago2 proteins in paired human breast primary tumors and metastatic tumors (Fig. 8D). Ago2 was distributed either in the entire tumor cells (#7512, #8364, #8549, #9455, #10482, and #10549; Fig. 8D) or in the nuclei of tumor cells (#7194 and #9456; Fig. 8D) in primary tumors. In contrast, Ago2

translocated to the plasma membrane of paired metastatic tumors (Fig. 8D). The levels of Ago2 associated with the plasma membrane were measured as the percentage of Ago2 signals overlapping with cytokeratin (8/18) signals (Fig. 8E). Compared with primary tumors, the levels of Ago2 associated with the plasma membrane

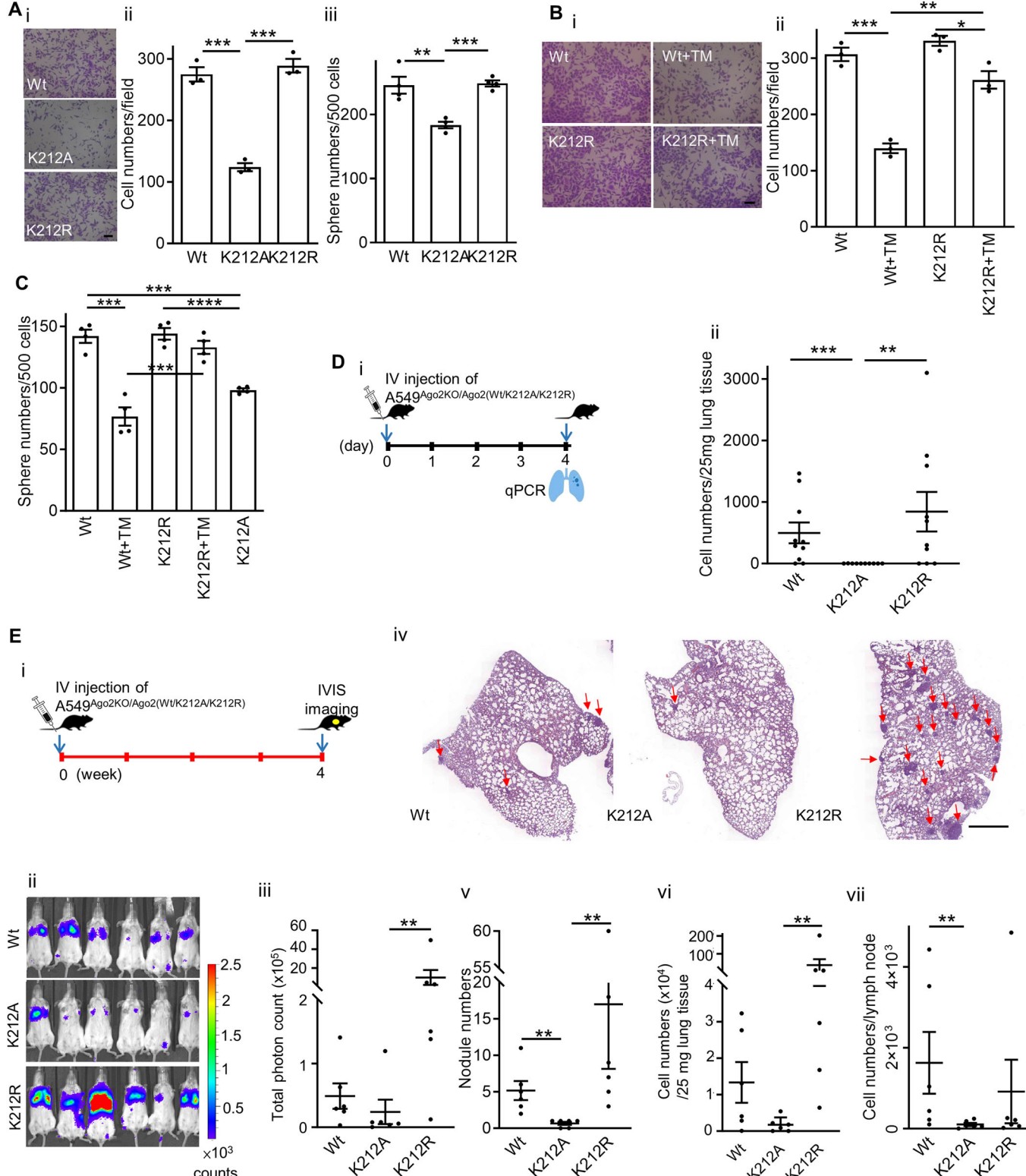

Figure 7. Positive charge of Ago2 lysine 212 plays a major role in Ago2/CAV1 interaction-dependent cancer cell behaviors.

(A) A549^Ago2-KO/HA-Ago2Wt (Wt), A549^Ago2-KO/HA-Ago2K212R (K212R), and A549^Ago2-KO/HA-Ago2K212A (K212A) cells were subjected to migration (panels i and ii) and tumorsphere formation (panel iii) assays. Bars are means ± SEM ($n = 3$, biological replicates). Student's $t$-test, ""$P ≤ 0.01$; """$P ≤ 0.001$. Scale bar = 100 μm. (B) A549^Ago2-KO/HA-Ago2Wt and A549^Ago2-KO/HA-Ago2K212R cells were subjected to migration assay under vehicle or TM treatment. Bars are means ± SEM ($n = 3$, biological replicates). Student's $t$-test, "$P ≤ 0.05$; ""$P ≤ 0.01$; """$P ≤ 0.001$. Scale bar = 100 μm. (C) A549^Ago2-KO/HA-Ago2Wt, A549^Ago2-KO/HA-Ago2K212R, and A549^Ago2-KO/HA-Ago2K212A cells were subjected to a tumorsphere formation assay after vehicle or TM treatment. Bars are means ± SEM ($n = 4$, biological replicates). Student's $t$-test, """$P ≤ 0.001$; """"$P ≤ 0.0001$. A549^Ago2-KO/HA-Ago2K212A was used as a negative control. (D) Panel (i): Scheme of an animal experiment, indicating the time points of IV injection of A549^Ago2-KO/HA-Ago2Wt, A549^Ago2-KO/HA-Ago2K212R, and A549^Ago2-KO/HA-Ago2K212A cells and tissue collection. Panel (ii): Quantification of lung targeting by A549 cells at 96 h after administration. qPCR was used to detect A549 cell genomic DNA in tissues. Each spot describes the number of tumor cells per 25 mg of lung tissue in an individual mouse. Bars are means ± SEM ($n = 10$). Mann–Whitney $U$-test, ""$P ≤ 0.01$; """$P ≤ 0.001$. (E) Panel i: Scheme of an animal experiment, indicating the time points of IV injection of A549^Ago2-KO/HA-Ago2Wt, A549^Ago2-KO/HA-Ago2K212R, and A549^Ago2-KO/HA-Ago2K212A cells and IVIS imaging and tissue collection. Panel (ii): Tumor colonization was monitored using noninvasive imaging at 4 weeks after injection. Panel (iii): Luminescent counts of lung area in the images of panel (ii). Panels (iv) and (v): Number of tumor nodules, indicated with arrows, in the lung sections of mice. Scale bar = 0.1 cm. Panels (vi) and (vii): Numbers of tumor cells in the lungs (panel vi) and lymph nodes (panel vii) of mice were quantified using qPCR for human gDNA. Each spot describes the counts of an individual mouse. Bars are means ± SEM ($n = 6$). Mann–Whitney $U$-test, ""$P ≤ 0.01$. Source data are available online for this figure.

were substantially higher in 89% of paired metastatic tumors (Fig. 8E). These results indicated that increased Ago2/CAV1 interaction is associated with increased plasma membrane association of Ago2 in metastatic tumors.

Examination of the distribution of Ago2 in various breast diseases (Figs. 8F and EV4A,B) revealed that in mammary epithelial cells of breast tissues with hyperplasia, inflammation, or fibroadenoma and cancer-adjacent breast tissues, Ago2 was primarily located in the nuclei (Figs. EV4Ai,Aii) and rarely overlapped with cytokeratin 8/18. In primary breast carcinomas, Ago2 was distributed in the entire tumor cells, both in the nuclei and in the cytoplasm (Fig. EV4Aiii). By contrast, in metastatic tumor cells in lymph nodes, Ago2 was located at the plasma membrane, labeled with anti-cytokeratin (8/18) antibodies (Fig. EV4Aiv). Compared with noncancerous breast tissues (i.e., normal breast tissues, breast tissues with hyperplasia, inflammation, or fibroadenoma, and cancer-adjacent breast tissues), the levels of plasma membrane-associated Ago2 were significantly higher in primary carcinomas ($P = 2.64 \times 10^{-10}$; Fig. 8F). Among different stages of carcinoma, the levels of plasma membrane-associated Ago2 increased in stage IIIB carcinomas, which are large primary tumors that have spread to nearby lymph nodes (Fig. EV4B). Among all types of tissues with breast disease, metastatic carcinomas had the highest levels of plasma membrane-associated Ago2 (Fig. 8F). We also analyzed the distribution of Ago2 in lung tissues and lung carcinomas (Fig. EV4C). Compared with noncancerous lung tissues, the levels of plasma membrane-associated Ago2 were significantly higher in lung carcinomas ($P = 1.37 \times 10^{-4}$; Fig. EV4C). These results indicated that both Ago2/CAV1 interaction and plasma membrane association of Ago2 increase with human carcinoma progression and metastasis.

## Ago2 is sorted into extracellular vesicles through interaction with CAV1

In addition to their action in cells, RNAs, mainly miRNAs, are secreted through RNA-binding proteins or vesicles into the extracellular space to regulate cell physiology (Patton et al, 2015; Valadi et al, 2007). We also investigated the role of Ago2/CAV1 interaction in sorting Ago2 into EVs. In addition to small EV markers, e.g., exosome markers CD63, TSG101, and CD81, Ago2 proteins were detected in the P4 fraction of A549 cancer cell–conditioned media (Figs. 9A and EV5A), which contains small EVs measuring approximately 100 nm (Fig. EV5B), but not in the

P3 fraction which contains large EVs (Fig. EV5B). Generally, to prevent contamination with protein aggregates, small vesicles in the P4 fraction can be further purified using a continuous iodixanol density gradient (Lin et al, 2017a). In this study, both Ago2 and CAV1 proteins co-existed with small EV markers (i.e., CD81, CD63, CD9, and TSG101) in the ~20% iodixanol subfraction (Fig. EV5C, fifth fraction), indicating that the secreted Ago2 was associated with EVs. Membrane lysis with RIPA buffer was also required for the complete digestion of EV-associated Ago2 and CAV1 by protease K (Fig. EV5D), indicating the presence of both Ago2 and CAV1 proteins in the cancer cell-derived EVs.

We investigated whether Ago2/CAV1 interaction is necessary for sorting Ago2 into EVs. Ago2/CAV1 interaction was detected in cancer cell-derived EVs (Figs. 9B and EV5E). Small EVs were collected from cells expressing HA-Ago2(1–226) or HA-Ago2(226–859) fragments. In these EVs, HA-Ago2(1–226), which binds CAV1 (Fig. 1G), was detected, whereas HA-Ago2(226–859), which does not bind CAV1 (Fig. 1F), was not (Figs. 9C and EV5F). While HA-Ago2Wt was sorted into the EVs, the removal of the CBM prevented HA-Ago2Δ from being sorted into EVs (Figs. 9D and EV5G). These results indicate that Ago2 is sorted into EVs through interactions with CAV1. Ago2/CAV1 interaction regulates the distribution of Ago2 in intracellular and extracellular compartments, including the cytoplasm, plasma membrane, and EV.

## Ago2/CAV1 interaction is associated with EV-mediated release of miRNA-3613-3p from metastatic tumors as a biomarker

Ago2/CAV1 interaction regulates Ago2 sorting into EVs. Furthermore, disrupting Ago2/CAV1 interaction also decreased the level of miRNA-3613-3p in the cancer cell-derived EVs (Fig. 9E). The elevated miRNA-3613-3p in cancer cells with Ago2/CAV1 interaction can be released via EVs. We further evaluated whether the elevated miRNAs in cancer cells in which Ago2/CAV1 interaction is maintained could be released via EVs into the circulatory system and be detected in plasma as biomarkers. Figure 9F shows the miRNAs that were differentially released through plasma EVs of A549^Ago2-KO/HA-Ago2Wt (Wt) and A549^Ago2-KO/HA-Ago2Δ (Dm) tumor-bearing mice (Fig. 9F; fold difference >1.5), analyzed with miRNA microarray. We compared the miRNAs of Fig. 9F and the miRNAs which were differentially expressed in A549^Ago2-KO/HA-Ago2Wt vs. A549^Ago2-KO/HA-Ago2Δ cancer cells (Fig. 6H, Ago2Dm). The EV-

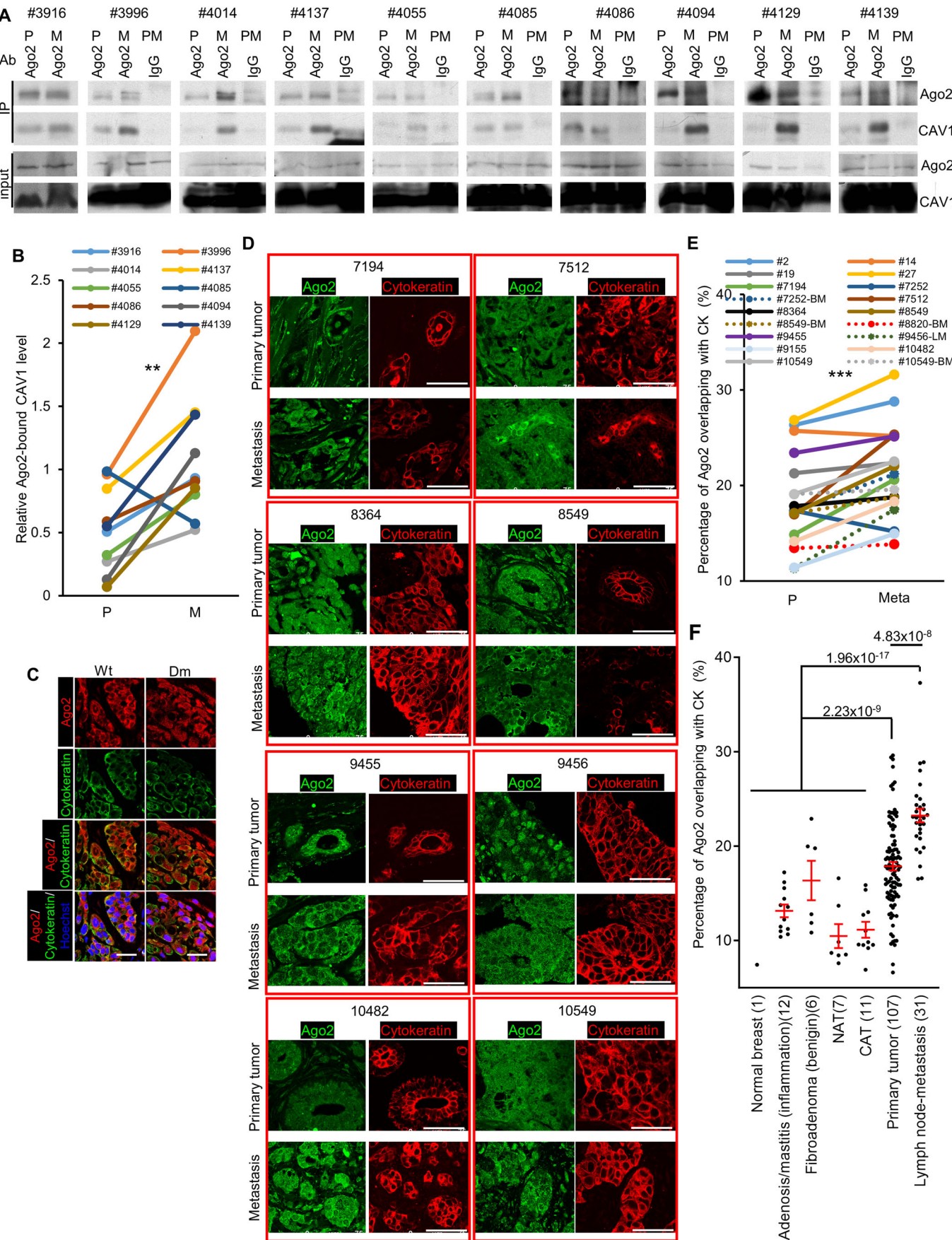

◄

**Figure 8. Increased Ago2/CAV1 interaction in human metastatic tumors depends on the plasma membrane association of Ago2.**

(A) In human primary breast tumors (P) and paired lymphatic metastases (M), Ago2 was immunoprecipitated with anti-Ago2 antibodies, and coprecipitation of CAV1 was analyzed. Each number indicates an individual patient. (B) The ratio of coprecipitated CAV1 to precipitated Ago2 was calculated for each sample in panel A ($n = 10$, patient number). Student's $t$-test, $**P \leq 0.01$. (C) Distribution of Ago2 in A549$^{Ago2-KO/HA-Ago2Wt}$ and A549$^{Ago2-KO/HA-Ago2\Delta}$ tumors. Ago2 was stained with anti-Ago2 antibodies (Ago2, red), and cytokeratin was stained with anti-cytokeratin 8/18 antibodies (Cytokeratin, green). Cell nuclei were stained with Hoechst 33258 (blue). Scale bar = 25 μm. (D) In human primary breast tumors and paired metastases, Ago2 was stained with anti-Ago2 antibodies (Ago2, green), and cytokeratin was stained with anti-cytokeratin 8/18 antibodies (Cytokeratin, red). Cell nuclei were stained with Hoechst 33258 (blue). Images of representative samples are shown. Each number indicates an individual patient. Scale bar = 50 μm. (E) Ago2 distribution is evaluated with the percentage of Ago2 overlapping with cytokeratin 8/18 in primary breast tumors (P) and paired metastases (Meta). BM bone metastasis, LM lung metastasis, Meta without labeling: lymphatic metastasis. ($n = 18$, patient number). Student's $t$-test, $***P \leq 0.001$. (F) Ago2 distribution in images of normal breast tissue, inflamed breast tissue (mastitis), and breast tumors in a breast disease spectrum array. Representative immunofluorescence images are depicted in Fig. EV4A. Each spot describes the percentage of Ago2 overlapping with cytokeratin 8/18 in a tissue sample. Bars are means ± SEM. The sample size ($n$) of each category is depicted on the $x$-axis. Student's $t$-test, $P$ values are indicated in each plot. Source data are available online for this figure.

mediated miRNA release of the tumor-bearing mice did not necessarily correspond with the miRNA expression of the cancer cells. Only miRNA-3613-3p and miRNA-877-5p were altered with Ago2/CAV1 interaction in both cancer cells and plasma EV of cancer cell-bearing mice (Fig. 9F, Blue). The EV-mediated release of miRNAs, miRNA-3613-3p, miRNA-877-5p, and miRNA-6126, in the plasma of Wt and Dm tumor-bearing mice and non-tumor-bearing mice (Ctrl) was further analyzed using qPCR (Fig. 9G). Corresponding to the miRNA expression in cancer cells, the elevated levels of miRNA-3613-3p were detected in plasma EVs of A549$^{Ago2-KO/HA-Ago2Wt}$ tumor-bearing mice, and disruption of Ago2/CAV1 interaction decreased EV-mediated miR-3613-3p release in plasma EV of A549$^{Ago2-KO/HA-Ago2\Delta}$ tumor-bearing mice. The elevated EV miRNA-3613-3p in the plasma of tumor-bearing mice is associated with Ago2/CAV1 interaction in the tumor cells.

Since Ago2/CAV1 interaction increases with human carcinoma progression and metastasis, we also investigated whether EV-mediated miR-3613-3p release increases in the plasma of patients with metastatic tumors. Quantitative PCR analysis of EV miRNA-3613-3p, miRNA-877-5p, and miRNA-6126 in the plasma of patients with breast cancer at different stages showed that only miRNA-3613-3p was increased in the plasma of patients at stage II and stage IV; such patients had invasive and metastatic tumors with increased Ago2/CAV1 interaction (Fig. 9H). Consistent with the results of cultured cancer cells (Fig. 9E) and tumor xenografts (Fig. 9G), this increase in EV miRNA-3613-3p in patient plasma was positively correlated with increased Ago2/CAV1 interaction in human metastatic carcinomas. These results indicate that elevated EV miRNA-3613-3p in plasma may be a biomarker of human metastatic carcinomas.

## Discussion

CAV1 functions as both a tumor promotor and a tumor suppressor, and this dual behavior of CAV1 often depends on downstream interacting partners (Bender et al, 2002; Burgermeister et al, 2008; Shatz and Liscovitch, 2008). CAV1 interacts with cholesterol in lipid rafts and proteins through the CSD (Byrne et al, 2012; Hoop et al, 2012). This interaction allows CAV1 to bridge its interacting proteins and lipid rafts on the plasma membranes and to regulate the functions of the interacting proteins (Fridolfsson et al, 2014; Nwosu et al, 2016). In this study, we discovered that Ago2 directly bound to the CSD of CAV1 (Fig. 1C–E), and we identified an

aromatic-rich sequence, amino acids 199–212, as the CBM of Ago2 (Fig. 1F–J). We also discovered that Ago2/CAV1 interaction was disrupted by the mutation of all four aromatic residues in the CBM (Fig. 1K), whereas individual mutations had no effect on Ago2/CAV1 interaction (Fig. EV1O). These results indicated that, instead of directly binding CAV1, the role of aromatic residues in the CBM of Ago2 is to maintain the tertiary protein structure required for CAV1 interaction (Figs. 1K,L and EV1N). These findings are consistent with those of other CAV1-interacting proteins, such as Slo1 (Alioua et al, 2008; Brainard et al, 2009), ILK1 (Chun et al, 2005a), PDK1 (Chun et al, 2005b), P-glycoprotein (Jodoin et al, 2003), and insulin receptor (Nystrom et al, 1999). Charged residues often confer direct binding and reinforce protein binding specificity with partners during protein interaction (Zhou and Pang, 2018). In this study, we discovered that the positive charge of K212, which is located next to an aromatic residue in the CBM, played a major role in the Ago2/CAV1 interaction (Fig. 6A, B). Substitution of K212 with charged arginine maintained Ago2/CAV1 interaction, whereas substitution with uncharged alanine disrupted this interaction. These findings are consistent with those of another CAV1-binding protein, Spry2. In the putative CBM of Spry2, the substitution of positively charged arginine with negatively charged aspartate negatively affects its interaction with CAV1 (Cabrita et al, 2006). In both cases, the positively charged amino acids in the CBM mediate the interactions between CAV1 and CAV1-binding proteins (i.e., Ago2 and Spry2).

The majority of Ago2 is diffusely distributed in the cytoplasm (Meister et al, 2004). When cells are subjected to stress or enter different states, Ago2 forms foci or accumulates to specific structures, such as nuclei (Gagnon et al, 2014), endosomes (McKenzie et al, 2016), ER (Barman and Bhattacharyya, 2015; Stalder et al, 2013), GW-bodies, P-bodies (Leung and Sharp, 2013; Patel et al, 2016), stress granules (Detzer et al, 2011), midbodies (Casey et al, 2019), or apical junctions (Kourtidis et al, 2017). Because of its association with a variety of interacting proteins, Ago2 can be detected in various locations. In this study, we compared Ago2 distribution among the plasma membranes, endosomes, and ER of normal epithelial cells and cancer cells. We observed that the association of Ago2 with the plasma membranes and with endosomes increases in cancer cells, compared to that of normal epithelial cells (Figs. 2C,Di,Diii, E–G and EV2E, left panel). The increased plasma membrane association of Ago2 in cancer cells corresponds with the observation that PDAC progression is associated with increased plasma

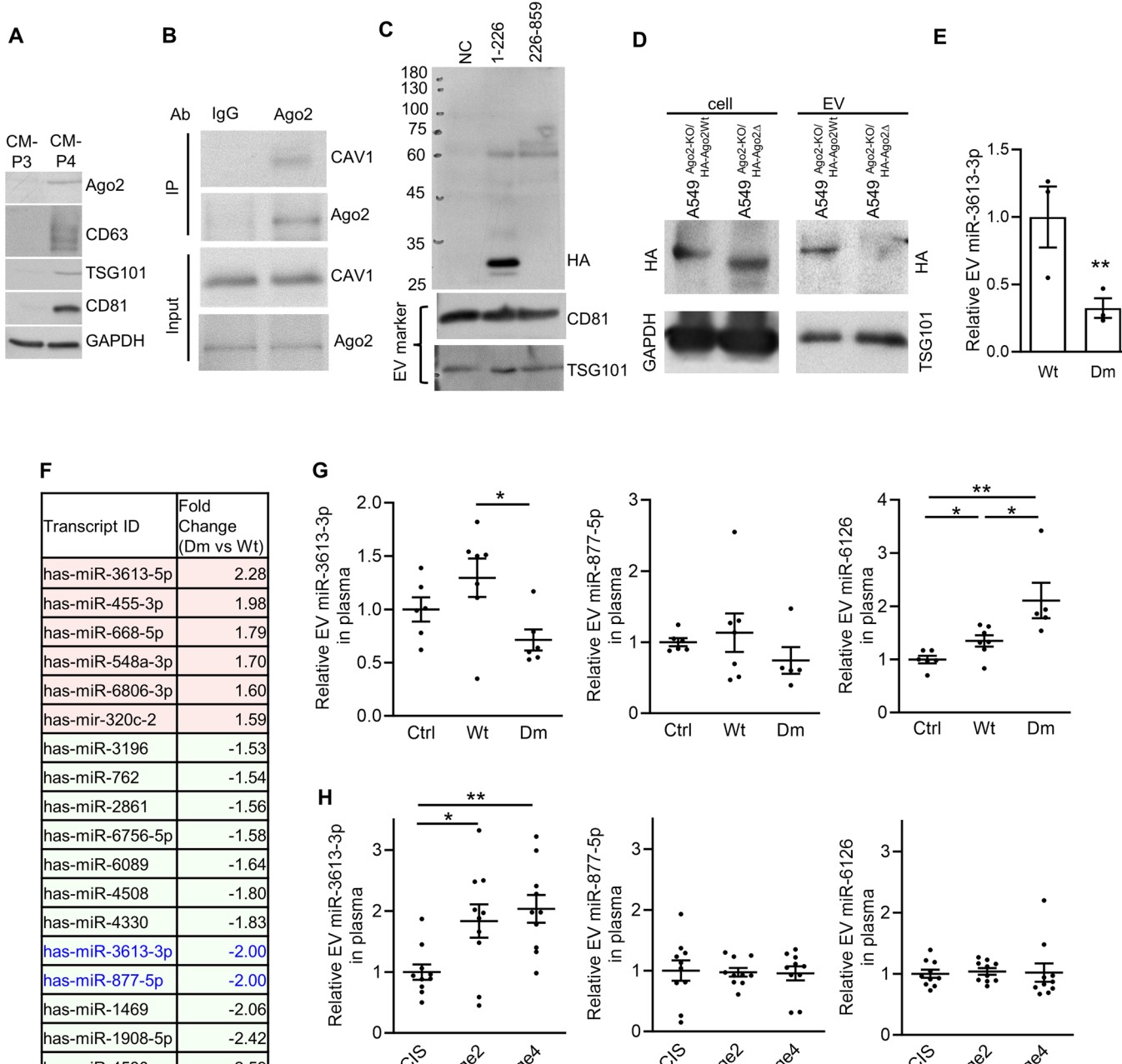

© The Author(s)

EMBO reports Volume 25 | May 2024 | 2441–2478

membrane localization of Ago2 (Shankar et al, 2020). CAV1 has been found predominantly in the plasma membranes and marginally in endosomes and ER (Albacete-Albacete et al, 2020; Williams and Lisanti, 2004). It corresponds with our observation that blocking Ago2/CAV1 interaction disrupts Ago2 association with the plasma membranes, which contain abundant CAV1 (Figs. 2H,I, 6G), but does not significantly interfere with Ago2 association with endosomes and ER, which only contain slight CAV1, in the cancer cells (Figs. 2Div, Dvi, 6G and EV2E). Ago2/CAV1 interaction allows specific Ago2 association with lipid rafts, which are rich in CAV1, of the plasma membranes (Fig. EV2G). There are determinants, other than Ago2/CAV1 interaction, responsible for Ago2 association with endosomes and ER (McKenzie et al, 2016;

Stalder et al, 2013). We observed that S387 phosphorylation of Ago2, the determinant of Ago2 association with endosomes in cancer cells (McKenzie et al, 2016), did not affect Ago2/CAV1 interaction (Fig. EV1H). It supports that Ago2/CAV1 interaction is not predominantly responsible for Ago2 association with endosomes in cancer cells.

RNA compartmentalization within the cytosol is a key mechanism of local translation-mediated regulation of protein expression (Holt and Bullock, 2009). RNA compartmentalization was initially discovered in polarized cells and migrating cells and is currently widely regarded as a mechanism for regulating numerous physical and pathological processes (Fernandopulle et al, 2021; Katz et al, 2012; Rongo et al, 1995). One of the mechanisms of RNA

**Figure 9. Ago2/CAV1 interaction is associated with EV-mediated miRNA-3613-3p release into the circulatory system.**

(A) Ago2 in the fraction containing small extracellular vesicles (EVs). Ago2, CD63, TSG101, CD81, and GAPDH proteins were measured using Western blotting in the P4 and P3 fractions of A549 cell–conditioned media. (B) Ago2/CAV1 interaction in A549 cell-derived EVs. In EVs, Ago2 was immunoprecipitated with anti-Ago2 antibodies, and coprecipitation of CAV1 was analyzed. (C) Ago2 amino acids 1–226, containing a CAV1-interacting domain, sorted into EVs, unlike Ago2 amino acids 226–859. The EVs of HEK293 cells expressing HA-Ago2(1–226) or HA-Ago2(226–859) were collected and subjected to Western blotting to measure HA-Ago2 fragments. The EVs of HEK293 cells were used as negative controls. (D) CBM of Ago2 is necessary for its sorting into EVs. In the cell and EV fractions of A549 $^{Ago2-KO/HA-Ago2Wt}$ and A549 $^{Ago2-KO/HA-Ago2\Delta}$ cells, HA-Ago2Wt, HA-Ago2$\Delta$, TSG101, and GAPDH were measured using Western blotting. (E) Levels of EV miR-3613-3p in A549$^{Ago2-KO/HA-Ago2Wt}$ (Wt) and A549 $^{Ago2-KO/HA-Ago2\Delta}$ (Dm) cells measured using qPCR, normalized to 5S RNA, in EVs. Each spot describes the relative EV miR-3613-3p level of the sample to that of the A549 $^{Ago2-KO/HA-Ago2Wt}$ cells. Bars are means ± SEM ($n = 3$, biological replicates). Student's $t$-test, $^{**}P \le 0.01$. (F, G) Levels of miRNAs in the plasma EVs of mice bearing A549$^{Ago2-KO/HA-Ago2Wt}$ (Wt) and A549$^{Ago2-KO/HA-Ago2\Delta}$ (Dm) tumors. The same amount of plasma was collected from the mice and subjected to EV isolation and miRNA array assays. Panel (F) illustrates EV miRNAs that differed significantly between Wt and Dm mice by more than 1.5 folds, measured by miRNA array assays. The green indicates downregulated miRNAs and the red indicates upregulated miRNAs within the plasma EVs of mice bearing A549$^{Ago2-KO/HA-Ago2\Delta}$ (Dm) tumors. The miRNAs altered in both A549$^{Ago2-KO/HA-Ago2Wt}$ cells vs A549$^{Ago2-KO/HA-Ago2\Delta}$ cells and in plasma EVs of mice bearing A549$^{Ago2-KO/HA-Ago2Wt}$ (Wt) and A549$^{Ago2-KO/HA-Ago2\Delta}$ (Dm) tumors are labeled in blue. In panel (G), differences are validated using qPCR. Each spot describes relative EV miRNAs in the plasma of mice to the average amount of mice not bearing tumor (Crtl). Bars are means ± SEM ($n = 5$–7, biological replicates). Student's $t$-test, $^{*}P \le 0.05$; $^{**}P \le 0.01$. (H) Levels of exosomal miRNAs in the plasma of patients with breast tumors at different stages. Plasma EV RNA was collected and analyzed using qPCR. Each spot describes the relative EV miRNA in the plasma of patients to the average amount of patients at stage DCIS. Bars are means ± SEM ($n = 8$ and 10, patient number). Student's $t$-test, $^{*}P \le 0.05$; $^{**}P \le 0.01$. Source data are available online for this figure.

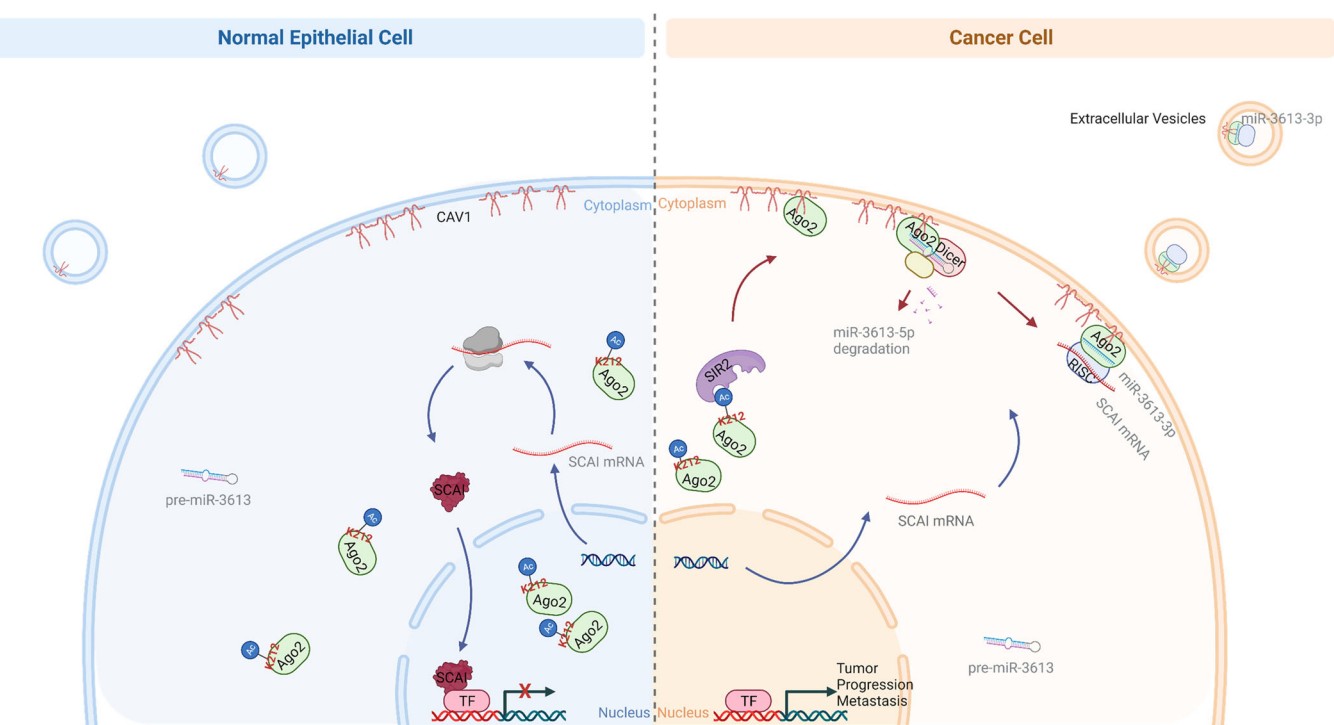

**Figure 10. A proposed mechanism for the formation of Ago2/CAV1 interaction and its roles in cancer cells.**

Left panel: In normal epithelial cells, K212-acetylated Ago2 does not interact with CAV1 on the plasma membrane. SCAI is expressed to repress gene expression required for tumor promotion and metastasis (blue arrows). Right panel: In cancer cells, Ago2 is deacetylated by SIR2 at K212 and is recruited to the plasma membrane through Ago2/CAV1 interaction (red arrows). MiR-3613-3p is generated through the maturation process catalyzed by CAV1-associated Ago2. During miRNA maturation, miR-3613-5p is cleaved by CAV1-associated Ago2 and degraded (red arrows). Subsequently, miR-3613-3p-mediated SCAI translational suppression (blue arrows) and release of miR-3613-3p-containing EVs (black arrows) happen in an Ago2/CAV1-dependent manner.

compartmentalization is the granule formation of RNAs and RNA-binding proteins (RBPs) (Banani et al, 2017). Through Ago2/CAV1 interaction, Ago2 is recruited into a distinct region associated with plasma membranes or caveolae, which results in an increase in specific miRNAs and, in turn, increases miRNA-mediated translational suppression in distinct regions of cancer cells. These results indicated that, in cancer cells, a pool of Ago2 is located at the

caveolae or lipid rafts of the plasma membranes. By interacting with CAV1, this pool regulates the selective miRNA-mediated translational repression of genes involved in cancer development in a compartment-dependent manner (Fig. 10, red arrows in right panel). For example, many types of human cancer are associated with increased miR-3613-3p expression, as measured in clinical tumor samples (Castro-Magdonel et al, 2017; Chatterjee al, 2015;

Edward et al, 2015; Fricke et al, 2015; Pu et al, 2016). However, in different types of cancer, miR-3613-3p functions as an oncogene (Boratyn et al, 2016; Castro-Magdonel et al, 2017) or tumor suppressor (Song et al, 2019). This discrepancy suggests that, in addition to miR-3613-3p expression, a secondary event regulates the functions of miR-3613-3p in a context-dependent manner. In this study, we discovered that Ago2/CAV1 interaction plays a major role in the miR-3613-3p-mediated suppression of SCAI (Fig. 3E–H; Appendix Fig. S1F). Many of these Ago2/CAV1-dependent aggressive phenotypes of cancer cells (i.e., invasion (Brandt et al, 2009; Chen et al, 2014; Kressner et al, 2013; Wang et al, 2019), metastasis (Zhang et al, 2022), EMT (Fintha et al, 2013), drug resistance (Zhao et al, 2019), and stemness (Chen et al, 2014), are associated with miR-3613-3p-mediated suppression of SCAI. MicroRNA-3613-3p-mediated suppression of SCAI depends on the secondary event, Ago2/CAV1 interaction, and contributes to the aggressive phenotypes of cancer cells.

Overall, our findings suggest that the different locations of Ago2, which are regulated by Ago2 interacting proteins (e.g., CAV1), are responsible for context-dependent regulation in gene silencing processes. These Ago2 interacting proteins have regulatory effects on both function and activity. In this study, we discovered a direct interaction between Ago2 and CAV1 in cancerous epithelial cells but not in normal epithelial cells (Fig. 1A). As shown in Fig. 8A, this interaction increases in human metastatic tumors. This condition-dependent interaction suggests a regulatory role of Ago2/CAV1 interaction in the distribution and function of Ago2 in carcinoma progression and metastasis. This mechanism of context-dependent regulation of Ago2 is complemented by other condition-specific interactions with Ago2, which regulate its function in a tissue- or cell-context-dependent manner. For example, in epithelial cells, recruitment of Ago2 into adherens junctions is required for silencing a set of mRNAs and enabling epithelial homeostasis (Kourtidis et al, 2017). In neural cells, Nova1 interacts with Ago2 to regulate the development and plasticity of synapses by controlling neuronal RISC function (Storchel et al, 2015). Under neuronal stimulation, Ago2 is recruited into endosomes by the interacting protein PICK1, which inhibits Ago2-mediated repression in the synapses (Antoniou et al, 2014).

Circulating miRNAs are secreted and protected by cell-derived membrane vesicles [e.g., exosomes (Valadi et al, 2007; Zhang et al, 2010)] or RBP–miRNA complexes [e.g., Ago2-miRNA complex (Arroyo et al, 2011; Turchinovich et al, 2011)]. In this study, we discovered that Ago2 is secreted by cancer cells through EVs (Fig. 9A). It has also been shown by others and us in previous studies that the plasma membrane-associated proteins are sorted into EVs in a CAV1-dependent manner. Cav1 relocalizes to multivesicular bodies (MVB) through endocytosis, and depletion of CAV1 selectively decreases plasma membrane-associated proteins in EVs (Albacete-Albacete et al, 2020; Lin et al, 2017b). The evidence suggests that CAV1 mediates the sorting of plasma membrane-associated components into EVs. In this study, we observed that CBM of Ago2 is necessary for Ago2 to be sorted into the EVs of cancer cells (Fig. 9D). These results suggest that by docking at the plasma membrane via Ago2/CAV1 interaction, Ago2 can be sorted with CAV1, which relocalizes to MVBs through endocytosis, into the EVs. Both Ago2 and its associated miRNAs (e.g., miR-3613-3p) are actively sorted into cancer cell– or tumor-derived EVs in an Ago2/CAV1 interaction manner (Fig. 9D–G).

Further, corresponding to the increased Ago2/CAV1 interaction in metastatic breast tumors (Fig. 8A), we observed that the levels of EV miR-3613-3p in circulation increase with tumor progression into metastasis (Fig. 9H). In human clinical samples, elevated miR-3613-3p in circulation is used as a biomarker of different diseases, including different stages of squamous cell carcinoma of the lung (Pu et al, 2016), dedifferentiated liposarcoma (Fricke et al, 2018), and Alzheimer's disease (Kumar et al, 2017; Lugli et al, 2015). Therefore, the release of specific miRNAs (e.g., miR-3613-3p) from metastatic tumors in circulation through EVs, facilitated by the increased Ago2/CAV1 interaction in metastatic tumors, can be used as a biomarker of tumor progression (Fig. 10, right panel).

We identified in the current study a unique interaction between Ago2 and CAV1, which can be regulated through lysine 212 acetylation in the CBM of Ago2, in cancer cells (Fig. 10, red arrows in right panel). Increased Ago2/CAV1 interaction with tumor progression promotes aggressive behaviors, including metastasis and EV miRNA release (Fig. 10, right panel). Because Ago2/CAV1 interaction is required, as a secondary event, in miRNA-mediated mRNA suppression (e.g., miR-3613-3p-mediated suppression of SCAI), this interaction increases the complexity of miRNA actions in cancer (Fig. 10, blue arrows in right panel). Therefore, further research is required on the effects of Ago2/CAV1 interaction on cancer therapy and the methods of selectively targeting Ago2/CAV1 interaction in clinical cancer therapy.

## Methods

### Cell culture

Human mammary epithelial cells (HMLE) were generated as described (Elenbaas et al, 2001). HMLE cells were propagated in a MEGM medium (Lonza, CA). HEK293, BEAS-2B, hTERT-HPNE, A549, H1299, MCF10A, HCC1806, LNCaP 104-S, LNCaP C4-2, LNCaP R-1, and BxPC-3 cells were obtained from American Type Culture Collection and were grown in DMEM containing 10% fetal bovine serum with penicillin-streptomycin. HMLE cells from Dr. Weinberg Lab and HCC1806, the breast cancer cell intensively used in the manuscript, were authenticated by Bioresource Collection and Research Center, Taiwan using STR profiling analysis.

For generating A549$^{Ago2-KO}$ and HCC$^{Ago2-KO}$ with CRISPR/Cas9-mediated gene targeting, A549 and HCC1806 cells were infected with the lentiviral vector LentiCRISPRv2 (Addgene). Lenti-CRISPRv2 vector was digested with BsmBI enzyme (Thermo Fisher). LentiCRISPRv2-Ago2 was generated by inserting 5' CACCGGGCGGCGCCACCATGTACTC 3' and 5' CGAGTACATGGTGGCGCCGCCCAAA 3' hybridized oligonucleotides in between BsmBI restriction sites of the LentiCRISPRv2 vector. For generating A549$^{Ago2-KO/HA-Ago2Wt}$ and A549$^{Ago2-KO/HA-Ago2\Delta}$, A549$^{Ago2-KO}$ cells were infected with the lentiviral vector pLV-EF1a-HA-Ago2-Blast and pLV-EF1a-HA-Ago2($\Delta$175–226)-Blast respectively.

For generating A549$^{Ago2-KO/HA-AID-Ago2Wt/HA-Ago2\Delta}$ and HCC1806$^{Ago2-KO/HA-AID-Ago2Wt/HA-Ago2\Delta}$, Ago2-KO cancer cells were infected with the lentiviral vector pLV-EF1a-OsTIR-IRES-Hygo and pLV-EF1a-HA-AID-Ago2-Neo. OsTIR1 (a plant-specific E3 ligase), HA-AID-Ago2Wt (auxin-inducible degron [AID]-fused HA-Ago2), HA-Ago2$\Delta$ (HA-tagged Ago2 with CAV1-interacting domain 175–226 a.a. deleted) are expressed in the Ago2-KO

cancer cells, in which the two alleles of Ago2 gene is knocked out with CRESPER/case9 system (Figs. EV3Ai, EV3Aii, Ago2-KO/HA-AID-Ago2Wt/HA-Ago2Δ). In the presence of auxin (IAA, a natural plant hormone), OsTIR1 E3 ligase increases ubiquitylation of AID-fused proteins and, in turn, causes degradation of the protein (Fig. EV3A). In Ago2-KO(HA-AID-Ago2Wt/HA-Ago2Δ) cells, the AID-fused Ago2 protein is degraded upon the treatment of IAA. In the presence of IAA, Ago2-KO(HA-AID-Ago2Wt/HA-Ago2Δ) cells only express HA-Ago2Δ (Fig. EV3Aii). In the absence of IAA, Ago2-KO/HA-AID-Ago2Wt/HA-Ago2Δ cancer cells express both full-length Ago2 (Ago2Wt) and CAV1-interacting domain deleted Ago2 (Ago2Δ) (Fig. EV3Ai).

## Plasmid constructions

Portions of Ago2 (HG11079-NY, Sino Biological) and Caveolin-1 (HG11440-CF, Sino Biological) were generated as HindIII-XbaI restriction fragments by PCR, using appropriate synthetic primers, and then cloned into the pCMV3-N-HA and pCMV3-C-FLAG (Sino Biological), respectively. The plasmid expressing Ago2-His was generated by cloning HindIII-AgeI restriction Ago2 cDNA into pcDNA4-His vectors. Mutants of Ago2 and Caveolin-1 were generated with pCMV3-HA-Ago2 and pCMV3-CAV1-Flag using KOD-Plus-Mutagenesis Kit (cat. No. SMK-101, TOYOBO) according to the manufacturer's instructions. The eGFP-Ago2(175–226)-His fusion constructs were generated by the insertion of Ago2 175–226 DNA fragment in frame with the 3′ end of the GFP gene into the pCMV3-eGFP-His plasmid.

## MiRNA mimic and inhibitor

The cells in a 50% confluent well of a six-well dish were transfected with 25 pmol miR-3613-3p mimic 5' acaaaaaaaaaaagcccaacccuuc 3', miR-6126 mimic 5' gugaaggcccggcggaga 3', or miR-3613-3p miRIDIAN™ Hairpin inhibitor (cat. No. SO-2759113G, Dharmacon™) respectively, using Lipofectamine™ 2000 (cat. No. 11668019, Invitrogen). At 24 h after transfection, the cells were seeded with P2S, P2 (10 μM), or IAA (500 μM) in a 6 cm dish for another 2 days. After 48 h, the cells were harvested for RNA analysis by qPCR.

## RNA preparation and qPCR analysis of miRNA and mRNA

Total RNA was extracted with TRIzol® reagent (Invitrogen) following the manufacturer's instructions. The extracted RNA (1.5 μg) was reversely transcribed with iScript™ Reverse Transcription Supermix (cat. No. 1708840, BioRad) for mRNA and qSTAR miRNA qPCR Detection System (cat. No. HP100042, OriGene Technologies) for miRNA, following the manufacturer's instructions. The cDNA samples and SYBR® Green PCR Master Mix (Applied Biosystems) were used for Real-time PCR experiments with LightCycler 480 instrument (Roche Diagnostics).

## miRNA microarray analysis

Total RNA was extracted with TRIzol® reagent. The RNA quality and yield were analyzed using a Bioanalyzer 2100 (Agilent) and NanoDrop® ND-1000 (NanoDrop Technologies). The RNA was then subjected to GeneChip™ miRNA 4.0 analysis. The miRNA

microarray data have been deposited in the GEO database (accession code: GSE239499, GSE239599, and GSE255941).

## Cell fractionation

In experiments of Figs. 2A, B, 3F, and 6J, the cells were subjected to Thermo Scientific Subcellular Protein Fractionation Kit (cat. No. 78840, Thermo) for separating cytosolic, total membrane, and nuclear fractions, following the manufacturer's instructions. In experiments of Fig. 2E, the cells were subjected to BioVision Plasma Membrane Protein Extraction Kit (cat. No. K268-50/ab65400, BioVision) for separating plasma membrane and cellular organelles (non-plasma membrane from the bottom phase of step B4 of the manufacturer's instructions), following the manufacturer's instructions. In experiments of Fig. 6G, the cells were subjected to Minute™ Plasma Membrane Protein Isolation and Cell Fractionation Kit (cat. No. SM-005, Inventbiotech) for isolating plasma membranes and subjected to Minute™ Endosome Isolation and Cell Fractionation Kit (cat. No. ED-028, Inventbiotech) for isolating endosomes, following the manufacturer's instructions.

## Immunoprecipitation

All research involving patient samples has been reviewed and approved by the Research Ethics Committee of the National Taiwan University Hospital. Following cell lysing for 40 min at 4 °C, the cells were centrifuged for 10 min at 16,000 ×g, at 4 °C. The supernatant, 500 μg protein in 1 ml PBS containing 0.5% Nonidet™ P-40 (Cat. No. 21-3277; Sigma-Aldrich), 5% BSA, and protease inhibitors, were pre-cleared with 20 μl Protein A-Sepharose beads (Cat. No. 20421; Thermo Fisher Scientific Inc.) for 1 h at 4 °C. The cleared supernatant was then immunoprecipitated using the antibody (1:100 dilution, refer to the table below) overnight at 4 °C. After incubating with mixing overnight, the supernatant was incubated with 20 μl of Protein A/G Agarose for 3 h at 4 °C. The beads were then washed six times with PBS containing 0.5% Nonidet™ P-40 and then eluted using a 2× SDS–PAGE loading buffer. The proteins were resolved by 12% SDS–PAGE and were analyzed with western blotting.

| Antibody | Cat. No | Source | Working dilution |
|---|---|---|---|
| HA | 3724 | Cell Signaling Technology | 1:100 |
| Ago2 | 2897 | Cell Signaling Technology | 1:100 |
| Caveolin-1 | 3267 | Cell Signaling Technology | 1:100 |
| Normal Rabbit IgG | 2729 | Cell Signaling Technology | |

## In vitro binding assay

HEK293 cells stably transfected with either Ago2-His construct or CAV1-His construct were rinsed twice with PBS and lysed in equilibration buffer, 50 mM NaH2PO4 (pH 7.0), 300 mM NaCl, 10 mM imidazole, 0.4% Nonidet P-40, 10% glycerol and mixed protease inhibitors, for 40 min at 4 °C. The His-tagged fusion protein in the supernatant was purified with Metal-affinity purification. The cleared supernatant was incubated with TALON™ metal-affinity resins (50 μ in 1 ml supernatant;

Clontech) for 1 h at 4 °C. After incubating with mixing overnight, the supernatant was incubated with 20 μl of Protein A/G Agarose for 3 h at 4 °C. The beads were then washed twice with each buffer containing 20, 30, or 40 mM imidazole, respectively, and then eluted using 300 mM imidazole. The eluate was then subjected to buffer exchange with Zeba™ spin desalt column 40MWCO (cat. No. 87766, Thermo) following the manufacturer's instructions. The purity of the fusion proteins was tested by SDS–PAGE and Coomassie staining.

The purified Ago2-His and CAV1-His proteins were mixed and incubated in 1 ml PBS containing 5% BSA and protease inhibitors for 6 h at 4 °C. The mixture was then pre-cleared with 20 μl Protein A-Sepharose beads (Cat. No. 20421; Thermo Fisher Scientific Inc.) for 1 h at 4 °C. The cleared supernatant was then immunopreci-pitated using anti-Ago2 antibodies (1:100) overnight at 4 °C. After incubating with mixing overnight, the supernatant was incubated with 20 μl of Protein A/G Agarose for 3 h at 4 °C. The beads were then washed six times with PBS containing 0.5% Nonidet™ P-40 and then eluted using 2× SDS–PAGE loading buffer. The proteins were resolved by 12% SDS–PAGE and were analyzed with western blotting.

## Preparation and analysis of detergent-resistant membranes

Detergent-resistant membranes were extracted with the Caveolae/ Rafts isolation Kit (Sigma-Aldrich) according to the manufacturer's protocol. The whole cells were lysed on ice in a lysis buffer contained 0.2%Triton X-100. After 30 min, OptiprepTM was added to bring the concentration to 35% (w/v), and the mixture was overlaid with aliquots of 30, 25, 20, and 0% (w/v) to form the OptiPrep density gradient Layers. The mixture was centrifuged at 200,000×g using Beckman SW41 rotor for 6 h. After centrifugation, the distribution of Ganglioside-GM1 was analyzed by dot-blot with cholera toxin B subunit-peroxidase (CTB-HRP) (Cat. No. C3741s; Sigma-Aldrich) detection and the proteins were analyzed by WB using specific antibodies.

## Western blotting

Total protein was extracted with RIPA lysis buffer, containing 25 mM Tris-HCl (pH 7.6), 150 mM NaCl, 1% Nonidet P-40, 1% sodium deoxycholate, 0.1% SDS, complete protease inhibitor, phosphatase inhibitor. Protein lysates were resolved on Bis-Tris gels, transferred to PVDF membranes, probed with primary antibodies overnight at 4 °C and then with HRP-linked secondary antibodies (GE Healthcare, IL) and visualized with ECL reagent (Thermo Fisher Scientific, MA). The following antibodies were used:

| Antibody | Cat. No | Source | Working dilution |
| --- | --- | --- | --- |
| HA | 3724 | Cell Signaling Technology | 1:2000 |
| Ago2 | 2897 | Cell Signaling Technology | 1:1000 |
| Caveolin-1 | 3267 | Cell Signaling Technology | 1:1000 |

| Antibody | Cat. No | Source | Working dilution |
| --- | --- | --- | --- |
| Acetylated lysine | 9441 | Cell Signaling Technology | 1:1000 |
| DDK(Flag) | GTX11043 | GeneTex | 1:1000 |
| EEA1 | 2411 | Cell Signaling Technology | 1:1000 |
| Calreticulin | GTX111627 | GeneTex | 1:1000 |
| His | A00186-100 | Genescript | 1:2000 |
| Sirt1 | 9475 | Cell Signaling Technology | 1:1000 |
| Sirt2 | GTX129154 | GeneTex | 1:1000 |
| HDAC6 | GTX100722 | GeneTex | 1:1000 |
| HDAC1 | 06-720 | Milipore | 1:1000 |
| GFP | 2956 | Cell Signaling Technology | 1:2000 |
| Fibronectin | BD610078 | BD Biosciences | 1:1000 |
| N-cadherin | 13116 | Cell Signaling Technology | 1:1000 |
| Dicer | 5362 | Cell Signaling Technology | 1:1000 |
| GW182 | SC374458 | SantaCruz | 1:500 |
| GAPDH | GTX100118 | GeneTex | 1:5000 |
| α-tubulin | TA307175 | Origene | 1:5000 |
| Integrin β1 | 303010 | Biolegend | 1:250 |
| CD81 | GTX101766 | GeneTex | 1:1000 |
| CD63 | GTX135220 | GeneTex | 1:1000 |
| Alix | GTX135282 | GeneTex | 1:1000 |
| TSG101 | GTX118736 | GeneTex | 1:1000 |
| CD9 | GTX100912 | GeneTex | 1:1000 |
| Apo-A1 | GTX12692 | GeneTex | 1:1000 |

## Tissue preparation and immunofluorescence from tissue sections

All research involving patient samples has been reviewed and approved by the Research Ethics Committee of National Taiwan University Hospital, NTUH-REC number 201910005RINC. The informed consent was obtained from all subjects. Paraformaldehyde-fixed tissues were embedded in paraffin blocks and cut into 4-μm sections. Haematoxylin and eosin (H&E) staining was conducted according to conventional procedures. Tissue sections were deparaffinized/hydrated and then were subjected to antigen retrieval in Citrate buffer (pH = 6.0) for 10 min. The sections were incubated with primary antibodies overnight at 4 °C and then with secondary antibodies (Invitrogen, CA) for 1 h at room temperature. Cell nuclei were visualized with Hoechst 33258 (cat. No. 14530, Merck). Slides were mounted with ProLong® Gold Antifade Reagent (cat. No. P10144, Thermo) and imaged using a TCS SP5 II confocal microscope (Leica, Germany). MetaMorph 5.0 software (Molecular Devices) was used for image quantification. The levels of total Ago2 and Ago2, which

overlapped with cytokeratin 8/18 were quantified. The colocalization was presented as the percentage of Ago2 overlapping with cytokeratin 8/18. In the experiments of Fig. EV4, the tissue arrays, including Breast disease spectrum tissue array BR2082c and Lung disease spectrum tissue array LC2083 were obtained from US Biomax.

The following antibodies were used:

| Antibody | Cat. No | Source | Working dilution |
|---|---|---|---|
| Ago2 | H00027161-M01 | Abnova | 3 ug/ml |
| Cytokeratin 8/18 | M3652 | Agilent | 1:100 |
| HA-tag | 3724 | Cell Signaling Technology | 1:100 |
| Anti-Mouse Alexa Fluor™ 488 | A21202 | Invitrogen, Thermo | 1:400 |
| Anti-Mouse Alexa Fluor™ 647 | A21235 | Invitrogen, Thermo | 1:400 |
| Anti-Rabbit Alexa Fluor™ 591 | A21207 | Invitrogen, Thermo | 1:400 |
| Anti-Rabbit Alexa Fluor™ 647 | A21244 | Invitrogen, Thermo | 1:400 |

| Antibody | Cat. No | Source | Working dilution |
|---|---|---|---|
| EEA1 | 2411 | Cell Signaling Technology | 1:250 |
| Caveolin-1 | 3267 | Cell Signaling Technology | 1:100 |
| Calnexin | 2679 | Cell Signaling Technology | 1:250 |
| Anti-Mouse Alexa Fluor™ 488 | A21202 | Invitrogen, Thermo | 1:400 |
| Anti-Mouse Alexa Fluor™ 647 | A21235 | Invitrogen, Thermo | 1:400 |
| Anti-Rabbit Alexa Fluor™ 594 | A21207 | Invitrogen, Thermo | 1:400 |
| Anti-Rabbit Alexa Fluor™ 647 | A21244 | Invitrogen, Thermo | 1:400 |
| Anti-Rat Alexa Fluor™ 488 | A11006 | Invitrogen, Thermo | 1:400 |
| Anti-Goat Alexa Fluor™488 | A11055 | Invitrogen, Thermo | 1:500 |
| Cholera Toxin Subunit B-647 | C34778 | Invitrogen, Thermo | 1:100 |
| Cholera Toxin Subunit B-594 | C34777 | Invitrogen, Thermo | 1:100 |

## Immunofluorescence studies on cells

Cells on coverslips were fixed in 4% PFA solution in PBS for 15 min at room temperature, then washed with PBS. Cells were permeabilized with 0.3% Triton X-100, then blocked with 5% normal donkey serum in PBS at room temperature for 60 min. The coverslips were incubated with primary antibodies overnight at 4 °C, then with secondary antibodies (Invitrogen, CA) for 1 h at room temperature. Cell nuclei were visualized with Hoechst 33258 (cat. No. 14530, Merck). Slides were mounted with ProLong® Gold Antifade Reagent (No. P10144, Thermo) and imaged using a TCS SP5 II confocal microscope (Leica, Germany). FIJI (Image J2) with MorphoLibJ (Filtering; Morphological filters) plugin was used for image quantification as described (Shihan et al, 2021). For Ago2 association with CAV1, EEA1, and CANX (Fig. 2D), the levels of total Ago2 and Ago2 which overlapped with CAV1, EEA1, or CANX were quantified. The association was presented as the percentage of Ago2 overlapping with CAV1, EEA1, or CANX. For quantitation of Ago2 that overlapped with CAV1 on the plasma membranes (Fig. 2G, I), the cell edge was selected with freehand selections to set the cell membrane area as a Mask. CAV1 in the cell membrane Mask was set as mCAV1. The levels of mCAV1 and Ago2, which overlapped with mCAV1, were quantified. The following antibodies were used:

| Antibody | Cat. No | Source | Working dilution |
|---|---|---|---|
| Ago2 | H00027161-M01 | Abnova | 3 ug/ml |
| Ago2 | SAB4200085 | Sigma-Aldrich | 1:250 |
| HA-tag | NB600-362 | Novus | 1:250 |

## Cell migration and invasion assays

A549 and HCC1806 cells were pretreated with (1) P2S or P2 (10 µM) and (2) PBS or IAA (500 µM) for 2 days. For migration assay, the treated A549 cells ($5 \times 10^4$ cells/well) or HCC1806 cells ($8 \times 10^5$ cells/well) were added to the upper chambers of the non-coated transwell chamber, 8.0 µm (Falcon cell culture insert, cat. No. 353097; BD Biosciences). For invasion assay, the treated A549 cells ($1 \times 10^5$ cells/well) or HCC1806 cells ($5 \times 10^5$ cells/well) were added to the upper chambers of Biocoat Matrigel Chambers, 8.0 µm (cat. No. 40480. BD Biosciences) After 24 h, residual cells were removed from the top of the membrane and the cells on the underside of the membrane were washed in PBS, fixed in 4% PFA, then stained with 0.1% crystal violet for 20 min. The cells that migrated through the membrane during the incubation period were counted in nine randomly selected regions.

## Tumorsphere formation

Tumorsphere-forming analysis was performed as described by Dontu et al (Dontu et al, 2003). A549 and HCC1806 cells were pretreated with (1) P2S or P2 (10 µM) and (2) PBS or IAA (500 µM) for 2 days, and then were dissociated into single-cell suspensions and plated in 96 wells of ultra-low attachment plates (cat. No. 7007, Corning Life Sciences) at 300 to 500 (A549 serials) and 1000–2000 (HCC1806 serials) cells/well respectively in MammoCult™ medium (cat. No. 05621, STEMCELL Technologies) supplemented with MammoCult™ proliferation supplements (cat. No. 05622, STEMCELL Technologies), 4 µg/mL heparin (cat. No. 07980, STEMCELL Technologies), 0.48 µg/mL hydrocortisone (cat. No. H0888, Merck), 100 U/mL penicillin, 100 µg/mL streptomycin,

and 1% methylcellulose (cat. No.M7027, Merck) After 2 weeks, the number of spheres (diameter >100 μm) of each well was calculated.

## Anoikis assay (suspension-induced apoptosis-resistant assay)

A549 cells that had been treated with P2S or P2 (10 μM) for 48 h and A549$^{Ago2-KO/HA-AID-Ago2Wt/HA-Ago2Δ}$ that had been treated with PBS or IAA (500 μM) were detached. A single-cell suspension in DMEM medium containing 0.5% methylcellulose was seeded into an ultra-low attachment plate (cat. No. 7007, Corning Life Sciences) at a density of $1.5 \times 10^4$ cells/ml. The cells were then harvested and counted at 48 h after the seeding.

## Cell viability assay

A549 cells were pretreated with P2S or P2 for 48 h and then were seeded in 96-well plates ($2.5 \times 10^3$ cells/well) in the presence of PBS, Paclitaxel (0.25 and 5 μM) or Gefitinib (1 and 2 μM) for 48 h. After drug treatment, 5 mg/mL MTT (3-[4,5-dimethylthiazol-2-yl]-2,5-diphenyl tetrazolium bromide, cat. No. M5655, Merck) was added to each well at 1/10 of the medium volume, followed by a 2 h incubation at 37 °C. The formazan crystals were dissolved using 200 μl of DMSO, and absorbance was measured at 570 nm was recorded, using EnSpire® Multimode Plate Readers (PerkinElmer).

## Animal experiments

All research involving animals complied with protocols approved by the NHRI Committee on Animal Care, approval number NHRI-IACUC-109007. Six-week-old female NOD.CB17-Prkdcscid/JNarl mice were obtained from the National Laboratory Animal Center (NLAC, Taiwan). The number of cells, the route of injection and the time for tumor growth are indicated in the figure legends.

## Measurement of tumor growth

Tumor growth was monitored using IVIS optical imaging. To image tumor burden, D-luciferin (100 μl; 125 mg/kg) was injected via tail vein, and mice were scanned for 1 min using the Caliper IVIS Spectrum Optical Imaging System to monitor tumor luciferase activity. Bioluminescent images were analyzed using Living Image software (PerkinElmer, MA). Regions of interest (ROI) were drawn over tumors, and total photons were calculated.

## Tumor cell detection using qPCR

Mouse tissues, with or without tumors, were homogenized, and genomic DNA was purified, using the DNeasy tissue kit (Qiagen). Human glyceraldehyde-3-phosphate dehydrogenase (GAPDH) DNA and mouse GAPDH DNA were measured by qPCR. The human-specific GAPDH primer pair is 5′ primer ACGCTTTC TTTCCTTTCGCGCTCTGCGGGG/3′ primer CTAACGGCTGC CCATTCATTTCCTTCCCGG, which can amplify a DNA region in a human GAPDH intron. The mouse-specific GAPDH primer pair is 5′ primer CTGCGGAAATGGTGTGATCTTCCCCAAGGG/ 3′ primer AGGGAGCTCCATTCATGTGCTAAACAGGCC, which can amplify a DNA region in a murine GAPDH intron. The human-to-mouse DNA ratios were expressed by the comparative Ct method [A/B

DNA ratio = $2^{-(Ct\ of\ A\ -\ Ct\ of\ B)}$]. The standard curve to determine tumor burden was determined by qPCR of DNAs extracted from mixtures of mouse tissue homogenate and cancer cells. Tumor cells per 25 mg tissue were determined by comparison of PCR data from extracts of tumor-bearing livers with this standard curve. Since the metastatic tumor cells did not uniformly distribute in the lungs, in Fig. 5Fiv, two portions of tissue in different lobes of each mouse lung were taken for qPCR respectively.

## EV isolation

EVs were isolated from cell culture media and from plasma by differential ultracentrifugation as previously described (Lin et al, 2017a). For preparing cell culture media, EVs in FBS were depleted using ultracentrifugation and filtration as described (Eitan et al, 2015; Shelke et al, 2014). FBS was centrifuged at 120,000×g for 8 h at 4 °C. The supernatant was diluted with DMEM at a 1:9 ratio and then filtered through a 0.22 μm filter before the DMEM containing 10% EV-depleted FBS was used for cell culture. The depletion of FBS EVs was confirmed using nanoparticle tracking analysis (NTA) and western blot analysis.

After cell culture, the culture media were centrifuged at 300 g for 5 min to remove cells (P1 fraction), at 2000×g for 20 min (P2 fraction), then at 10,000×g for 30 min (P3 fraction), all at 4 °C. Finally, EVs (P4 fraction) were separated from the supernatant by centrifugation at 110,000×g for 60 min. The EV pellet was resuspended in PBS and centrifuged at 110,000×g for 60 min again. The final EV pellet was resuspended in PBS for further analysis.

For collecting patient plasma, all research involving patient samples has been reviewed and approved by the Research Ethics Committee of National Taiwan University Hospital, NTUH-REC number 201910005RINC. Informed consent was obtained from all subjects. About 1 ml of whole blood from patients or mice was collected into the EDTA-treated tube. Blood cells were removed from the plasma by centrifugation for 15 min at 2500×g, 4 °C, and the supernatant was centrifuged again at 2500×g, 4 °C for 15 min. About 250 ul supernatant was diluted with PBS to 1 ml and then was centrifuged at 10,000×g, 4 °C for 30 min using Rotor tla-120.2. Finally, EVs were separated from the supernatant by centrifugation at 110,000×g for 120 min. The EV pellet was resuspended in PBS and centrifuged at 110,000×g for 120 min again. The final EV pellet was resuspended in PBS for further analysis.

## EV purification using a density gradient

EV pellet from ultracentrifugation was resuspended with 40% (w/v) iodixanol in 2 ml PBS. The mixture was overlaid in sequence with aliquots of 30, 20, 10, and 5% (w/v) iodixanol in PBS (2 mL each) to form a density gradient in an ultracentrifuge tube. The mixture was then centrifuged at 200,000×g for 8 h at 4 °C. After the centrifugation, each gradient fraction (10 fractions; 1 mL/fraction) was collected with a pipette from the top of the tube. The presence of Ago2, CAV1, exosome markers (e.g., CD81, CD9, CD63, and Tsg101), and housekeeping protein GAPDH in each fraction was analyzed by SDS–PAGE and Western blot.

## Nanoparticle tracking analysis (NTA)

The concentration and size of EVs were measured by nanoparticle tracking analysis (NTA) using a NS300 system (NanoSight)

(Dragovic et al, 2011). The CM-P3 and CM-P4 fractions were suspended in PBS at 20 μg total protein/ml and then were diluted 10,000- fold for NTA analysis.

## Proteinase K protection assay

EV suspensions were pre-incubated with RIPA buffer, containing 25 mM Tris-HCl (pH 7.6), 150 mM NaCl, 1% Nonidet P-40, 1% sodium deoxycholate, 0.1% SDS, at 37 °C for 15 min. After incubation, these EV suspensions were treated with 0.25 ug/ml proteinase K at room temperature for 15 min. These samples were dissolved into an SDS loading buffer for SDS–PAGE and western blotting.

## Statistical analysis

Data were presented as mean ± SEM. A Mann–Whitney $U$-test, an alternative to a $t$-test when the data are not normally distributed (Hart, 2001), was applied to explore differences between different groups of mice. A paired, two-sided Student's $t$-test was used to compare two groups of the experiments in Fig. 8B, E. Otherwise, a Student's $t$-test (unpaired, two-sided) was used to compare two groups of the experiments as indicated in the legends.

# Data availability

The miRNA microarray data have been deposited in the GEO database (accession code: GSE239499, GSE239599, and GSE255941). All data are available in the main text or the appendix.

# Peer review information

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

## Acknowledgements

We thank all the Li lab members for their advice and support. We would like to recognize the Microarray Core Lab, Optical Biology Core Lab, Pathology Core Lab, and Laboratory Animal Center at the National Health Research Institutes (NHRI) for help with experiments. We also acknowledge Dr. Shaw Fang Yet of NHRI and Dr. Wei-Yi Chen of National Yang Ming Chiao Tung University for reviewing the articles. This work has been supported by National Science and Technology Council (NSTC) grants: 109-2326-B-400-004-MY3 (grant of Ta-You Wu Memorial Award), 103-2320-B-400-015-MY3, by NSTC postdoctoral grants: 109-2811-B-400-528, 110-2811-B-400-523, 111-2811-B-400-022, by NHRI grants: CS-104-PP-16, CS-105-PP-16, and by NHRI Young Scientist Award: CS-111-SP-02.

## Author contributions

**Meng-Chieh Lin:** Conceptualization; Data curation; Formal analysis; Validation; Investigation; Visualization; Methodology; Writing—original draft. **Wen-Hung Kuo:** Resources; Formal analysis; Validation; Investigation; Methodology. **Shih-Yin Chen:** Conceptualization; Data curation; Formal analysis; Validation; Investigation; Visualization; Methodology. **Jing-Ya Hsu:** Data curation; Formal analysis; Validation; Investigation; Visualization; Methodology. **Li-Yu Lu:** Data curation; Formal analysis; Validation; Investigation; Visualization; Methodology. **Chen-Chi Wang:** Resources; Data curation; Formal analysis; Investigation; Methodology. **Yi-Ju Chen:** Data curation; Formal analysis; Validation; Investigation; Methodology. **Jia-Shiuan Tsai:** Data curation; Formal analysis; Validation; Investigation. **Hua-Jung Li:** Conceptualization; Resources; Data curation; Formal analysis; Supervision; Funding acquisition; Validation; Investigation; Visualization; Methodology; Writing—original draft; Project administration; Writing—review and editing.

## Disclosure and competing interests statement

MCL has finished his postdoctoral training at NHRI, and he is currently an employee of Anbogen Therapeutics.

# Expanded View Figures

**Figure EV1. Mapping the interacting domains of Ago2 and CAV1.**                                                                         ▶

(A) The quantitation of Western blots of experiments in Fig. 1A. Each spot describes the ratio of coprecipitated CAV1 (pCAV1) to precipitated Ago2 (pAgo2) in the sample, relative to that of the cancer cell sample in each replicated experiment. Bars are means ± standard error of the mean (SEM, $n = 3$–6, biological replicates). Student's $t$-test, $^{**}P \leq 0.01$; $^{****}P \leq 0.0001$. (B) Direct interaction of Ago2 with CAV1 in vitro. In the mixture of Ago2-His and CAV1-His, Ago2-His was immunoprecipitated with anti-Ago2 antibodies, and coprecipitation of CAV1 was analyzed. The mixtures contain CAV1-His only, or IgG serves as negative controls. (C) The quantitation of Western blots of experiments in Fig. 1B. Each spot describes the ratio of coprecipitated CAV1 (pCAV1) to input CAV1 (inCAV1) in the sample, relative to that of the mixture of Ago2-His and CAV1-His (+Ago2-His) in each replicated experiment. Bars are means ± SEM ($n = 3$, biological replicates). Student's $t$-test, $^{****}P \leq 0.0001$. (D) The quantitation of Western blots of experiments in Fig. 1C. Each spot describes the ratio of coprecipitated CAV1 (pCAV1) to precipitated Ago2 (pAgo2), relative to that of the mixture with full-length HA-Ago2 (Full) in each replicated experiment. Bars are means ± SEM ($n = 4$, biological replicates). Student's $t$-test, $^{**}P \leq 0.01$; $^{***}P \leq 0.001$; $^{****}P \leq 0.0001$. (E) The quantitation of Western blots of experiments in Fig. 1D. Each spot describes the ratio of coprecipitated CAV1 (pCAV1) to precipitated Ago2 (pAgo2), relative to that of the mixture with His-CAV1(1-102)-mRuby. $n = 1$. (F) The quantitation of Western blots of experiments in Fig. 1E. Each spot describes the ratio of coprecipitated CAV1 (pCAV1) to precipitated Ago2 (pAgo2), relative to that of the mixture with wild-type CAV1-Flag (Wt) in each replicated experiment. Bars are means ± SEM ($n = 3$, biological replicates). Student's $t$-test, $^{****}P \leq 0.0001$. (G) Schematic of human Ago2 deletion constructs. The full-length (Full) construct contains four characteristic domains, namely N-terminal (N), PAZ, MID, and C-terminal PIWI domains, where PAZ is an RNA-binding module and MID and PIWI are catalytic activity domains. (H) A series of HA-tagged human Ago2 (HA-Ago2), including C-terminal-deleted, N-terminal-deleted, and serine 387-substituted Ago2, coexpressed with Flag-labeled human CAV1 (CAV1-Flag) in HEK293 cells. Panel i: HA-Ago2 was immunoprecipitated with anti-HA antibodies, and coprecipitation of CAV1-Flag was analyzed using anti-Flag antibodies. Panel ii: the quantitation of Western blots of experiments in panel i. Each spot describes the ratio of coprecipitated CAV1 (pCAV1) to precipitated Ago2 (pAgo2), relative to that of the mixture with full-length HA-Ago2 (Full) in each replicated experiment. Bars are means ± SEM ($n = 2$). (I) The quantitation of Western blots of experiments in Fig. 1F. Each spot describes the ratio of coprecipitated CAV1 (pCAV1) to precipitated Ago2 (pAgo2), relative to that of the mixture with full-length HA-Ago2 (Full) in each replicated experiment. Bars are means ± SEM ($n = 3$, biological replicates). Student's $t$-test, $^{*}P \leq 0.05$. (J) The quantitation of Western blots of experiments in Fig. 1G. Each spot describes the ratio of coprecipitated CAV1 (pCAV1) to precipitated Ago2 (pAgo2), relative to that of the mixture with HA-Ago2(1–226) in each replicated experiment. Bars are means ± SEM ($n = 2$). (K) The quantitation of Western blots of experiments in Fig. 1H. Each spot describes the ratio of coprecipitated CAV1 (pCAV1) to precipitated Ago2 (pAgo2), relative to that of the mixture with wild-type HA-Ago2 (Wt) in each replicated experiment. Bars are means ± SEM ($n = 6$, biological replicates). Student's $t$-test, $^{****}P \leq 0.0001$. (L) The quantitation of Western blots of experiments in Fig. 1I. Each spot describes the ratio of coprecipitated CAV1 (pCAV1) to precipitated His-tagged eGFP (peGFP), relative to that of the mixture with eGFP-Ago2(175–226)-His in each replicated experiment. Bars are means ± SEM ($n = 3$, biological replicates). Student's $t$-test, $^{***}P \leq 0.001$. (M) Aromatic amino acids in Ago2 are displayed using Pepinfo. The black bars indicate the positions of aromatic residues among a total of 859 residues in Ago2. The red box indicates the CBM of Ago2 amino acids 199–212. (N) Changes in the protein thermodynamic stability ($\Delta\Delta G$) of the indicated amino acid mutations, estimated using the PoPMuSiC algorithm. Aromatic amino acids are marked in red, and lysine is marked in blue. SER: surface exposed ratio. SERs exceeding 45% are indicated by a yellow background, suggesting an exposed amino acid. SERs below 20% are indicated by a gray background, suggesting a buried amino acid. Mutation-inducing protein thermodynamic instability is indicated by a red background. (O) Effects of Ago2 CBM mutation on Ago2/CAV1 interaction. In HEK293 cells, wild-type (Wt) or mutated (WF199/200AA, F202A, W211A) HA-Ago2 was coexpressed with CAV1-Flag. Panel i: in HEK293 cells, HA-Ago2 was immunoprecipitated with anti-HA antibodies, and coprecipitation of CAV1-Flag was analyzed using anti-Flag antibodies. The mixture contains HA-Ago2 serves as a negative control. Panel ii: the quantitation of Western blots of experiments in panel i. Each spot describes the ratio of coprecipitated CAV1 (pCAV1) to precipitated Ago2 (pAgo2), relative to that of the mixture with wild-type HA-Ago2 (Wt) in each replicated experiment. Bars are means ± SEM ($n = 3$, biological replicates). Student's $t$-test, $^{*}P \leq 0.05$; $^{**}P \leq 0.01$. (P) The quantitation of Western blots of experiments in Fig. 1K. Each spot describes the ratio of coprecipitated CAV1 (pCAV1) to input CAV1 (inCAV1) in the sample, relative to that of the mixture with wild-type HA-Ago2 (Wt) in each replicated experiment. Bars are means ± SEM ($n = 3$, biological replicates). Student's $t$-test, $^{****}P \leq 0.0001$.

   

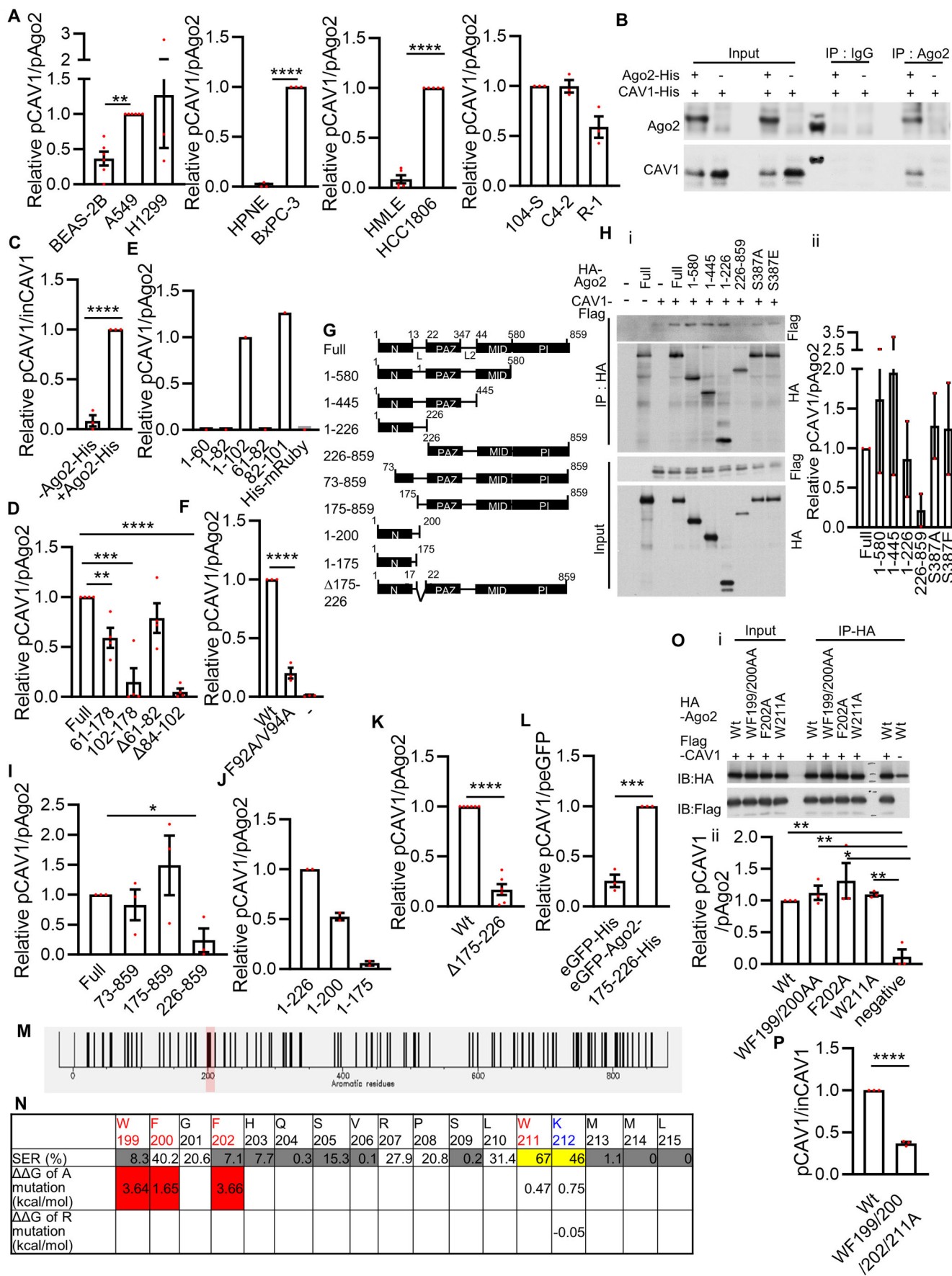

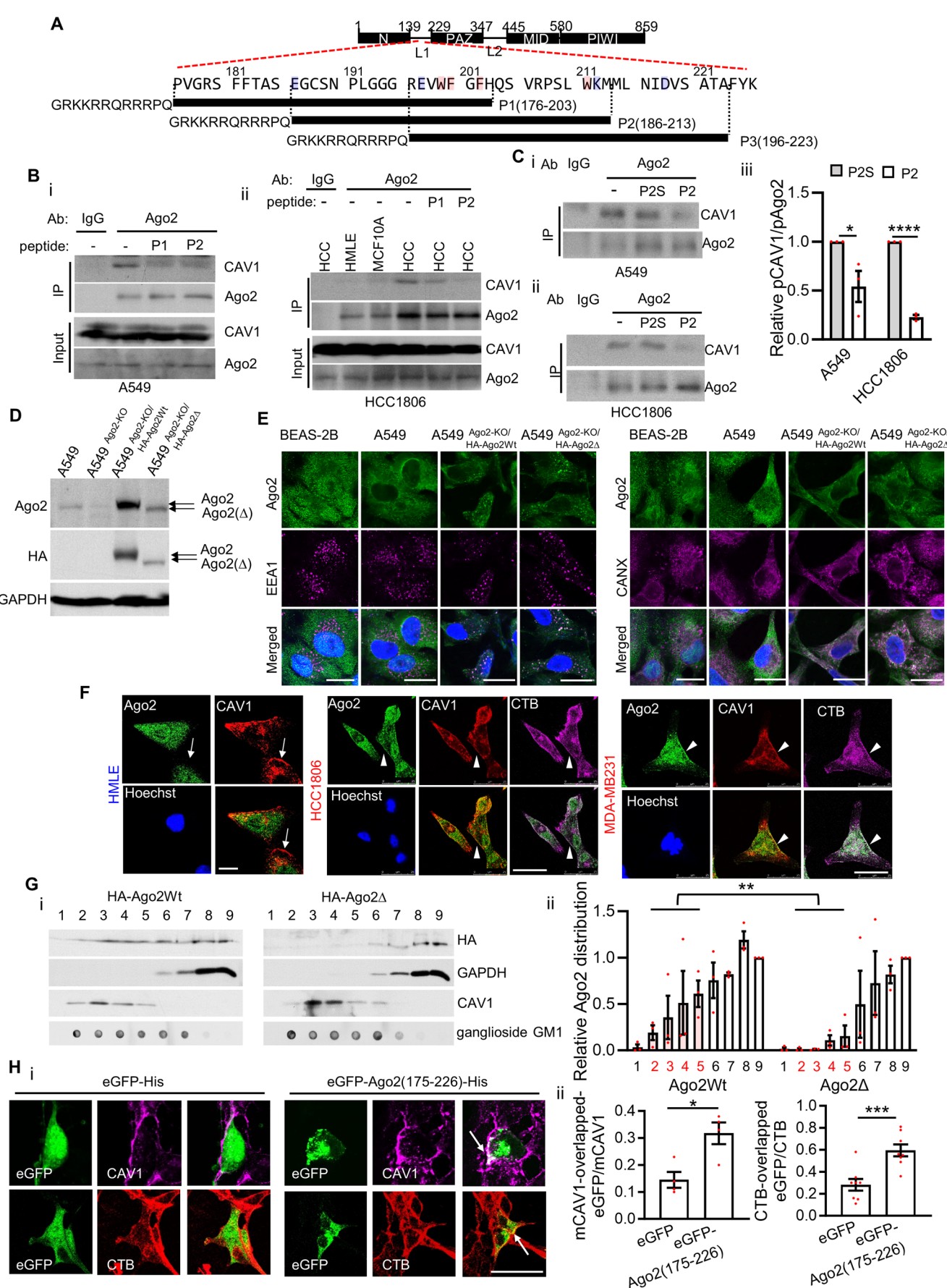

◀ **Figure EV2. The CBM of Ago2 affects plasma membrane-associated distribution of Ago2.**

(A) Amino acid sequences of blocking peptides P1, P2, and P3, which are designed in accordance with the amino acid sequence 175–226 of Ago2. Cell-penetrating sequence GRKKRRQRRRPQ was added to the blocking peptides. The red box indicates an aromatic amino acid, and the blue box indicates a charged amino acid. (B) Disruption of Ago2/CAV1 interaction with blocking peptides P1 and P2 was analyzed using co-immunoprecipitation. In A549 (panel i) and HCC1806 (panel ii) cancer cells treated with P1 and P2 peptides, respectively, Ago2 was immunoprecipitated with anti-Ago2 antibodies, and coprecipitation of CAV1 was analyzed. Normal IgG was used as a negative control for precipitation, and HMLE and MCF10A cells were used as negative controls for the interaction. (C) Disruption of Ago2/CAV1 interaction by peptides. In A549 (panel i) and HCC1806 (panel ii) cancer cells treated with P2 and P2S peptides, respectively, Ago2 was immunoprecipitated with anti-Ago2 antibodies, and coprecipitation of CAV1 was analyzed. Panel iii: The quantitation of Western blots of experiments in panel i and ii. Each spot describes the ratio of coprecipitated CAV1 (pCAV1) to precipitated Ago2 (pAgo2), relative to that of the sample treated with P2S in each replicated experiment. Bars are means ± SEM ($n = 3$, biological replicates). Student's $t$-test, $^{*}P \leq 0.05$; $^{****}P \leq 0.0001$. (D) Expression of Ago2 in CRISPER/Cas9 gene-edited A549 cancer cells. A549$^{Ago2-KO}$: A549 cells with Ago2 knocked out by CRISPER/Cas9 gene editing. A549$^{Ago2-KO/HA-Ago2Wt}$: Ago2-knockout A549 cells expressing HA-tagged wild-type Ago2. A549$^{Ago2-KO/HA-Ago2Δ}$: Ago2-knockout A549 cells expressing HA-tagged CBM-deleted (amino acids 175–226) Ago2. The expression of Ago2 was analyzed using anti-Ago2 and anti-HA antibodies, with GAPDH used as a loading control. (E) Distribution of Ago2 (green), endosome marker EEA1 (purple, left panel), and ER marker CANX (purple, right panel) analyzed using immunofluorescence in normal epithelial cells BEAS-2B and cancer cells A549, A549$^{Ago2-KO/HA-Ago2Wt}$ and A549$^{Ago2-KO/HA-Ago2Δ}$. Cell nuclei were stained with Hoechst (blue). Scale bar = 20 μm. (F) Distribution of Ago2 (green) and CAV1 (red) was analyzed using immunofluorescence in normal epithelial cells and cancer cells. Lipid rafts were stained with CTB (purple), and cell nuclei were stained with Hoechst (blue). The arrowheads indicate the colocalization of Ago2 and CAV1, and the arrows indicate CAV1 without colocalization with Ago2. Scale bar = 25 μm. (G) Western blot analysis of HA-Ago2(Wt), HA-Ago2(Δ175–226), CAV1, and GAPDH in cell membrane fractions enriched in lipid rafts (LRF, fractions 2–5) and non-lipid rafts (non-LRF, fractions 7–9). HA-Ago2(Wt) and HA-Ago2(Δ175–226) were expressed in HEK293 cells. Panel i: proteins were detected in these membrane fractions through Western blotting, with ganglioside-GM1 (CTB dot blotting) and CAV1 being markers of lipid rafts and GAPDH being a marker of non-lipid rafts. Panel ii: The quantitation of Western blots of experiments in panel i. Each spot describes the level of Ago2 in the fraction, relative to that of the 9th fraction in each replicated experiment. Bars are means ± SEM ($n = 3$, biological replicates). Student's $t$-test, $^{**}P \leq 0.01$. (H) Distribution of eGFP-His, eGFP-Ago2(175–226)-His (green), and CAV1 (purple) in HEK293 cells analyzed using immunofluorescence in panel i. Lipid rafts were stained with CTB (red). The arrows indicate protein colocalization. Scale bar = 25 μm. Panel ii: the quantification of eGFP-His/eGFP-Ago2(175–226)-His association with CAV1 (left) and CTB (right) on the plasma membrane of normal epithelial cells and cancer cells in panel i. Each spot indicates the level of Ago2 overlapping with cell surface CAV1/CTB, normalized with total mCAV1/CTB, respectively. Each condition was quantified from 2 and 4 fields per replicate, 2 replicates per experiment. Bars are means ± SEM ($n = 4$ and 8). Student's $t$-test, $^{*}P \leq 0.05$, $^{***}P \leq 0.001$.

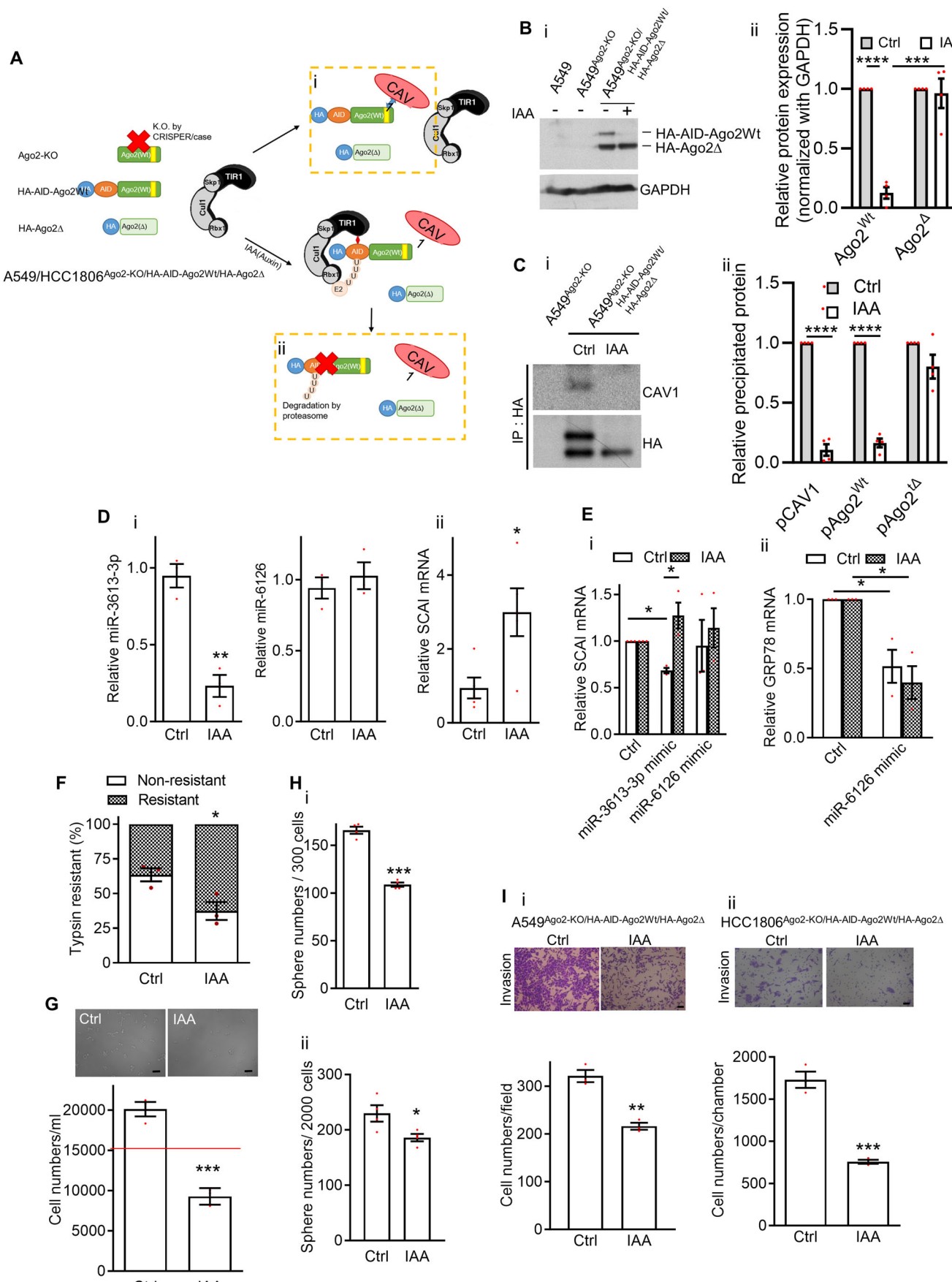

◀ **Figure EV3. Cancer cells with inducible disruption of Ago2/CAV1 interaction.**

(A) Blockage of Ago2/CAV1 interaction with AID (see "Materials and Methods"). In A549$^{Ago2-KO/HA-AID-Ago2Wt/HA-Ago2\Delta}$ and HCC1806$^{Ago2-KO/HA-AID-Ago2Wt/HA-Ago2\Delta}$ cells, OsTIR1 E3 ligase increased the ubiquitylation of HA-AID-Ago2Wt in the presence of IAA, which in turn resulted in protein degradation. IAA-treated cells expressed only HA-Ago2$\Delta$, which does not bind to CAV1. Ago2/CAV1 interaction was blocked in IAA-treated cancer cells with an auxin-inducible degron (AID) system. (B) Expression of HA-Ago2 and HA-Ago2$\Delta$ of A549$^{Ago2-KO/HA-AID-Ago2Wt/HA-Ago2\Delta}$ under auxin-inducible degradation. A549$^{Ago2-KO}$: A549 cells with Ago2 knocked out by CRISPER/Cas9 gene editing. A549$^{Ago2-KO/HA-AID-Ago2Wt/HA-Ago2\Delta}$: Ago2-knockout A549 cells expressing HA-AID-tagged wild-type Ago2 and HA-tagged CBM-deleted (amino acids 175–226) Ago2. Panel i: The expression of Ago2 was analyzed using anti-HA antibodies, with GAPDH used as a loading control. Panel ii: The quantitation of Western blots of experiments in panel i. Each spot describes the level of Ago2 in the sample, relative to that of vehicle (Ctrl)-treated A549$^{Ago2-KO/HA-AID-Ago2Wt/HA-Ago2\Delta}$ cells in each replicated experiment. Bars are means ± SEM ($n = 4$, biological replicates). Student's $t$-test, $^{***}P \leq 0.001$, $^{****}P \leq 0.0001$ (C) Panel: In vehicle (Ctrl)- and IAA-treated A549$^{Ago2-KO/HA-AID-Ago2Wt/HA-Ago2\Delta}$ cells, HA-AID-Ago2Wt and HA-Ago2$\Delta$ proteins were immunoprecipitated with anti-HA antibodies, and coprecipitation of CAV1 was analyzed. A549$^{Ago2-KO}$ was used as a negative control. Panel ii: The quantitation of Western blots of experiments in panel i. Each spot describes the level of the precipitated protein, relative to that of vehicle (Ctrl)-treated A549$^{Ago2-KO/HA-AID-Ago2Wt/HA-Ago2\Delta}$ cells in each replicated experiment. Bars are means ± SEM ($n = 4$, biological replicates). Student's $t$-test, $^{****}P \leq 0.0001$. (D) Expression of miR-3613-3p, miR-6126 (panel i), and SCAI mRNAs (panel ii) in PBS- and IAA-treated A549$^{Ago2-KO/HA-AID-Ago2Wt/HA-Ago2\Delta}$ cells. Each spot describes the relative miRNA or mRNA expression level of the sample to that of the PBS-treated sample. miRNA data normalized to U6 snRNA and mRNA data normalized to GAPDH mRNA. Bars are means ± SEM ($n = 3$ and 4, biological replicates). Student's $t$-test, $^{*}P \leq 0.05$; $^{**}P \leq 0.01$. (E) Expression of SCAI (left panel) and GRP78 (right panel) mRNAs in PBS- and IAA-treated A549$^{Ago2-KO/HA-AID-Ago2Wt/HA-Ago2\Delta}$ cells with miR-3613-3p or miR-6126 mimics. Each spot describes the relative mRNA expression level of the sample to that of the A549 cells without miRNA mimics (Ctrl). Bars are means ± SEM ($n = 3$, biological replicates). Student's $t$-test, $^{*}P \leq 0.05$. (F) Blockage of Ago2/CAV1 interaction increased the resistance of cancer cells to trypsinization. A549$^{Ago2-KO/HA-AID-Ago2Wt/HA-Ago2\Delta}$ cells were treated with vehicle or IAA. Treated cells were evaluated in terms of resistance to 0.05% trypsin. Bars are means ± SEM ($n = 3$, biological replicates). Student's $t$-test, $^{*}P \leq 0.05$. (G) Blockage of Ago2/CAV1 interaction decreased the resistance of cancer cells to anoikis. The numbers of IAA/PBS-treated A549$^{Ago2-KO/HA-AID-Ago2Wt/HA-Ago2\Delta}$ cells in suspension were evaluated in terms of resistance to anoikis. The red line indicates the initial cell number on day 0. Bars are means ± SEM ($n = 3$, biological replicates). Student's $t$-test, $^{***}P \leq 0.001$. Scale bar $= 50$ μm. (H) Blockage of Ago2/CAV1 interaction decreased tumorsphere formation. A549$^{Ago2-KO/HA-AID-Ago2Wt/HA-Ago2\Delta}$ (panel i) and HCC1806$^{Ago2-KO/HA-AID-Ago2Wt/HA-Ago2\Delta}$ (panel ii) cells were treated with vehicle or IAA. The cells were subjected to tumorsphere assays, and treatment was suspended during the assays. Bars are means ± SEM ($n = 4$, biological replicates). Student's $t$-test, $^{*}P \leq 0.05$; $^{***}P \leq 0.001$. (I) Blockage of Ago2/CAV1 interaction decreased the invasion and migration of cancer cells. A549$^{Ago2-KO/HA-AID-Ago2Wt/HA-Ago2\Delta}$ (panel i) and HCC1806$^{Ago2-KO/HA-AID-Ago2Wt/HA-Ago2\Delta}$ (panel ii) cells were treated with vehicle or IAA. The cells were subjected to invasion assays, and treatment was suspended during the assays. Bars are means ± SEM ($n = 3$, biological replicates). Student's $t$-test, $^{**}P \leq 0.01$, $^{***}P \leq 0.001$. Scale bar $= 100$ μm.

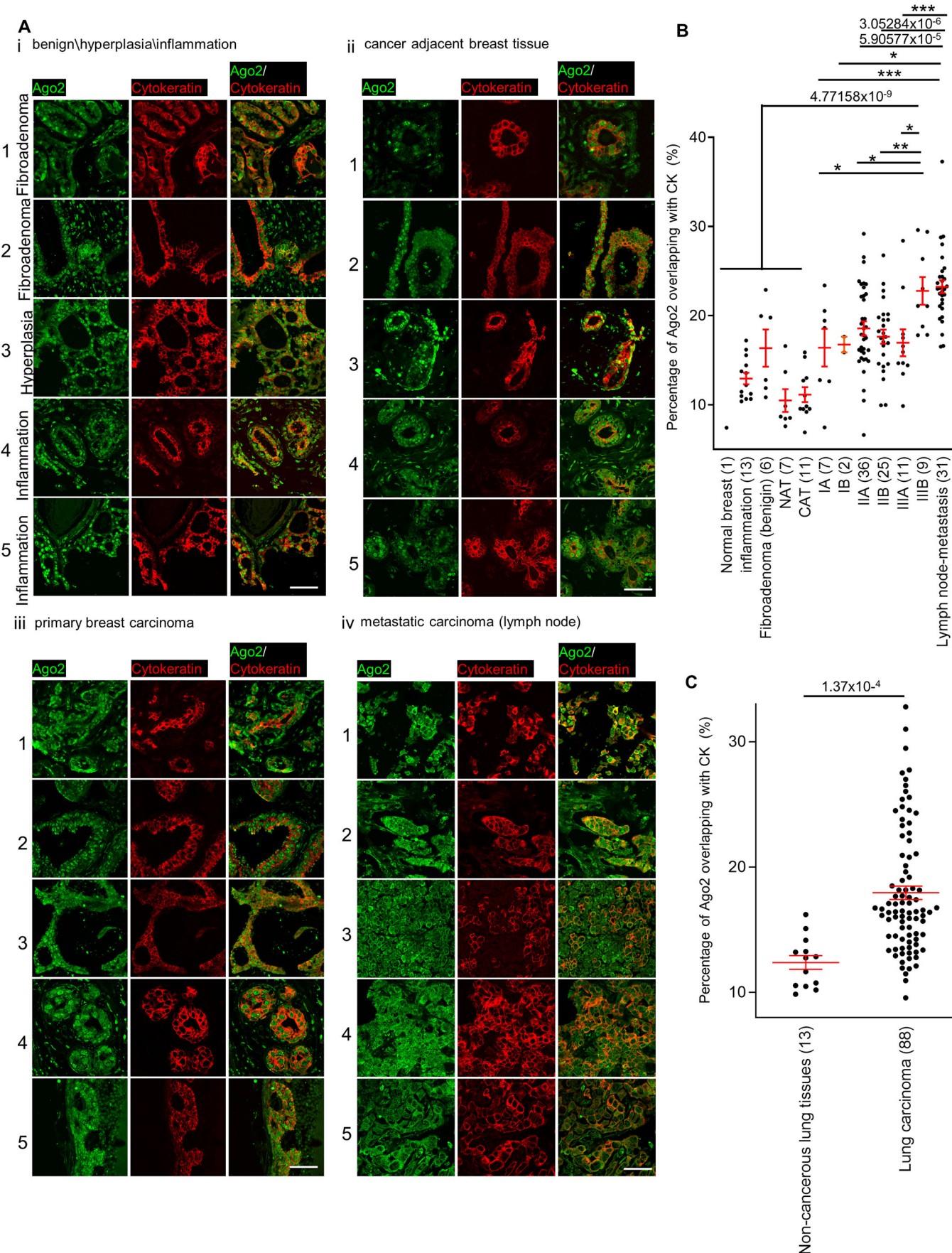

◄ **Figure EV4. Ago2 distribution in breast and lung disease.**

(**A**) Ago2 distribution (red) in tissues in the breast disease spectrum array, analyzed using immunofluorescence. Cytokeratin was recognized using anti-cytokeratin 8/18 (green). Images of representative samples are shown. Scale bar = 50 μm. Representative immunofluorescence images of tissues in the breast disease spectrum array are depicted. Scale bar = 50 μm. (**B**) Percentage of Ago2 overlaps with cytokeratin 8/18 in images of tissues in the breast disease spectrum array. Data were identical to those shown in Fig. 8F and are re-sorted by the cancer stage along the *x*-axis. Each spot describes the percentage of Ago2 overlaps with cytokeratin 8/18 in a tissue sample. Bars are means ± SEM. The sample size (*n*, patient number) of each category is depicted on the *x*-axis. Student's *t*-test, $^*P \leq 0.05$; $^{**}P \leq 0.01$; $^{***}P \leq 0.001$. (**C**) Ago2 distribution in lung tissues and lung carcinomas in the lung disease spectrum array. Each spot describes the percentage of Ago2 overlaps with cytokeratin 8/18 in a tissue sample. Bars are means ± SEM (*n* = 13 and 88, patient number). Student's *t*-test, *P* values are indicated on each plot.

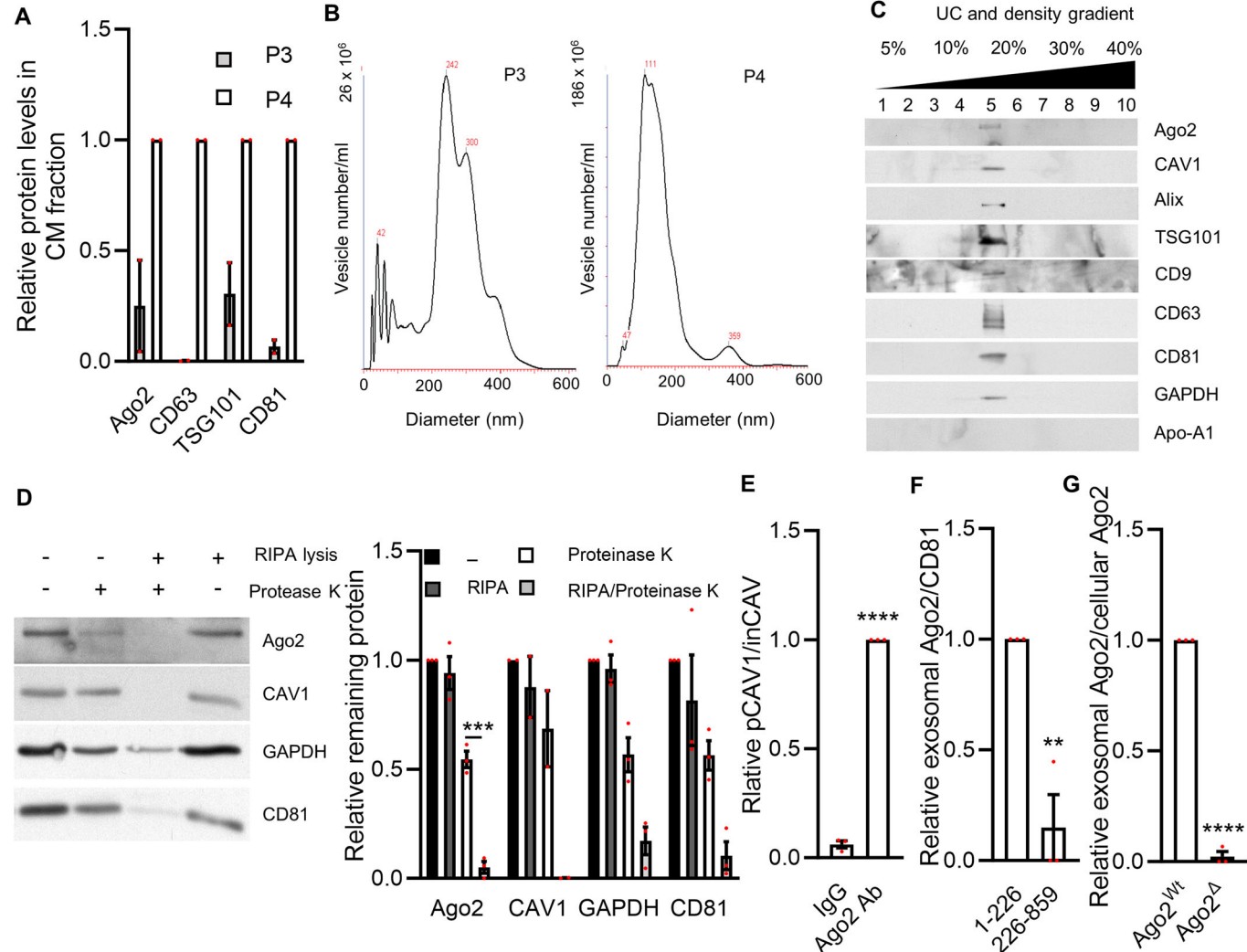

**Figure EV5. Both Ago2 and CAV1 exist in cancer cell-derived EVs.**

(A) The quantitation of Western blots of experiments in Fig. 9A. Each spot describes the protein level of the sample normalized by GAPDH, relative to that of the fourth fraction sample in each replicated experiment. Bars are means ± SEM ($n = 2$). (B) P3 and P4 fractions of A549 cell-conditioned media were collected and subjected to NanoSight nanoparticle tracking analysis, revealing their particle size and concentration. (C) Ago2 and small EVs in the same fraction of A549 cell-conditioned media separated with a density gradient. Ago2, CAV1, CD63, TSG101, CD9, and CD81 were measured using Western blotting in each fraction of the density gradient. (D) Proteinase K protection assay revealing Ago2 in EVs. A549 cell-derived EVs were treated using RIPA lysis buffer or protease K as indicated. Ago2, CAV1, and CD81 were measured using Western blotting in the treated EVs. Right panel: the quantitation of Western blots. Each spot describes the protein level, relative to that of PBS(-)-treated sample in each replicated experiment. Bars are means ± SEM ($n = 3$, biological replicates). Student's $t$-test, ***$P \le 0.001$. (E) The quantitation of Western blots of experiments in Fig. 9B. Each spot describes the ratio of precipitated CAV1 (pCAV1) to input CAV1 (inCAV1) in the sample, relative to that of the sample precipitated by anti-Ago2 antibodies in each replicated experiment. Bars are means ± SEM ($n = 3$, biological replicates). Student's $t$-test, ****$P \le 0.0001$. (F) The quantitation of Western blots of experiments in Fig. 9C. Each spot describes the exosomal Ago2 protein level of the sample normalized by exosome marker CD81, relative to that of the exosome sample of HEK293 cells expressing HA-Ago2(1–226) in each replicated experiment. Bars are means ± SEM ($n = 3$, biological replicates). Student's $t$-test, **$P \le 0.01$. (G) The quantitation of Western blots of experiments in Fig. 9D. Each spot describes the ratio of exosomal Ago2 to cellular Ago2 in the sample, relative to that of A549$^{Ago2KO/HA-Ago2Wt}$ (Ago2$^{Wt}$) sample in each replicated experiment. Bars are means ± SEM ($n = 3$, biological replicates). Student's $t$-test, ****$P \le 0.0001$.

