## [Peer Review File · EMBO Reports]

Ago2/CAV1 interaction potentiates metastasis via controlling Ago2 localization and miRNA action

Meng-Chieh Lin, Wen-Hung Kuo, Shih-Yin Chen, Jin-Ya Hsu, Li-Yu Lu, Chen-Chi Wang, Yi-Ju Chen, Jia-Shiuan Tsai, and Hua-Jung Li

Corresponding author(s): Hua-Jung Li (annli@nhri.edu.tw)

Review Timeline:

Submission Date:	7th Nov 23
Editorial Decision:	28th Nov 23
Revision Received:	1st Mar 24
Editorial Decision:	20th Mar 24
Revision Received:	22nd Mar 24
Accepted:	27th Mar 24

Editor: Esther Schnapp

Transaction Report:

Dear Dr. Li,

Thank you for the submission of your manuscript to EMBO reports. We have now received the full set of referee reports as well as referee cross-comments that are pasted below.

As you will see, the referees acknowledge that the findings are potentially interesting. However, together they also raise several concerns that should all be addressed to strengthen the data and the study, except for referee 1 point 5 and referee 3 point 2 (whether CAV1/Ago2 functions through the miR-3613-3P/SCAI axis), which do not have to be addressed experimentally. Naturally, if you have data to address these issues, you are very welcome to include these. If no further data on the role of miR-3613-3p downstream of the Ago2/Cav1 interaction will be added, please tone down all statements in the manuscript that the pro-tumorigenic effects of the Ago2/Cav1 interaction are mediated by this miRNA. Referee 3 specifies in her/his cross-comments which experiments should be repeated with the Ago2 interaction mutants, and we agree that these experiments should be performed. It would also be helpful to know whether these Ago2 mutants remain active.

If you have any further questions or comments regarding the revisions, please contact me. We can also make an appointment for a video chat, if this is helpful.

I would thus like to invite you to revise your manuscript with the understanding that the referee concerns must be fully addressed and their suggestions taken on board. Please address all referee concerns in a complete point-by-point response. Acceptance of the manuscript will depend on a positive outcome of a second round of review. It is EMBO reports policy to allow a single round of major revision only and acceptance or rejection of the manuscript will therefore depend on the completeness of your responses included in the next, final version of the manuscript.

We realize that it is difficult to revise to a specific deadline. In the interest of protecting the conceptual advance provided by the work, we recommend a revision within 3 months (28th Feb 2024). Please discuss the revision progress ahead of this time with the editor if you require more time to complete the revisions.

- 1) A data availability section providing access to data deposited in public databases is missing. If you have not deposited any data, please add a sentence to the data availability section that explains that.
- 2) Your manuscript contains statistics and error bars based on $n=2$. Please use scatter blots in these cases. No statistics should be calculated if $n=2$.

5) a complete author checklist, which you can download from our author guidelines <https://www.embopress.org/page/journal/14693178/authorguide>. Please insert information in the checklist that is also

reflected in the manuscript. The completed author checklist will also be part of the RPF.

6) Please note that all corresponding authors are required to supply an ORCID ID for their name upon submission of a revised manuscript (<<https://orcid.org/>>). Please find instructions on how to link your ORCID ID to your account in our manuscript tracking system in our Author guidelines <<https://www.embopress.org/page/journal/14693178/authorguide#authorshipguidelines>>

- the name of the statistical test used to generate error bars and P values,
- the number (n) of independent experiments (please specify technical or biological replicates) underlying each data point,
- the nature of the bars and error bars (s.d., s.e.m.),
- If the data are obtained from n=2, please use scatter blots showing the individual data points.

I look forward to seeing a revised form of your manuscript when it is ready.

Referee #1:

The manuscript entitled "Interaction of CAV1 and Ago2 regulates metastasis via controlling Ago2 localization and miRNA action" describes novel findings about Argonaute 2 and caveolin-1 interaction and its functional impact on tumor metastasis. The manuscript by Lin et al. contains a lot of data and in many places is thorough. They carry out a detailed dissection of how Ago2 and Caveolin interact and its regulation by acetylation. They also analyze the impact of this interaction on miRNA levels in normal and cancer cells, as well as on tumor metastasis. Their findings are very interesting and may lead the way to understanding the role of localization to membranes for RISC function. A few things need tightening up in the paper, particularly some claims about the Ago2/Cav-1 interaction on the plasma membrane and its subsequent release into the extracellular vesicles. They have not used methods that properly assess the localization to the plasma membrane as opposed to other cellular membranes, nor have they quantitated it. There are also a few other things that need tightening up, especially lack of quantitation of the numerous Western blots that are throughout the manuscript and lack of description of how many times those unquantitated Western blot experiments were performed. Please see the comments below for details.

1. In Figure 2, the authors show Ago2, caveolin-1, and CTB all colocalizing in plasma membrane. What percentage of colocalization do authors find in the plasma membrane compared to the rest of the cell? What about localization to endosomes or other cellular membranes? Localization with proper plasma membrane and endosome markers should be done and quantification of all imaging experiments should be performed, plotted and analyzed for statistics. The number of replicates also needs to be specified for all imaging experiments.
2. The authors show protein distribution in cytosolic and membrane fractions in Figure 2 (Panel E, F, and G). They use a Subcellular Protein Fractionation Kit (cat. No. 78840, Thermo). The kit only fractionates cytosolic fractions from the total membrane and nuclear fractions. How do authors depict that the total membrane fraction is only comprised of the plasma membrane where the kit mentioned that it could dissolve plasma membrane, ER-Golgi membrane, and mitochondria membrane as well (and presumably endosomes too)? How do they infer that Ago2 and Cav-1 are only present in the plasma membrane and interact with each other? Previous studies show RISC machinery, including Ago2, present in the endoplasmic reticulum membrane (Stalder et al., EMBO J, 2013; Barman et al., JBC, 2015) and at endosomal multivesicular bodies before being secreted into the extracellular vesicles (McKenzie et al., Cell Reports, 2016, Albacete-Albacete et al., JCB, 2020). How does this manuscript converge or diverge from the previous studies? Please discuss in the manuscript and address other localizations, as requested in point 1.
3. There is a lot of missing quantitations for Western blots. All data for which conclusions are being drawn should have quantitations, with the replicate number indicated in the legends. This includes data in Figs 1, 2, 3, 6, 9, S1, S2, S4, S7C.
4. In Figure 3, the authors show that synthetic P2 peptide blocks Ago2/Cav-1 interaction and thus affects microRNA expression in the cancer cells. Panel A shows miR-3613-3p was the most repressed microRNA when treated with P2 peptide. In panel D, the authors show the subcellular distribution of the same microRNA. What is the relevance of this experiment if miR-3613-3p is already repressed in P2-treated cells? Also, how do they normalize this QRT-PCR? Was it by U6? What is the distribution of U6 in CF and MF? If it is a relative miR-3613-3p graph, which one do you consider 1? Please specify.
5. In Figure 7, the authors show that Ago2/cav-1 interaction is necessary for Ago2 sorting into the small extracellular vesicles. Can they describe how Ago2-containing small extracellular vesicles are budding off the plasma membrane? What is the role of Cav-1 in this process?
6. In Figure 9, the authors show the miR-3613-3p secretion is reduced into the extracellular vesicles. What is the rationale of this experiment if miR-3613-3p is already less expressed in their parental cell? Some analysis of EV/cell ratio would likely be informative. Also, what are those other microRNAs in panel F? How did they pick them? Panel G shows plasma miR-3613-3p levels in WT and DM-injected A549 cells in mice. What is the level of miR-3613-3p levels in regular (placebo injected) mice?
7. Fig 1A,B, missing some controls for IPs. Not all cell samples have an IgG only control and no samples show the supernatant from the IPs.
8. More information is needed about how EVs were isolated. What media was used to condition the cells? Does it contain serum (might explain all the extravesicular Ago2 if so)? For the plasma isolation, how was it done? For the density gradient, that should be included in the methods.
9. In many places, RIPA buffer is said to be used, but there are many types of RIPA buffer, please specify what the composition is.

Referee #2:

In the manuscript titled "Interaction of CAV1 and Ago2 regulates metastasis via controlling Ago2 localization and miRNA" by Lin et al., the authors report the recruitment of Ago2 to the plasma membrane through interaction with Caveolin-1 (CAV1). The study identifies crucial residues, K212 and W211, in the Ago2/CAV1 binding motif and demonstrates that disrupting this complex reduces tumor sphere formation, anoikis, and metastasis. Additionally, the authors find that SIRT2-mediated deacetylation on K211 enhances Ago2-CAV1 interaction. Membrane-bound Ago2 increases the levels of miR-3613-3p, targeting the tumor suppressor SCAI. While the spatial regulation of microRNA functions is a novel aspect, concerns include the need for validation of the roles of miR-3613-3p and SCAI in cancer cells, insufficient study of miR-3613-3p regulation on SCAI, and the potential distraction introduced by extracellular vesicle data (Fig. 9).

The manuscript could benefit from addressing several key concerns.

Major points include the need for confirmation of the roles of miR-3613-3p and SCAI in cancer cells, requiring validation of whether CAV1/Ago2 functions through the miR-3613-3P/SCAI axis. Further study on miR-3613-3P regulation of SCAI is necessary, and supporting evidence like binding site mutations or Ago2 IP would strengthen the understanding of their interaction. Additionally, the extracellular vesicle data (Fig. 9) is intriguing but may not be essential for this manuscript. Certain figures, such as Fig. 1 and Fig. 2, could be condensed for clarity.

Minor points include the need for thorough English editing for accuracy, addressing typos (e.g., "SCAISCAI" on page 8), defining the term "MF" upon first use, and avoiding redundancy in defining indole-3-acetic acid (IAA) twice. To enhance conciseness, some experimental details could be relocated to the methods section, for instance, the description of mouse injections (e.g., "At 2 weeks after implantation, the mice were injected with phosphate-buffered saline (PBS) or IAA for 3 weeks (Fig. 5Ai)" on page 10).

Referee #3:

Lin et al identify Argonaute-2 (Ago2) as a novel Caveolin-1 (Cav-1)-interacting protein. They find that deacetylation of K212 of Ago2 by Sirtuin2 (SIRT2) is necessary for binding to Cav-1. They describe how Cav-1-mediated sequestration of Ago2 at the plasma membrane promotes miR-3613-3p-mediated suppression of SCAI (suppressor of cancer cell invasion) in cancer cells. They provide evidence that the interaction between Cav-1 and Ago2 promotes tumor formation and metastasis in mice and it is associated with human carcinoma progression and metastasis. They show that the Cav-1/Ago2 interaction promotes the release of miRNA-3613-3p, through extracellular vesicles (EVs), from cancer cells both in culture and in tumor-bearing mice. Consistent with these findings, elevated plasma levels of EVs containing miRNA-3613-3p correlates with increased interaction between Cav-1 and Ago2 in human metastatic carcinomas, suggesting that miRNA-3613-3p in EVs may represent a biomarker of human metastatic carcinomas. They conclude that Cav-1-mediated sequestration of Ago2 at the plasma membrane controls the pro-metastatic functions of miRNA-3613-3p. The manuscript is well-written and easy to follow. The study is significant in the cancer field because it provides novel insights into the pro-tumorigenic role of Ago2-mediated miRNA regulation.

The following concerns need to be addressed:

Main points:

1) The use of the P2 peptide to determine the role of Cav-1 in Ago2 function is a concern. The P2 peptide is likely to prevent the interaction of Cav-1 with a number of other Cav-1-interacting proteins, in addition to Ago2. As such, in contrast to what stated by the authors, the results obtained using the P2 peptide (in Figures 2D, 2E, and in most of Figure 3) do not specifically link a lack of sequestration of Ago2 in caveolae at the plasma membrane to miRNA processing. The P2 peptide could alter miRNA function at the plasma membrane, including miR-3613-3p-mediated SCAI suppression, in an Ago2-independent manner. Similarly, the outcome of the functional studies described in Figure 4 using the P2 peptide could also be explained by a lack of interaction with Cav-1 of proteins other than Ago2.

2) While the authors convincingly show that the interaction of Ago2 with Cav-1 at the plasma membrane plays an important role in tumor growth and metastasis, it is not clear whether these functional outcomes are mediated by the miR-3613-3p-dependent suppression of SCAI expression. Does forcing SCAI downregulation in their system rescue tumor growth and metastasis in cancer cells expressing the mutant forms of Ago2 (Ago2delta, Ago2K212A) that can not interact with Cav-1 and in which SCAI expression remains elevated?

Minor points:

- 1) Quantification of IF data in Figure 2 should be included.
- 2) Quantification of immunoblotting data in Figure 3H should be included.
- 3) Figures 4Gi and 4Gii are mentioned on page 9 but there is no panel G in Figure 4.
- 4) The authors should at least comment on how Ago2 moves to caveolar membranes.

Cross-comments from referee 1:

For the P2 peptide, I guess I was convinced by the authors acknowledging that it would disrupt other interactions with Cav1 and

therefore complementing the peptide work with an Ago2 mutant that couldn't bind Cav1. However, perhaps there is a key experiment that was not done with the Ago2 mutant that Reviewer 3 thinks should be done?

For the functional outcomes, the authors did not try to say the miR-3613-3p and SCAI were responsible for the in vivo metastasis functions and I think it would be too much to ask. They instead focused on it as a biomarker. I think if they want to investigate this as a function, that could be a separate paper. I would prefer to see the authors solidify the findings and claims that they have with much more quantitation.

Cross-comments from referee 2:

1. I think the P2 peptide data supports that disruption of the interaction between Ago2 and CAV1 impair tumor growth and metastasis. Given that the mechanism of Ago2 and CAV1 interaction is well demonstrated, which is the main part of this paper, I do not think further work on the P2 peptide is really necessary.
2. I agree that functional analysis of mir-3613-3p and SCAI is necessary because the key finding of this work is that CAV1 sequestered Ago2 modulate the functions of specific miRNAs.

Cross-comments from referee 3:

- 1) Among the experiments in which the authors have used the P2 peptide to disrupt the Ago2/Cav-1 interaction (which likely disrupts also the interaction of Cav-1 with other Cav-1-interacting proteins), the miRNA array data (Fig 3A) and the data on mesenchymal markers (Fig 3H) have not been confirmed using the Ago2 mutant that can not interact with Cav-1. The authors should at least confirm the data of Fig 3A and 3H with the Ago2 mutant that can not interact with Cav-1.
- 2) I agree with Reviewer #2 and remain of the idea that while the authors convincingly show that the interaction of Ago2 with Cav-1 at the plasma membrane plays an important role in tumor growth and metastasis, it is not clear whether these functional outcomes are mediated by the miR-3613-3p-dependent suppression of SCAI expression.

Point-by-point replies to the comments

Referee #1:

The manuscript entitled "Interaction of CAV1 and Ago2 regulates metastasis via controlling Ago2 localization and miRNA action" describes novel findings about Argonaute 2 and caveolin-1 interaction and its functional impact on tumor metastasis. The manuscript by Lin et al. contains a lot of data and in many places is thorough. They carry out a detailed dissection of how Ago2 and Caveolin interact and its regulation by acetylation. They also analyze the impact of this interaction on miRNA levels in normal and cancer cells, as well as on tumor metastasis. Their findings are very interesting and may lead the way to understanding the role of localization to membranes for RISC function. A few things need tightening up in the paper, particularly some claims about the Ago2/Cav-1 interaction on the plasma membrane and its subsequent release into the extracellular vesicles. They have not used methods that properly assess the localization to the plasma membrane as opposed to other cellular membranes, nor have they quantitated it. There are also a few other things that need tightening up, especially lack of quantitation of the numerous Western blots that are throughout the manuscript and lack of description of how many times those unquantitated Western blot experiments were performed. Please see the comments below for details.

1. In Figure 2, the authors show Ago2, caveolin-1, and CTB all colocalizing in plasma membrane. What percentage of colocalization do authors find in the plasma membrane compared to the rest of the cell? What about localization to endosomes or other cellular membranes? Localization with proper plasma membrane and endosome markers should be done and quantification of all imaging experiments should be performed, plotted and analyzed for statistics. The number of replicates also needs to be specified for all imaging experiments.

ANS 1:

[Ago2 localization to different cellular membranes]

To answer the question regarding the percentage of Ago2/CAV1 colocalization in plasma membrane, and other cellular membranes including endosome and ER, additional experiments analyzing Ago2 localization with plasma membrane and endosome markers were conducted with BEAS-2B normal epithelial cells and A549 cancer cells. The distribution of Ago2, CAV1, endosome marker EEA1, and ER marker Calnexin (CANX) in BEAS-2B and A549 cells was analyzed (Fig. 2C, 2D and Supplementary Fig. 2E of the revised manuscript). There were ~5.2% of Ago2 overlapping with CAV1 in A549 cancer cells and ~2.5% of Ago2 overlapping with CAV1 in BEAS-2B normal epithelial cells (Fig. 2C and 2Di of the revised manuscript). The levels of Ago2 overlapping with CAV1 in A549 cancer cells (Fig. 2C, arrowheads of the revised manuscript) were higher than that in BEAS-2B normal epithelial cells (Fig. 2C and 2Di of the revised manuscript). To analyze the association of Ago2 with endosomes, we observed that the levels of Ago2 overlapping with endosome marker EEA1 in A549 cancer cells were higher than that in BEAS-2B cells (Supplementary Fig. 2E, left panel, and 2Diii of the revised manuscript; 1.2% vs. 0.5%). For Ago2 association with ER, we observed that the levels of Ago2 overlapping with ER marker CANX in both BEAS-2B normal cells and A549 cancer cells were similar (Supplementary Fig. 2E, right panel, and 2Dv of the revised manuscript; 7%). The results of the additional experiments are described on page 7 of the text of the revised manuscript and are shown in Figure 2C, 2D and Supplementary Figure 2E of the revised manuscript.

[Quantification of imaging experiments]

Quantification of imaging experiments of the manuscript has been performed, plotted, analyzed for statistics, and present in figures 2Di-vi, 2G, 2I, 8E, 8F, and supplementary figures 2Hi and 8B with the number of replicates in figure legends. The software and method used for the quantification are described in Materials and Methods on page 23 and 24 of the revised manuscript.

2. The authors show protein distribution in cytosolic and membrane fractions in Figure 2 (Panel E, F, and G). They use a Subcellular Protein Fractionation Kit (cat. No. 78840, Thermo). The kit only fractionates cytosolic fractions from the total membrane and nuclear fractions. How do authors depict that the total membrane fraction is only comprised of the plasma membrane where the kit mentioned that it could dissolve plasma membrane, ER-Golgi membrane, and mitochondria membrane as well (and presumably endosomes too)? How do they infer that Ago2 and Cav-1 are only present in the plasma membrane and interact with each other? Previous studies show RISC machinery, including Ago2, present in the endoplasmic reticulum membrane (Stalder et al., EMBO J, 2013; Barman et al., JBC, 2015) and at endosomal multivesicular bodies before being secreted into the extracellular vesicles (McKenzie et al., Cell Reports, 2016, Albacete-Albacete et al., JCB, 2020). How does this manuscript converge or diverge from the previous studies? Please discuss in the manuscript and address other localizations, as requested in point 1.

ANS 2:

[4 different cell fractionation kits used in the study]

To validate Ago2 distribution, there were 4 different cell fractionation kits used in the study ---

- (1) Subcellular Protein Fractionation Kit (cat. No. 78840, Thermo) for separating cytosolic, nuclear and total membrane fractions,
- (2) Plasma Membrane Protein Extraction Kit (cat. No. K268-50/ab65400, BioVision) for separating plasma membrane fractions and non-plasma membrane organelle (the bottom phase of step B4 of the manufacturer's instructions) fractions,
- (3) Minute™ Plasma Membrane Protein Isolation and Cell Fractionation Kit (cat. No. SM-005, Inventbiotech) for isolating plasma membranes,
- (4) Minute™ Endosome Isolation and Cell Fractionation Kit (cat. No. ED-028, Inventbiotech) for isolating endosomes.

All of the isolated fractions were validated with fraction-specific markers, including α -tubulin for cytosolic fractions, HDAC1 for nuclear fraction, CAV1 for plasma membrane fractions, EEA1 for endosome fractions, CALR for non-plasma membrane organelle (ER) fractions (Ref: <https://www.abcam.com/primary-antibodies/subcellular-marker-resources>). In the experiments of figure 2A and 2B with (1) Thermo Subcellular Protein Fractionation Kit, we found that blocking Ago2/CAV1 interaction interferes with Ago2 distribution between cytosolic and total membrane fractions (Fig. 2A and 2B of the revised manuscript). In the experiments of figure 2E with (2) Biovision Plasma Membrane Protein Extraction Kit, we found that the association of Ago2 with the plasma membranes increases in lung cancer cells A549, compared with that of normal epithelial cells BEAS-2B (Fig. 2E of the revised manuscript). In

the additional experiment of Fig. 6G of the revised manuscript with (3) Minute™ Plasma Membrane Protein Isolation and Cell Fractionation Kit and (4) Minute™ Endosome Isolation and Cell Fractionation Kit, we isolated the plasma membrane fractions and endosome fractions of A549^{Ago2-KO/HA-Ago2Wt} and A549^{Ago2-KO/HA-Ago2K212A} cells to explore effects of Ago2 lysine 212 substitution on Ago2 association with the plasma membranes and endosomes (Fig. 6G of the revised manuscript). The application of cell fractionation kits for each experiment is described in “Materials and Methods” on page 21 of the revised manuscript.

[Ago2/CAV1 interaction in Ago2 association with different cellular membranes]

Previous studies show that Ago2 is present in the endoplasmic reticulum (ER) membrane (Barman and Bhattacharyya, 2015; Stalder et al., 2013), endosome (McKenzie et al., 2016), and plasma membrane (Shankar et al., 2020). To address whether Ago2/CAV1 interaction is also involved in Ago2 association with cellular membranes other than plasma membrane, we have performed additional experiments analyzing the distribution of Ago2, CAV1, endosome marker EEA1, and ER marker Calnexin calreticulin (CANXLR) in BEAS-2B normal lung epithelial cells, A549, A549Ago2-KO/HA-Ago2Wt, and A549Ago2-KO/HA-Ago2Δ lung cancer cells was analyzed (Fig. 2C and Supplementary Fig. 2E of the revised manuscript). The levels of Ago2 overlapping with CAV1 in A549 cancer cells (Fig. 2C of the revised manuscript, arrowheads) were higher than that in BEAS-2B normal lung epithelial cells (Fig. 2C and 2Di of the revised manuscript). Furthermore, the levels of HA-Ago2Wt overlapping with CAV1 in A549Ago2-KO/HA-Ago2Wt cancer cells (Fig. 2C, arrowheads) were higher than that of CBM (caveolin binding motif)-deleted HA-Ago2Δ in A549Ago2-KO/HA-Ago2Δ cells (Fig. 2C and 2Dii of the revised manuscript). Deletion of CBM of Ago2 decreases Ago2 association with CAV1 in cancer cells, suggesting that CBM of Ago2 is responsible for the increased association of Ago2 with CAV1 in A549 lung cancer cells.

To analyze the association of Ago2 with endosomes, we observed that the levels of Ago2 overlapping with endosome marker EEA1 in A549 cancer cells were higher than that in BEAS-2B normal cells (Supplementary Fig. 2E, left panel, and 2Diii). However, the levels of HA-Ago2Wt and CBM-deleted HA-Ago2Δ overlapping with EEA1 in A549 cancer cells were not different (Supplementary Fig. 2E, left panel, and 2Div). The association of Ago2 with endosomes increases in cancer cells but CBM of Ago2 is not required for the increased association with endosomes in the cancer cells. It has been shown that S387 phosphorylation of Ago2 controls Ago2 association with endosomes in cancer cells (McKenzie et al., 2016). However, in additional experiments we observed that neither S387A substitution, dephosphorylation mimic, or S387E substitution, phosphorylation mimic, of Ago2 altered Ago2/CAV1 interaction (Supplementary Fig. 1H), supporting that Ago2/CAV1 interaction is not responsible for Ago2 association with endosomes in cancer cells.

For Ago2 association with ER, we observed that the levels of Ago2 overlapping with ER marker CANX in both BEAS-2B normal lung epithelial cells and A549 cancer cells were similar (Supplementary Fig. 2E, right panel, and 2Dv). In addition, the levels of HA-Ago2Wt and CBM-deleted HA-Ago2Δ overlapping with CANX in A549A cancer cells were not different (Supplementary Fig. 2E, right panel, and 2Dvi). The results suggest that the association of Ago2 with ER remains the same in normal lung epithelial cells and lung cancer cells and deletion of CBM of Ago2 does not affect the association of Ago2 with ER in cancer cells.

Similar to blocking Ago2/CAV1 interaction with P2 peptides and with deletion of the CBM domain of Ago2 (Fig. 2C, 2D, 2H, and 2I), blocking Ago2/CAV1 interaction with Ago2 K212A substitution also decreased

association of Ago2 with the plasma membranes and did not alter Ago2 association with endosomes in cancer cells (Fig. 6G and Supplementary Fig. 6F). We found that Ago2/CAV1 interaction is necessary for the plasma membrane association of Ago2 in the cancer cells. CAV1 has been found predominantly at the plasma membranes and marginally in endosomes and ER (Albacete-Albacete et al., 2020; Williams and Lisanti, 2004). It corresponds with our observation that blocking Ago2/CAV1 interaction disrupts Ago2 association with the plasma membranes, which contain abundant CAV1 (Fig. 2H, 2I and 6G), but does not significantly interfere with Ago2 association with endosomes and ER, which only contain slight CAV1, in the cancer cells (Fig. 2D, 6G and Supplementary Fig. 2E).

The results of additional experiments are described on page 7, 8, 13 of the text of the revised manuscript, are discussed on page 17 of the revised manuscript, and are shown in Figure 2C, 2D, 6G and Supplementary Figure 2E and 6F of the revised manuscript.

3. There is a lot of missing quantitations for Western blots. All data for which conclusions are being drawn should have quantitations, with the replicate number indicated in the legends. This includes data in Figs 1, 2, 3, 6, 9, S1, S2, S4, S7C.

ANS 3:

[Quantification of Western blots]

The quantification of Western blots of the manuscript has been performed, plotted, analyzed for statistics, and present in figures 2Aii, 2Bii, 2Eii, 6Fii, and supplementary figures 1A, 1C, 1D, 1E, 1F, 1Hii, 1I, 1J, 1K, 1L, 1Oii, 1P, 2Ciii, 2Gii, 2Hii, 3F, 3H, 4Bii, 4Cii, 6, 8A, 8D, 8E, 8F, 8G of the revised manuscript with the number of replicates in figure legends.

4. In Figure 3, the authors show that synthetic P2 peptide blocks Ago2/Cav-1 interaction and thus affects microRNA expression in the cancer cells. Panel A shows miR-3613-3p was the most repressed microRNA when treated with P2 peptide. In panel D, the authors show the subcellular distribution of the same microRNA. What is the relevance of this experiment if miR-3613-3p is already repressed in P2-treated cells? Also, how do they normalize this QRT-PCR? Was it by U6? What is the distribution of U6 in CF and MF? If it is a relative miR-3613-3p graph, which one do you consider 1? Please specify.

ANS 4:

[No relevance of the subcellular distribution of a repressed miRNA/mRNA]

We agree with the referee's concern on the relevance of the distribution of miR-3613, which is already suppressed in the cells. In the Fig. 3D of the previous version of manuscript (shown as Fig. a here; removed from the revised manuscript), we showed that miR-3613-3p was significantly suppressed in the membrane fraction (MF) of cancer cells by P2 peptides. Although there was also a downward

Figures for referees not shown.

trend of miR-3613-3p expression in the cytosolic fraction (CF) of cancer cells treated by P2 peptides, it is not statistically different from that of P2S-treated cells. Therefore, in the previous version of manuscript, we concluded that “blocking Ago2/CAV1 interaction decreased the level of miR-3613-3p in the MF of cancer cells but did not affect the level of miR-3613-3p in the cytosolic fraction”. However, we did not consider that the distribution of miR-3613-3p in the untreated cancer cells is already different. Since the level of miR-3613-3p in CF fraction of the untreated cancer cells is already lower than that of MF fraction, the non-significant suppression of miR-3613-3p in CF fraction by P2 peptide could be due to the very low level of miR-3613-3p in CF fraction of the untreated cells. It is also not relevant to compare the distribution of miR-3613-3p in the P2-treated cancer cells, in which miR-3616-39 expression is suppressed. Therefore, we withdraw the claim that “blocking Ago2/CAV1 interaction decreased the level of miR-3613-3p in the MF of cancer cells but did not affect the level of miR-3613-3p in the cytosolic fraction” and remove Fig.3D and the related 3G (P2 effects on miR-3613-3p target mRNA in CF vs. MF) from the revised manuscript.

[Normalization of qPCR data]

Here we explain how we normalized the qPCR data in Fig. 3D of the previous version of manuscript (shown as Fig. a here; removed from the revised manuscript). The miR-3613-3p levels in Fig. a are normalized by the levels of U6. The distribution of U6 in CF and MF is shown in Fig. b here. For relative miR-3613-3p of Fig. a, we set the miR-3613-3p level (normalized with U6) of the replicated experiment which is closest to the average of the four replicated experiments (the red spot in Fig. a) as 1. The method is also applied to normalization of other qPCR data in the revised manuscript.

5. In Figure 7, the authors show that Ago2/cav-1 interaction is necessary for Ago2 sorting into the small extracellular vesicles. Can they describe how Ago2-containing small extracellular vesicles are budding off the plasma membrane? What is the role of Cav-1 in this process?

ANS 5:

[The role of CAV1 in release of Ago2-containing EVs]

It has also been shown by others and us that the plasma membrane-associated proteins are sorted into EVs in a CAV1-dependent manner (Albacete-Albacete et al., 2020; Lin et al., 2017b). In the process of EV biogenesis, Cav1 relocates to multivesicular bodies (MVBs) through endocytosis (Albacete-Albacete et al., 2020). Our previous study has shown that both depletion of CAV1 with siRNA and disruption of lipid rafts with M β CD of the cells selectively decrease levels of plasma membrane-associated proteins in the cell derived EVs (Lin et al., 2017b). These studies demonstrate that CAV1 is responsible for sorting the plasma membrane-associated components into EVs. In this study, we observed that by docking at the plasma membrane via Ago2/CAV1 interaction, Ago2 can be sorted into the EVs of cancer cells (Fig. 9B-9D and Supplementary Fig. 8E-8G of the revised manuscript). It suggests that Ago2 may be sorted with CAV1, which relocates to MVBs and then EVs through endocytosis, into the EVs.

The results of related experiments are described on page 15 of the text of the revised manuscript and are shown in Figure 9B, 9C, 9D, and Supplementary Figure 8E-8G. The notion is discussed on page 18 and 19 of the revised manuscript.

6. In Figure 9, the authors show the miR-3613-3p secretion is reduced into the extracellular vesicles. What is the rationale of this experiment if miR-3613-3p is already less expressed in their parental cell? Some

analysis of EV/cell ratio would likely be informative. Also, what are those other microRNAs in panel F? How did they pick them? Panel G shows plasma miR-3613-3p levels in WT and DM-injected A549 cells in mice. What is the level of miR-3613-3p levels in regular (placebo injected) mice?

ANS 6:

[miRNA in cancer cell-derived EVs]

The rationale of the experiments is to evaluate whether the elevated miRNAs in cancer cells in which Ago2/CAV1 interaction is maintained could be released via EVs and be detected in plasma as biomarkers. The miRNAs listed in Fig. 9F are EV miRNAs that differed significantly, by more than 1.5 folds, between the plasma of mice bearing A549^{Ago2-KO/HA-Ago2Wt} (Wt) and A549^{Ago2-KO/HA-Ago2Δ} (Dm), measured by miRNA assays. The green indicates down-regulated miRNAs, and the red indicates up-regulated miRNAs in the plasma EVs of mice bearing A549^{Ago2-KO/HA-Ago2Δ} (Dm) tumors. We compared the miRNAs of Figure 9F and the miRNAs which were differentially expressed in A549^{Ago2-KO/HA-Ago2Wt} and A549^{Ago2-KO/HA-Ago2Δ} cancer cells (Fig. 6H, Ago2Dm). Only miRNA-3613-3p and miRNA-877-5p were increased with Ago2/CAV1 interaction in both cancer cells and plasma EV of the cancer cell-bearing mice (Fig. 9F, Blue), suggesting that the miRNAs elevated in A549^{Ago2-KO/HA-Ago2Wt} cells, compared to that of A549^{Ago2-KO/HA-Ago2Δ} cells, are not necessarily elevated in the plasma EV of A549^{Ago2-KO/HA-Ago2Wt} tumor-bearing mice.

We further examined whether these elevated miRNAs in the cancer cells can be detected in plasma EV as biomarkers (Fig. 9G and 9H of the revised manuscript). The EV-mediated release of miRNAs, miRNA-3613-3p, miRNA-877-5p, and miRNA-6126, in the plasma of Wt and Dm tumor-bearing mice and non-tumor-bearing mice (Ctrl; Fig. 9G) and in the plasma of patients with tumors (Fig. 9H) was analyzed using qPCR.

Additional experiments have been performed to examine the level of miR-3613-3p miRNA-877-5p, and miRNA-6126, in regular mice (placebo injected non-tumor-bearing mice). The results of experiments are described on page 15 of the text of the revised manuscript and are shown in Figure 9F, 9G, and 9H of the revised manuscript.

7. Fig 1A,B, missing some controls for IPs. Not all cell samples have an IgG only control and no samples show the supernatant from the IPs.

ANS 7:

[Additional experiments with IgG control for all cell samples]

Additional experiments with IgG control and anti-Ago2 IgG for all cell samples in Figures 1A and 1B have been performed. The results of the experiments are shown in Figure 1A and Supplementary Fig. 1B of the revised manuscript.

[Supernatant from the IPs]

The western blotting results of supernatant from the IPs of Fig. 1A and 1B are shown here. Since our IP buffer [PBS with 0.5% Nonidet™ P-40, 5% BSA and mixed protease inhibitors] contains 5% BSA, it interfered with the gel-running and antibody blotting. Both anti-CAV1 blotting and anti-Ago2 blotting were affected by BSA as seen in Figure c. Ago2 is hard to be visualized in the blotting under the effect of

BSA since the location of Ago2 is close to BSA on the membrane. The bands of CAV1 are indicated with red arrowheads in Figure c.

8. More information is needed about how EVs were isolated. What media was used to condition the cells? Does it contain serum (might explain all the extravesicular Ago2 if so)? For the plasma isolation, how was it done? For the density gradient, that should be included in the methods.

ANS 8:

[Information of EV isolation, plasma isolation, and density gradient]

EVs were isolated from cell culture media and from plasma by differential ultracentrifugation (Lin et al., 2017a). For preparing cell culture media, EVs in FBS were depleted using ultracentrifugation and filtration as previously described (Eitan et al., 2015; Shelke et al., 2014). FBS was centrifuged at 120,000 g for 8 h at 4 °C. The supernatant was diluted with DMEM at a 1:9 ratio and then filtered through 0.22 µm filter before the DMEM containing 10% EV-depleted FBS was used for cell culture. The fresh DMEM containing 10% EV-depleted FBS was subjected to EV isolation by differential ultracentrifugation as shown below and the depletion of FBS EVs was confirmed using nanoparticle tracking analysis (NTA) and western blot analysis.

After cell culture, the culture media were centrifuged at 300 g for 5 min to remove cells (P1 fraction), at 2,000 g for 20 min (P2 fraction), then at 10,000g for 30 min (P3 fraction) all at 4 °C. Finally, EVs (P4 fraction) were separated from the supernatant by centrifugation at 110,000g for 60 min. The EV pellet was resuspended in PBS and centrifuged at 110,000 g for 60 min again. The final EV pellet was resuspended in PBS for further analysis.

For collecting patient plasma, all research involving patient samples has been reviewed and approved by the Research Ethics Committee of National Taiwan University Hospital. 1 ml whole blood from patients or mice was collected into the EDTA-treated tube. Blood cells were removed from plasma by centrifugation for 15 minutes at 2500 g, 4 °C and the supernatant was centrifuged again at 2500 g, 4 °C for 15 min. 250 µl supernatant was diluted with PBS to 1 ml and then was centrifuged at 10,000 g, 4 °C for 30 min using Rotor tla-120.2. Finally, EVs were separated from the supernatant by centrifugation at 110,000g for 120 min. The EV pellet was resuspended in PBS and centrifuged at 110,000 g for 120 min again. The final EV pellet was resuspended in PBS for further analysis.

EV pellet from ultracentrifugation was resuspended with 40% (w/v) iodixanol in 2 ml PBS. The mixture was overlaid in sequence with aliquots of 30%, 20%, 10%, and 5% (w/v) iodixanol in PBS (2 mL each) to form a density gradient in an ultracentrifuge tube. The mixture was then centrifuged at 200,000 g for 8 h at 4 °C. After the centrifugation, each gradient fraction (10 fractions; 1 mL/fraction) was collected with a pipette from the top of the tube. The presence of Ago2, CAV1, exosome markers (e.g., CD81, CD9, CD63, and Tsg101) and housekeeping protein GAPDH in each fraction was analyzed by SDS-PAGE and Western blot.

The methods for preparation of FBS EV-depleted cell culture medium, plasma EV isolation, EV purification using a density gradient are described on page 26 and 27 of the text of the revised manuscript.

9. In many places, RIPA buffer is said to be used, but there are many types of RIPA buffer, please specify what the composition is.

ANS 9:

[The composition of RIPA buffer]

The composition of RIPA buffer used in this study is “25mM Tris-HCl (pH7.6), 150 mM NaCl, 1% Nonidet P-40, 1% sodium deoxycholate, 0.1% SDS”. The additional complete protease inhibitor and phosphatase inhibitor were added in the RIPA buffer for western blots.

The composition of RIPA buffer is described on page 22 and 27 of the Materials and Methods of the revised manuscript.

Referee #2:

In the manuscript titled "Interaction of CAV1 and Ago2 regulates metastasis via controlling Ago2 localization and miRNA" by Lin et al., the authors report the recruitment of Ago2 to the plasma membrane through interaction with Caveolin-1 (CAV1). The study identifies crucial residues, K212 and W211, in the Ago2/CAV1 binding motif and demonstrates that disrupting this complex reduces tumor sphere formation, anoikis, and metastasis. Additionally, the authors find that SIRT2-mediated deacetylation on K211 enhances Ago2-CAV1 interaction. Membrane-bound Ago2 increases the levels of miR-3613-3p, targeting the tumor suppressor SCAI. While the spatial regulation of microRNA functions is a novel aspect, concerns include the need for validation of the roles of miR-3613-3p and SCAI in cancer cells, insufficient study of miR-3613-3p regulation on SCAI, and the potential distraction introduced by extracellular vesicle data (Fig. 9).

The manuscript could benefit from addressing several key concerns.

Major points include the need for confirmation of the roles of miR-3613-3p and SCAI in cancer cells, requiring validation of whether CAV1/Ago2 functions through the miR-3613-3P/SCAI axis. Further study on miR-3613-3P regulation of SCAI is necessary, and supporting evidence like binding site mutations or Ago2 IP would strengthen the understanding of their interaction. Additionally, the extracellular vesicle data

(Fig. 9) is intriguing but may not be essential for this manuscript. Certain figures, such as Fig. 1 and Fig. 2, could be condensed for clarity.

ANS 10:

[miR-3613-3P/SCAI axis as an example of Ago2/CAV1 interaction-dependent regulation]

We observed that Ago2/VAV1 interaction is associated with tumor metastasis in both animal models and cancer patients. In this study, we would like to claim that through Ago2/CAV1 interaction, Ago2 selectively affects a certain group of miRNAs involved in metastasis and function of the group of miRNAs depends on Ago2/CAV1 interaction. We analyzed and compared miRNA fluctuation resulting from disruption of Ago2/CAV1 interaction by P2 peptides, deletion of Ago2 CBM, and Ago2 K212A substitution with miRNA arrays. There are 10 shared up-regulated miRNAs and 9 shared down-regulated miRNAs in P2-treated A549, A549Ago2-KO/HA-Ago2 Δ (Ago2Dm), and A549Ago2-KO/HA-Ago2K212A (Ago2K212A) cancer cells (Fig. 6Hi of the revised manuscript), including both down-regulated miR-3613-3p and up-regulated miR-6126 (Fig. 6Hii of the manuscript). Among these miRNAs, we used miR-3613-3p as an example to further demonstrate that function, mRNA repression, of some miRNAs also depends on Ago2/Cav1 interaction (Fig. 3H of the manuscript). In this manuscript, the function and release of miR-3613-3p are examples of Ago2/CAV1-dependent regulation in tumor metastasis. We have toned down the statements in the manuscript that the pro-tumorigenic effects of the Ago2/Cav1 interaction are solely mediated by miR-3613-3p. We agree with referee #2 that the experiment of binding site mutation is essential for studying miR-3613-3p regulation of SCAI but it is beyond the scope of this manuscript. We will pursue in the direction of miR-3613-3p function in tumor metastasis and will publish the finding in a separate paper.

[Figure condensed and complemented for clarity]

According to the suggestion of referees, the results of indirect association between GW182 and CAV1 via Ago2 are removed from Fig. 2 for clarity and the results of additional experiments addressing Ago2 association with CAV1, endosome, and ER and the dependence of Ago2/CAV1 interaction are added in Fig. 2 for solidifying the findings and claims (Fig. 2C and 2D of the revised manuscript).

Minor points include the need for thorough English editing for accuracy, addressing typos (e.g., "SCAISCAI" on page 8), defining the term "MF" upon first use, and avoiding redundancy in defining indole-3-acetic acid (IAA) twice. To enhance conciseness, some experimental details could be relocated to the methods section, for instance, the description of mouse injections (e.g., "At 2 weeks after implantation, the mice were injected with phosphate-buffered saline (PBS) or IAA for 3 weeks (Fig. 5Ai)" on page 10).

ANS 11:

[Editing for accuracy and conciseness]

The manuscript has received proofreading and editing by a professional English language editor. According to the suggestion, some experimental conditions have been relocated to the method section and figure legends.

Referee #3:

Lin et al identify Argonaute-2 (Ago2) as a novel Caveolin-1 (Cav-1)-interacting protein. They find that deacetylation of K212 of Ago2 by Sirtuin2 (SIRT2) is necessary for binding to Cav-1. They describe how Cav-1-mediated sequestration of Ago2 at the plasma membrane promotes miR-3613-3p-mediated suppression of SCAI (suppressor of cancer cell invasion) in cancer cells. They provide evidence that the interaction between Cav-1 and Ago2 promotes tumor formation and metastasis in mice and it is associated with human carcinoma progression and metastasis. They show that the Cav-1/Ago2 interaction promotes the release of miRNA-3613-3p, through extracellular vesicles (EVs), from cancer cells both in culture and in tumor-bearing mice. Consistent with these findings, elevated plasma levels of EVs containing miRNA-3613-3p correlates with increased interaction between Cav-1 and Ago2 in human metastatic carcinomas, suggesting that miRNA-3613-3p in EVs may represent a biomarker of human metastatic carcinomas. They conclude that Cav-1-mediated sequestration of Ago2 at the plasma membrane controls the pro-metastatic functions of miRNA-3613-3p. The manuscript is well-written and easy to follow. The study is significant in the cancer field because it provides novel insights into the pro-tumorigenic role of Ago2-mediated miRNA regulation.

The following concerns need to be addressed:

Main points:

1) The use of the P2 peptide to determine the role of Cav-1 in Ago2 function is a concern. The P2 peptide is likely to prevent the interaction of Cav-1 with a number of other Cav-1-interacting proteins, in addition to Ago2. As such, in contrast to what stated by the authors, the results obtained using the P2 peptide (in Figures 2D, 2E, and in most of Figure 3) do not specifically link a lack of sequestration of Ago2 in caveolae at the plasma membrane to miRNA processing. The P2 peptide could alter miRNA function at the plasma membrane, including miR-3613-3p-mediated SCAI suppression, in an Ago2-independent manner. Similarly, the outcome of the functional studies described in Figure 4 using the P2 peptide could also be explained by a lack of interaction with Cav-1 of proteins other than Ago2.

ANS 12:

[Ensure the effects of disrupting Ago2/CAV1 interaction with Ago2 mutants]

We agree with the referee that P2 peptides may also interfere with CAV1 CSD-interacting proteins other than Ago2. In addition, peptides for in vivo applications are limited because of stability and tissue penetrability challenges. To avoid these scenarios, we used auxin-inducible degradation (AID) to enable efficient and specific blockage of Ago2/CAV1 interaction in cancer cells, A549Ago2-KO/HA-AID-Ago2Wt/HA-Ago2 Δ cells, and HCC1806Ago2-KO/HA-AID-Ago2Wt/HA-Ago2 Δ cells (Supplementary Fig. 4A of the revised manuscript) both in vivo and in vitro. Blocking Ago2/CAV1 interaction with IAA in A549Ago2-KO/HA-AID-Ago2Wt/HA-Ago2 Δ cells decreased miR-3613-3p, increased SCAI mRNAs (Supplementary Figs. 4Di and 4Dii of the revised manuscript), selectively impaired the suppression of miR-3613-3p mimics on SCAI mRNAs (Supplementary Fig. 4E of the revised manuscript), increase resistance to trypsin disassociation (Supplementary Fig. 4F of the revised manuscript), decrease anoikis resistance (Supplementary Fig. 4G of the revised manuscript), attenuate tumorsphere formation (Supplementary Fig.

4H of the revised manuscript), and suppress invasion (Supplementary Fig. 4I of the revised manuscript) of cancer cells. The same effects of HA-Ago2 Δ -mediated blockage of Ago2/CAV1 interaction and P2 peptides suggest that the altered miRNA expression and function (Fig. 3 of the revised manuscript) and cancer cell behaviors (Fig. 4 of the revised manuscript) by P2 peptides result from the disruption of Ago2/CAV1 interaction.

We mention that P2 peptides may also interfere with CAV1 CSD-interacting proteins other than Ago2 on page 7 and 10 of the manuscript. Confirming the effects of P2 peptides due to blocking Ago2/CAV1 interaction are described on page 7 and 10 of the text of the revised manuscript and the results are shown in Figure 2B and Supplementary Fig 4 of the revised manuscript.

Furthermore, we perform additional experiments with Ago2 mutants, Ago2 Δ and Ago2K212A, to block Ago2/CAV1 interaction in cancer cells and evaluate their effects on colocalization of CAV1 (Fig. 2C, 2D, and 6G of the revised manuscript), miRNA expression (Fig. 6H and 6I of the revised manuscript), SCAI expression (Fig. 6I and 6J of the revised manuscript), mesenchymal marker expression (Fig. 6K of the revised manuscript). All the results show that blocking Ago2/CAV1 interaction with P2 peptides, deletion of Ago2 CBM (Ago2 Δ), and Ago2 K212A substitution (Ago2K212A) has the same effects on miRNA expression/function and cancer cell behaviors, also supporting the effects of P2 peptides due to blocking Ago2/CAV1 interaction.

The results of additional experiments are described on page 7, 8, 13 of the text of the revised manuscript, are discussed on page 17 of the revised manuscript, and are shown in Figure 2C, 2D, 6G and Supplementary Figure 2E and 6F of the revised manuscript.

2) While the authors convincingly show that the interaction of Ago2 with Cav-1 at the plasma membrane plays an important role in tumor growth and metastasis, it is not clear whether these functional outcomes are mediated by the miR-3613-3p-dependent suppression of SCAI expression. Does forcing SCAI downregulation in their system rescue tumor growth and metastasis in cancer cells expressing the mutant forms of Ago2 (Ago2delta, Ago2K212A) that can not interact with Cav-1 and in which SCAI expression remains elevated?

ANS 13:

[miR-3613-3P/SCAI axis as an example of Ago2/CAV1 interaction-dependent regulation]

We observed that Ago2/CAV1 interaction is associated with tumor metastasis in both animal models and cancer patients. In this study, we would like to claim that through Ago2/CAV1 interaction, Ago2 selectively affects a group of miRNAs involved in metastasis and function of the group of miRNAs depends on Ago2/CAV1 interaction. As we mention above in the answer to point 1 of referee #2 (ANS 10), in this manuscript, the miR-3613-3p-mediated SCAI suppression is one of the examples of Ago2/CAV1-dependent regulation in tumor metastasis. We have toned down the statements in the manuscript that the pro-tumorigenic effects of the Ago2/Cav1 interaction are solely mediated by miR-3613-3p/SCAI. We agree with referee #3 that the experiment of forcing SCAI downregulation in cancer cells expressing the mutant forms of Ago2 is important to know whether the effects of the Ago2 mutation on tumor metastasis is largely mediated by SCAI upregulation but it is beyond the scope of this manuscript. We will pursue in the direction of miR-3613-3p/SCAI function in tumor metastasis and will publish the finding in a separate paper.

Minor points:

1) Quantification of IF data in Figure 2 should be included.

ANS 14:

[Quantification of imaging experiments]

The quantification of imaging experiments of the manuscript has been performed, plotted, analyzed for statistics, and present in figures 2Di-vi, 2G, 2I, 8E, 8F, and supplementary figures 2Hi and 8B of the revised manuscript with the number of replicates in figure legends.

2) Quantification of immunoblotting data in Figure 3H should be included.

ANS 15:

[Quantification of Western blots]

The quantification of Western blots of the manuscript has been performed, plotted, analyzed for statistics, and present in figures 2Aii, 2Bii, 2Eii, 6Fii, and supplementary figures 1A, 1C, 1D, 1E, 1F, 1Hii, 1I, 1J, 1K, 1L, 1Nii, 1O, 2Ciii, 2Gii, 2Hii, 3F, 3H, 4Bii, 4Cii, 6, 8A, 8D, 8E, 8F, 8G of the revised manuscript with the number of replicates in figure legends.

3) Figures 4Gi and 4Gii are mentioned on page 9 but there is no panel G in Figure 4.

ANS 16: The superfluous words “Figures 4Gi and 4Gii” are removed from the revised manuscript.

4) The authors should at least comment on how Ago2 moves to caveolar membranes.

ANS 17:

[Ago2 association with caveolar membranes]

Blocking Ago2/CAV1 interaction with deletion of the CBM domain of Ago2 (Fig. 2C, 2D, 2H, and 2I of the revised manuscript) or Ago2 K212A substitution (Fig. 6G and Supplementary Fig. 6F of the revised manuscript) decreased the association of Ago2 with plasma membranes and did not alter Ago2 association with endosomes and ER in cancer cells. We found that Ago2/CAV1 interaction is necessary for the plasma association of Ago2 in the cancer cells. CAV1 has been found predominantly at the plasma membrane and marginally in endosome and ER (Albacete-Albacete et al., 2020; Williams and Lisanti, 2004). It corresponds with our observation that blocking Ago2/CAV1 interaction disrupts Ago2 association with the plasma membranes, which contain abundant CAV1 (Fig. 2H, 2I and 6G of the revised manuscript), but does not significantly interfere with Ago2 association with endosomes and ER, which only contain slight CAV1, in the cancer cells (Fig. 2D, 6G and Supplementary Fig. 2E of the revised manuscript). Therefore, Ago2 may move to caveolar membranes, which are CAV1-enriched invaginations of the plasma membranes, via the interaction with CAV1. Indeed, deletion of the CBM domain of Ago2 decreases the association of Ago2 with the CAV1-enriched invaginations of the plasma membranes (Supplementary Fig. 2G of the revised manuscript).

These notions are discussed on page 17 of the revised manuscript.

Cross-comments from referee 1:

For the P2 peptide, I guess I was convinced by the authors acknowledging that it would disrupt other interactions with Cav1 and therefore complementing the peptide work with an Ago2 mutant that couldn't bind Cav1. However, perhaps there is a key experiment that was not done with the Ago2 mutant that Reviewer 3 thinks should be done?

ANS 18:

[Ensure the effects of disrupting Ago2/CAV1 interaction with Ago2 mutants]

As we mention above in the answer to major point 1 of referee#3 (ANS 12) and in the manuscript, we agree with the referees that P2 peptides may also interfere with CAV1 CSD-interacting proteins other than Ago2. To avoid the scenario, we used two Ago2 mutants, Ago2 Δ and Ago2K212A, to block Ago2/CAV1 interaction in the experiments without affecting other interactions with Cav1. Blocking Ago2/CAV1 interaction with Ago2 mutants decreased colocalization with CAV1 (Fig. 2C, 2D, 6G of the revised manuscript), decreased miR-3613-3p (Fig. 6i and Supplementary Figs. 4Di of the revised manuscript), increased SCAI (Fig. 6I, 6J and Supplementary Fig. 4Dii of the revised manuscript), selectively impaired the suppression of miR-3613-3p mimics on SCAI mRNAs (Supplementary Fig. 4E of the revised manuscript), increased resistance to trypsin disassociation (Supplementary Fig. 4F of the revised manuscript), decreased anoikis resistance (Supplementary Fig. 4G of the revised manuscript), attenuated tumorsphere formation (Supplementary Fig. 4H of the revised manuscript), suppressed mesenchymal marker expression (Fig. 6K of the revised manuscript), and suppressed invasion (Supplementary Fig. 4I of the revised manuscript) of cancer cells. The same effects of Ago2 mutant-mediated blockage of Ago2/CAV1 interaction and P2 peptides suggest that the altered miRNA expression/function (Fig. 3 of the revised manuscript) and cancer cell behaviors (Fig. 4 of the revised manuscript) by P2 peptides result from the disruption of Ago2/CAV1 interaction.

For the functional outcomes, the authors did not try to say the miR-3613-3p and SCAI were responsible for the in vivo metastasis functions and I think it would be too much to ask. They instead focused on it as a biomarker. I think if they want to investigate this as a function, that could be a separate paper. I would prefer to see the authors solidify the findings and claims that they have with much more quantitation.

ANS 19:

[Solidify the findings and claims with quantitation]

As we mention above in ANS 10 and ANS 13, we observed that Ago2/CAV1 interaction is associated with tumor metastasis in both animal models and cancer patients. In this study, we would like to claim that through Ago2/CAV1 interaction, Ago2 selectively affects a certain group of miRNAs involved in metastasis and the function of the group of miRNAs depends on Ago2/CAV1 interaction. MiR-3613-3p/SCAI is one of the examples of Ago2/CAV1-dependent regulation in tumor metastasis. We have toned down the statements in the manuscript that the pro-tumorigenic effects of the Ago2/Cav1 interaction are solely mediated by miR-3613-3p. We will pursue in the direction of miR-3613-3p/SCAI function in tumor metastasis and will publish the finding in a separate paper. As we mention above in ANS 1 and ANS 3, the

quantification of imaging experiments of the manuscript has been performed, plotted, analyzed for statistics, and present in figures 2Di-vi, 2G, 2I, 8E, 8F, and supplementary figures 2Hi and 8B with the number of replicates in figure legends. The quantification of Western blots of the manuscript has been performed, plotted, analyzed for statistics, and present in figures 2Aii, 2Bii, 2Eii, 6Fii, and supplementary figures 1A, 1C, 1D, 1E, 1F, 1Hii, 1I, 1J, 1K, 1L, 1Nii, 1O, 2Ciii, 2Gii, 2Hii, 3F, 3H, 4Bii, 4Cii, 6, 8A, 8D, 8E, 8F, 8G of the revised manuscript with the number of replicates in figure legends.

Cross-comments from referee 2:

1. I think the P2 peptide data supports that disruption of the interaction between Ago2 and CAV1 impair tumor growth and metastasis. Given that the mechanism of Ago2 and CAV1 interaction is well demonstrated, which is the main part of this paper, I do not think further work on the P2 peptide is really necessary.

2. I agree that functional analysis of mir-3613-3p and SCAI is necessary because the key finding of this work is that CAV1 sequestered Ago2 modulate the functions of specific miRNAs.

ANS 19:

[miR-3613-3P/SCAI axis as an example of Ago2/CAV1 interaction-dependent regulation]

As we mention above in ANS 10 and ANS 13, we observed that Ago2/CAV1 interaction is associated with tumor metastasis in both animal models and cancer patients. In this study, we would like to claim that through Ago2/CAV1 interaction, Ago2 selectively affects a certain group of miRNAs involved in metastasis and the function of the group of miRNAs depends on Ago2/CAV1 interaction. MiR-3613-3p/SCAI is one of the examples of Ago2/CAV1-dependent regulation in tumor metastasis. We agree with referee #2 and referee #3 that more evidences are required to claim that the pro-tumorigenic effects of the Ago2/Cav1 interaction are solely mediated by miR-3613-3p/SCAI. Therefore, we have toned down the statements in the manuscript that the pro-tumorigenic effects of the Ago2/Cav1 interaction are solely mediated by miR-3613-3p. Further, we will pursue in the direction of miR-3613-3p/SCAI function in tumor metastasis and will publish the finding in a separate paper.

Cross-comments from referee 3:

1) Among the experiments in which the authors have used the P2 peptide to disrupt the Ago2/Cav-1 interaction (which likely disrupts also the interaction of Cav-1 with other Cav-1-interacting proteins), the miRNA array data (Fig 3A) and the data on mesenchymal markers (Fig 3H) have not been confirmed using the Ago2 mutant that can not interact with Cav-1. The authors should at least confirm the data of Fig 3A and 3H with the Ago2 mutant that can not interact with Cav-1.

ANS 20:

[Perform additional experiments with Ago2 mutants]

Among the experiments in which we have used the P2 peptide to disrupt the Ago2/Cav-1 interaction, we have performed additional experiments with Ago2 mutants, Ago2 Δ and/or Ago2K212A, including (1) colocalization with CAV1 (Fig. 2C, 2D, 6G), (2) miRNA array assay (Fig. 6H), (3) miR-3613-3p expression (Fig. 6I and Supplementary Figs. 4Di), (4) SCAI expression (Fig. 6I, 6J and Supplementary Fig. 4Dii), (5) suppression of miR-3613-3p mimics on SCAI mRNAs (Supplementary Fig. 4E), (6) resistance to trypsin disassociation (Supplementary Fig. 4F), (7) anoikis resistance (Supplementary Fig. 4G), (8) tumorsphere formation (Supplementary Fig. 4H), (9) mesenchymal marker expression (Fig. 6K), and (10) invasion (Supplementary Fig. 4I) of cancer cells.

With miRNA array analysis, we analyzed and compared miRNA fluctuation resulting from disruption of Ago2/CAV1 interaction by P2 peptides, deletion of Ago2 CBM, and Ago2 K212A substitution. There are 10 shared up-regulated miRNAs and 9 shared down-regulated miRNAs in P2-treated A549, A549Ago2-KO/HA-Ago2 Δ (Ago2Dm), and A549Ago2-KO/HA-Ago2K212A (Ago2K212A) cancer cells (Fig. 6Hi of the revised manuscript), including down-regulated miR-3613-3p and up-regulated miR-6126 (Fig. 6Hii of the manuscript).

The results of experiments with Ago2 mutants are described on page 7, 8, 10, and 13 of the text of the revised manuscript, are discussed on page 17 of the revised manuscript, and are shown in Figure 2C, 2D, 6G-6K and Supplementary Figure 4D-4I.

2) I agree with Reviewer #2 and remain of the idea that while the authors convincingly show that the interaction of Ago2 with Cav-1 at the plasma membrane plays an important role in tumor growth and metastasis, it is not clear whether these functional outcomes are mediated by the miR-3613-3p-dependent suppression of SCAI expression.

ANS 19:

[miR-3613-3P/SCAI axis as an example of Ago2/CAV1 interaction-dependent regulation]

As we mention above in ANS 10 and ANS 13, we observed that Ago2/CAV1 interaction is associated with tumor metastasis in both animal models and cancer patients. In this study, we would like to claim that through Ago2/CAV1 interaction, Ago2 selectively affects a certain group of miRNAs involved in metastasis and the function of the group of miRNAs depends on Ago2/CAV1 interaction. MiR-3613-3p/SCAI is one of the examples of Ago2/CAV1-dependent regulation in tumor metastasis. We agree with referee #2 and referee #3 that more evidences are required to claim that the pro-tumorigenic effects of the Ago2/Cav1 interaction are solely mediated by miR-3613-3p/SCAI. Therefore, we have toned down the statements in the manuscript that the pro-tumorigenic effects of the Ago2/Cav1 interaction are solely mediated by miR-3613-3p. Further, we will pursue in the direction of miR-3613-3p/SCAI function in tumor metastasis and will publish the finding in a separate paper.

References

Albacete-Albacete, L., Navarro-Lerida, I., Lopez, J.A., Martin-Padura, I., Astudillo, A.M., Ferrarini, A., Van-Der-Heyden, M., Balsinde, J., Orend, G., Vazquez, J., *et al.* (2020). ECM deposition is driven by caveolin-1-dependent regulation of exosomal biogenesis and cargo sorting. *J Cell Biol* 219.

Barman, B., and Bhattacharyya, S.N. (2015). mRNA Targeting to Endoplasmic Reticulum Precedes Ago Protein Interaction and MicroRNA (miRNA)-mediated Translation Repression in Mammalian Cells. *J Biol Chem* 290, 24650-24656.

Eitan, E., Zhang, S., Witwer, K.W., and Mattson, M.P. (2015). Extracellular vesicle-depleted fetal bovine and human sera have reduced capacity to support cell growth. *J Extracell Vesicles* 4, 26373.

Lin, M.C., Chen, S.Y., He, P.L., Luo, W.T., and Li, H.J. (2017a). Transfer of Mammary Gland-forming Ability Between Mammary Basal Epithelial Cells and Mammary Luminal Cells via Extracellular Vesicles/Exosomes. *J Vis Exp* e55736.

Lin, M.C., Chen, S.Y., Tsai, H.M., He, P.L., Lin, Y.C., Herschman, H., and Li, H.J. (2017b). PGE2 /EP4 Signaling Controls the Transfer of the Mammary Stem Cell State by Lipid Rafts in Extracellular Vesicles. *Stem Cells* 35, 425-444.

McKenzie, A.J., Hoshino, D., Hong, N.H., Cha, D.J., Franklin, J.L., Coffey, R.J., Patton, J.G., and Weaver, A.M. (2016). KRAS-MEK Signaling Controls Ago2 Sorting into Exosomes. *Cell Rep* 15, 978-987.

Shankar, S., Tien, J.C., Siebenaler, R.F., Chugh, S., Dommeti, V.L., Zelenka-Wang, S., Wang, X.M., Apel, I.J., Waninger, J., Eyunni, S., *et al.* (2020). An essential role for Argonaute 2 in EGFR-KRAS signaling in pancreatic cancer development. *Nat Commun* 11, 2817.

Shelke, G.V., Lasser, C., Gho, Y.S., and Lotvall, J. (2014). Importance of exosome depletion protocols to eliminate functional and RNA-containing extracellular vesicles from fetal bovine serum. *J Extracell Vesicles* 3.

Stalder, L., Heusermann, W., Sokol, L., Trojer, D., Wirz, J., Hean, J., Fritzsche, A., Aeschmann, F., Pfanzagl, V., Basselet, P., *et al.* (2013). The rough endoplasmic reticulum is a central nucleation site of siRNA-mediated RNA silencing. *EMBO J* 32, 1115-1127.

Williams, T.M., and Lisanti, M.P. (2004). The caveolin proteins. *Genome Biol* 5, 214.

Dear Dr. Li,

Thank you for the submission of your revised manuscript. We have now received the enclosed reports from all referees and I am happy to say that all support its publication now.

Only a few more minor editorial requests will need to be addressed before we can proceed with the official acceptance of your manuscript.

- Please remove all figures from the ms file. The legends need to stay though. All main and EV figures need to be uploaded as individual files.
- Please reduce the number of keywords to 5.
- Please correct the "Data and materials availability" subheading to "Data Availability Section". This section needs to be moved to the end of the materials and methods.
- Please correct the conflict of interest subheading to "Disclosure Statement and Competing Interests".
- Please remove the author credits from the ms file. All credits need to be entered during online ms submission.
- Please correct the REFERENCE FORMAT: it needs to be alphabetical, et al should be used after 10 author names. The EMBO reports reference style is also in EndNote.
- Please complete our author checklist, which you can download from our author guidelines <<https://www.embopress.org/page/journal/14693178/authorguide>>. The completed author checklist will also be part of the transparent peer-review process file.
- Please list all funding information also during ms submission in our online system.
- We replaced Supplementary Information with Expanded View (EV) Figures and Tables that are collapsible/expandable online. A maximum of 5 EV Figures can be typeset. EV Figures should be cited as "Figure EV1, Figure EV2" etc... in the text and their respective legends should be included in the main text after the legends of regular figures. Callouts of the supplementary figures will need to be updated in the ms text accordingly
- For the figures that you do NOT wish to display as Expanded View figures, they should be bundled together with their legends in a single PDF file called *Appendix*, which should start with a short Table of Content with page numbers. Appendix figures should be referred to in the main text as: "Appendix Figure S1, Appendix Figure S2" etc. See detailed instructions regarding expanded view here: <<https://www.embopress.org/page/journal/14693178/authorguide#expandedview>>
- The manuscript sections should be in the following order: Title page - Abstract & Keywords - Introduction - Results - Discussion - Materials & Methods - Data Availability - Acknowledgments - Disclosure Statement & Competing Interests - References - Figure Legends - Tables with legends - Expanded View Figure Legends.

I would like to suggest some changes to the abstract that also needs to be written in present tense. Please let me know whether you agree with the following:

Ago2 differentially regulates oncogenic and tumor-suppressive miRNAs in cancer cells. This discrepancy suggests a secondary event regulating Ago2/miRNA action in a context-dependent manner. We show here that a positive charge of Ago2 K212, that is preserved by SIR2-mediated Ago2 deacetylation in cancer cells, is responsible for the direct interaction between Ago2 and Caveolin-1 (CAV1). Through this interaction, CAV1 sequesters Ago2 on the plasma membrane and regulates miRNA-mediated translational repression in a compartment-dependent manner. Ago2/CAV1 interaction plays a role in miRNA-mediated mRNA suppression and in miRNA release via extracellular vesicles (EVs) from tumors into the circulation, which can be used as a biomarker of tumor progression. Increased Ago2/CAV1 interaction with tumor progression promotes aggressive cancer behaviors, including metastasis. Ago2/CAV1 interaction acts as a secondary event in miRNA-mediated suppression and increases the complexity of miRNA actions in cancer.

Both the title and abstract contain general terms such as "regulate" and "play a role". It would be better if title and abstract could be more specific, if possible. Also what kind of cancer type is used? If possible, please replace the general terms with more specific words. Thank you.

EMBO press papers are accompanied online by A) a short (1-2 sentences) summary of the findings and their significance, B) 2-3 bullet points highlighting key results and C) a synopsis image that is exactly 550 pixels wide and 200-600 pixels high (the height is variable). You can either show a model or key data in the synopsis image. Please note that text needs to be readable at the final size. Please send us this information along with the final manuscript.

Referee #1:

Overall, the authors have done a good job at responding to my comments, adding many quantitations, performing new experiments, and revising text.

Referee #2:

The authors have adequately addressed my concerns, and I believe the manuscript is suitable for publication.

Referee #3:

The study is significant because it provides novel insights into the pro-tumorigenic role of Ago2-mediated miRNA regulation. The authors have addressed my previous concerns. The revised manuscript is significantly improved.

All editorial and formatting issues were resolved by the authors.

Dr. Hua-Jung Li
National Health Research Institutes
Institute of Cellular and System Medicine
35 Keyan Road, R2-5031
Zhunan, Miaoli County, Taiwan 35053
Taiwan

Dear Dr. Li,

I am very pleased to accept your manuscript for publication in the next available issue of EMBO reports. Thank you for your contribution to our journal.

Yours sincerely,
